# Scalable Benchmarking and Robust Learning for Noise-Free Ego-Motion and 3D Reconstruction from Noisy Video

**Xiaohao Xu[1], Tianyi Zhang[2], Shibo Zhao[2], Xiang Li[2], Sibo Wang[1], Yongqi Chen[1], Ye Li[1], Bhiksha Raj[2], Matthew Johnson-Roberson[2], Sebastian Scherer[2], Xiaonan Huang[1]**
[1] University of Michigan, Ann Arbor, [2] Carnegie Mellon University

## Abstract

We aim to redefine robust ego-motion estimation and photorealistic 3D reconstruction by addressing a critical limitation: the reliance on noise-free data in existing models. While such sanitized conditions simplify evaluation, they fail to capture the unpredictable, noisy complexities of real-world environments. Dynamic motion, sensor imperfections, and synchronization perturbations lead to sharp performance declines when these models are deployed in practice, revealing an urgent need for frameworks that embrace and excel under real-world noise. To bridge this gap, we tackle **three core challenges**: *scalable data generation*, *comprehensive benchmarking*, and *model robustness enhancement*. **First**, we introduce a scalable noisy data synthesis pipeline that generates diverse datasets simulating complex motion, sensor imperfections, and synchronization errors. **Second**, we leverage this pipeline to create *Robust-Ego3D*, a benchmark rigorously designed to expose noise-induced performance degradation, highlighting the limitations of current learning-based methods in ego-motion accuracy and 3D reconstruction quality. **Third**, we propose *Correspondence-guided Gaussian Splatting (CorrGS)*, a novel method that progressively refines an internal clean 3D representation by aligning noisy observations with rendered RGB-D frames from clean 3D map, enhancing geometric alignment and appearance restoration through visual correspondence. Extensive experiments on synthetic and real-world data demonstrate that *CorrGS* consistently outperforms prior state-of-the-art methods, particularly in scenarios involving rapid motion and dynamic illumination.[1]

## 1 Introduction

The pursuit of reliable ego-motion estimation and photorealistic 3D reconstruction remains a fundamental challenge in computer vision and autonomous systems, particularly in the presence of real-world complexities such as sensor noise and dynamic motion. The current landscape of dense Neural SLAM (Simultaneous Localization and Mapping) research has made significant strides in noise-free, controlled settings using advanced neural representations like Neural Radiance Fields (NeRF) (Rosinol et al., 2022) and Gaussian Splatting (Kerbl et al., 2023). Despite their promise, the generalization of these methods to noisy, unstructured environments—reflecting realistic deployment conditions—remains largely unexplored (Tosi et al., 2024), highlighting the critical importance of studying the noise resilience of models to ensure robust performance in real-world scenarios.

We envision autonomous systems capable of robustly navigating and reconstructing 3D scenes even under adverse conditions, such as sensor degradation, unpredictable disturbances, or rapid camera motion. This vision demands overcoming critical limitations of existing methods that assume clean, synthetic datasets (Straub et al., 2019; Dai et al., 2017a), which often ignore practical sensing imperfections. Real-world datasets, while valuable, face issues of scalability, annotation quality, and capturing diverse degraded scenarios (Sturm et al., 2012; Schöps et al., 2019; Zhao et al., 2024). To bridge this gap, we propose a fundamental shift toward understanding and benchmarking models under generalized noisy conditions, enabling a systematic evaluation of their robustness.

---

[1]Project repo: `https://github.com/Xiaohao-Xu/SLAM-under-Perturbation`

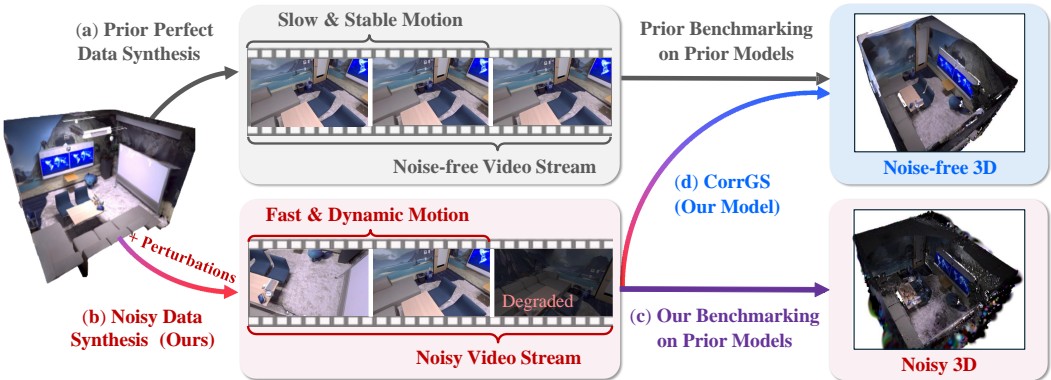

Figure 1: **Towards robust ego-motion and photorealistic 3D reconstruction.** (**a**) Previous approaches rely on synthetic datasets with ***perfect conditions*** (noise-free and smooth motion). (**b**) Real-world data introduce inherent noise and complexities. We present a customizable noisy data synthesis pipeline to evaluate methods under realistic ***noisy conditions***. (**c**) Our ***Robust-Ego3D*** benchmark reveals that existing methods produce ***noisy 3D reconstructions*** from ***noisy, sparse-view videos***, whereas (**d**) our proposed ***CorrGS*** achieves ***photorealistic, noise-free 3D reconstruction***.

To advance this vision, we pose three key research questions: **1**) **How to synthesize noisy data at scale? 2**) **How well do current SOTA models perform under noisy conditions? 3**) **How can we enhance model robustness against complex perturbations**? To address these questions, we present a comprehensive framework for robust ego-motion estimation and photorealistic 3D reconstruction in generalized noisy conditions, with **the following contributions**:

**1**) **Scalable noisy data generation pipeline**. We first present a comprehensive taxonomy of RGB-D sensing perturbations for mobile systems and develop a scalable noisy data synthesis pipeline that transforms clean 3D meshes into challenging datasets (see Fig. 1b), supporting sensor configurations from single to distributed multi-sensor setups. The datasets offer precise 3D map and trajectory ground-truth, scalability, and customizable perturbations. Our pipeline supports a more thorough and cost-effective evaluation than noise-free benchmarks, bridging the gap to real-world testing.

**2**) ***Robust-Ego3D* benchmark for generalized noisy conditions.** Utilizing our noisy data synthesis pipeline, we instantiate *Robust-Ego3D*, a large-scale benchmark that supports extensive and customizable RGB-D perturbations, offering 124 perturbed settings for comprehensive evaluation. Unlike existing benchmarks that focus on noise-free conditions, *Robust-Ego3D* provides a challenging testbed for assessing model robustness. Our extensive experiments and theoretical analyses reveal that the most common challenges of existing models are: pose tracking failure under dynamic motion, and degradation of 3D reconstruction fidelity under imaging perturbations due to the lack of restoration mechanisms (see Fig. 1c). Several key insights emerge: **i)** no model demonstrates consistent robustness across all perturbations, **ii)** individual perturbations have similar effects in both isolated and mixed settings, and **iii)** highly correlated perturbations can be leveraged as proxies for efficient benchmarking. We hope *Robust-Ego3D* prompts researchers to rethink model robustness under generalized noisy conditions and question existing dataset and model assumptions.

**3**) **CorrGS: Correspondence-guided Gaussian Splatting.** To tackle robust pose tracking under complex motion and consistent color reconstruction in noisy conditions identified in our *Robust-Ego3D* benchmark, we propose CorrGS. CorrGS employs a Gaussian Splatting 3D representation, swiftly rendered into RGB-D frames. By comparing these rendered images with noisy observations, it establishes correspondences that enhance geometric alignment for robust pose learning. These correspondences also support online appearance restoration learning, enabling noise-free 3D reconstruction from noisy videos. CorrGS significantly outperforms prior state-of-the-art methods under fast motion, achieving accurate ego-motion and photorealistic 3D reconstruction from both synthetic (see Fig. 1d) and real-world noisy videos, even with dynamic illumination and rapid motion.

This work rethinks robustness in dense Neural SLAM by moving beyond ideal-condition accuracy to true resilience under noisy sensing. By challenging existing benchmarks and assumptions, we pave the way for developing adaptive models that consistently perform in diverse, degraded conditions.

## 2   RELATED WORK

**Dataset synthesis for robustness benchmarking.** Most robustness evaluations rely on synthetic datasets due to the difficulty of controlling real-world perturbations. A pioneering benchmark (Hendrycks & Dietterich, 2019) assessed image classification under common corruptions, and subsequent research extended this approach to various tasks, such as 2D/3D object detection (Michaelis et al., 2019; Kong et al., 2023), segmentation (Kamann & Rother, 2020; Li et al., 2023; Xu et al., 2022), and embodied navigation (Chattopadhyay et al., 2021; Yokoyama et al., 2022). In ego-motion estimation and photorealistic 3D reconstruction, the challenges go beyond image-level corruptions, also involving temporal variations in sensor noise and pose transformations.

**Robustness benchmarking.** Robust localization and mapping are crucial for reliable performance in dynamic environments (Cadena et al., 2016). Real-world datasets (Schubert et al., 2018; Helmberger et al., 2022; Tian et al., 2023; Ebadi et al., 2023) evaluate classical SLAM under conditions like low illumination and motion blur. While existing benchmarks like TartanAir (Wang et al., 2020) and SLAMBench (Bujanca et al., 2021) incorporate some noise injection for robustness evaluation, none comprehensively address the full range of RGB-D perturbations encountered by mobile agents. Our benchmark, *Robust-Ego3D*, introduces diverse, controllable perturbations for systematic evaluation of dense SLAM systems under varied noise, including sensor noise, motion deviations, and synchronization issues. Unlike prior work focused on localization accuracy in classical SLAM, we assess dense Neural SLAM models considering both localization and photorealistic mapping, pioneering robustness evaluation to drive future research in this evolving field.

**Dense Neural SLAM model**. Classical SLAM models, *e.g.*, ORB-SLAM3 (Campos et al., 2021), excel at estimating ego-motion but produce only sparse maps. In contrast, dense non-neural SLAM models (Newcombe et al., 2011; Dai et al., 2017b) provide dense reconstructions but struggle to capture fine appearance details and textures. Dense Neural SLAM models overcome these limitations by reconstructing both geometry and appearance, enabling high-fidelity, photorealistic 3D rendering. Implicit neural representations, *e.g.*, NeRF (Rosinol et al., 2022), have been integrated for high-quality textured reconstruction (Sucar et al., 2021), with significant advancements in representation techniques (Wang et al., 2023; Johari et al., 2023; Zhu et al., 2022; Yang et al., 2022; Sandström et al., 2023). Additionally, some methods decouple learning-based pose estimation from dense mapping (Teed & Deng, 2021; Zhang et al., 2023). Building on the success of 3D Gaussian Splatting (Kerbl et al., 2023), recent approaches have adapted this technique to SLAM (Keetha et al., 2024; Matsuki et al., 2024), enabling efficient RGB-D rendering from reconstructed maps. Despite these advancements, most Neural models focus primarily on noise-free assumptions, highlighting a gap in pose tracking and 3D scene restoration under challenging conditions.

## 3   NOISY DATA SYNTHESIS WITH CUSTOMIZABLE PERTURBATIONS

### 3.1   PROBLEM FORMULATION

We address the challenge of robust ego-motion estimation and photorealistic 3D reconstruction from noisy data using a structured approach involving three interconnected components: noisy data synthesis, model estimation, and evaluation with feedback (see Fig. 2).

**Noisy data synthesis (from clean 3D to noisy video).** Perturbed observations $\mathbf{z}_{1:t}$ and trajectories $\mathbf{x}_{1:t}$ are generated from a clean 3D representation $\mathbf{m}$, a clean trajectory $\mathbf{x}_{1:t}^0$, and perturbations $\xi_{1:t}$:

$$p(\mathbf{z}_{1:t}, \mathbf{x}_{1:t} \mid \mathbf{m}, \mathbf{x}_{1:t}^0, \xi_{1:t}). \qquad (1)$$

**Model estimation (from noisy video to noise-free ego-motion and 3D reconstruction).** From noisy observations $\mathbf{z}_{1:t}$, the model estimates the 3D scene $\hat{\mathbf{m}}$ and the trajectory $\hat{\mathbf{x}}_{1:t}$:

$$p(\hat{\mathbf{m}}, \hat{\mathbf{x}}_{1:t} \mid \mathbf{z}_{1:t}). \qquad (2)$$

**Evaluation and feedback.** The estimates $\hat{\mathbf{m}}$ and $\hat{\mathbf{x}}_{1:t}$ are compared with ground truth $\mathbf{m}$ and $\mathbf{x}_{1:t}$, revealing perturbation-specific weaknesses caused by $\xi_{1:t}$.

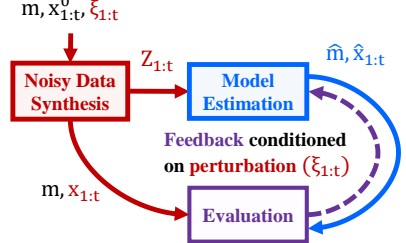

Figure 2: **Framework for noisy data synthesis, model estimation, and evaluation.**

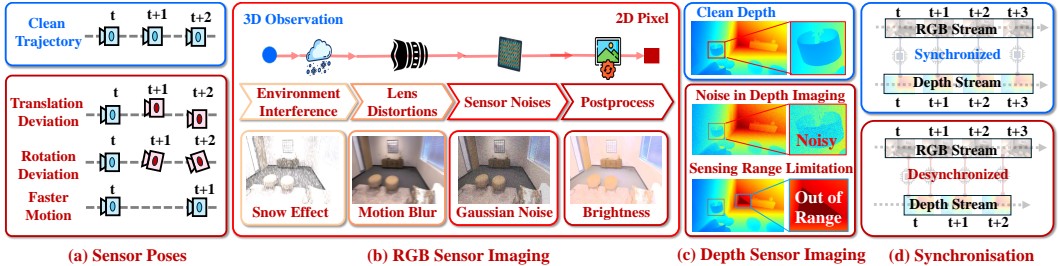

Figure 3: **Perturbation taxonomy for RGB-D sensing**.

## 3.2 PERTURBATION TAXONOMY FOR RGB-D SENSING SYSTEMS

**Perturbation sources.** As shown in Fig. 3, perturbations for RGB-D sensing stem from pose deviations, imaging inaccuracies, and sensor desynchronization. All perturbations are constructed using fundamental physical and kinematic modeling, *e.g.*, motion deviations are derived from rigid-body transformations. We briefly illustrate perturbations here; details are in Appendix Sec. A.

**(a) Perturbations on sensor pose motion.** Real-world mobile platforms can exhibit diverse and dynamic motions that challenge model robustness. As shown in Fig. 3a, we categorize motion perturbations into *Motion Deviations* (by applying a combination of rotation $\Delta\mathbf{R} \in SO(3)$ and translation $\Delta\mathbf{t} \in \mathbb{R}^3$ perturbations) and *Faster Motion Effect* .

**(b) Perturbation on RGB sensor imaging.** Imaging corruptions, such as motion blur and high illumination, are common in real-world data collection, affecting image quality. As shown in Fig. 3b, sixteen RGB imaging perturbations are modeled to mimic error sources that arise throughout the entire imaging procedural—from 3D scene capture to final image output. These perturbations stem from *environmental interference* affecting light transmission, *lens distortions* leading to blurring, *sensor-induced noise*, and artifacts introduced during *post-processing*.

**(c) Perturbation on depth sensor imaging.** We observed a significant discrepancy in depth distribution between the simulated SLAM benchmark (Replica (Straub et al., 2019)) and the real-world dataset (TUM-RGBD (Schubert et al., 2018)), with TUM-RGBD missing 25% of depth data compared to just 0.39% in Replica. To address this, we propose a set of depth perturbation operations (see Fig. 3c): *Gaussian Noise* to mimic depth noise; *Edge Erosion* and *Random Missing Depth* for missing data; *Range Clipping* for limited depth sensor perception range.

**(d) Perturbation on multi-sensor synchronization.** To emulate sensor delays when multiple sensors within an RGB-D sensing system are not well-synchronized (*e.g.,* due to varied signal sampling frequency), we introduce temporal misalignment between multiple sensor streams.

**Perturbation propagation and composition.** For an RGB-D sensing system, the specific composition is an ***ordered transformation*** derived from the generic formulation (see Eq. 1):

$$\mathbf{z}_{1:t}, \mathbf{x}_{1:t} = \left(\mathcal{F}_{\text{desync}} \circ \mathcal{F}_{\text{imaging}} \circ \mathcal{F}_{\text{render}} \circ \mathcal{F}_{\text{motion}}\right)\left(\mathbf{m}, \mathbf{x}^0_{1:t}, \xi_{1:t}\right), \tag{3}$$

where $\mathcal{F}_{\text{motion}}$ introduces motion perturbations, $\mathcal{F}_{\text{render}}$ generates clean observations via photorealistic rendering, $\mathcal{F}_{\text{imaging}}$ adds imaging perturbations, and $\mathcal{F}_{\text{desync}}$ simulates multi-sensor desyncronization.

**Perturbation mode and severity.** Perturbations are examined in two modes: static, with constant severity, and dynamic, with frame-to-frame variations. Severity levels are based on real-world data (*e.g.*, real depth distribution) or physics-based models (Kong et al., 2023; Hendrycks & Dietterich, 2019), and can be adjusted to progressively evaluate model performance under varying difficulties.

## 3.3 SCALABLE NOISY DATA SYNTHESIS PIPELINE

We present a scalable noisy data synthesis pipeline integrating customizable RGB-D perturbations (see Fig. 4). The key innovation lies in its ***customizability*** and ***physically-based modeling***, enabling the generation of photorealistic noisy data with controllable perturbations. The initial phase involves configuring the sensor setup, original trajectory, and 3D scene. Next, the clean and stable trajectories are processed by the motion perturbation composer to introduce deviations and faster motion effects. The render generates clean but unstable sensor data streams

from the 3D scene mesh, perturbed trajectory, and sensor configurations. Subsequently, sensor-related perturbations, including imaging corruptions and multi-sensor desynchronization, are composed, resulting in noisy and unstable sensor streams. This noisy data, with perturbed sensor streams as inputs and clean 3D and perturbed trajectories as ground truth, allows for perturbation-conditioned model performance evaluation.

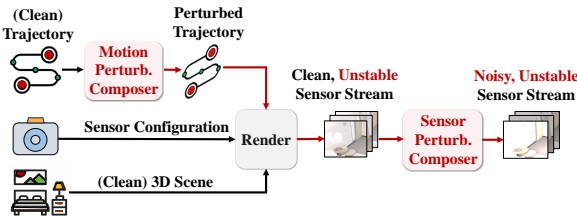

Figure 4: **Noisy data synthesis pipeline.**

## 4 RETHINKING MODEL ROBUSTNESS UNDER PERTURBATIONS

Leveraging our noisy data synthesis pipeline, we instantiate the **Robust-Ego3D** benchmark, designed as the **first comprehensive benchmark** that supports model evaluation under diverse noisy sensing conditions. *Robust-Ego3D* surpasses existing synthetic benchmarks in both diversity and

Table 1: Synthetic SLAM benchmark comparison.

| Benchmark | #Seq | #Perturb Setting | Perturbation Category | | | | Editable |
|---|---|---|---|---|---|---|---|
| | | | RGB | Motion | Depth | Sync. | |
| Replica | 8 | 0 | ✗ | ✗ | ✗ | ✗ | ✗ |
| TartanAir | 30 | 8 | ✓ | ✓ | ✗ | ✗ | ✗ |
| *Robust-Ego3D* | 1,000 | 124 | ✓ | ✓ | ✓ | ✓ | ✓ |

scope, offering unprecedented granularity (see Table 1). It offers *editable perturbation capabilities* and covers 124 distinct RGB-D perturbation settings across 1,000 videos. This level of granularity in perturbation settings is exceptional and opens the door to more systematic evaluations. See Appendix Sec. B for detailed *Robust-Ego3D* benchmark setup, hyperparameters, and statistics.

Next, we re-evaluate representative and top dense Neural SLAM models—including iMAP (Sucar et al., 2021), Nice-SLAM (Zhu et al., 2022), CO-SLAM (Wang et al., 2023), GO-SLAM (Zhang et al., 2023), and SplaTAM (Keetha et al., 2024)—under perturbations (detailed in Appendix Sec. B.3). While we focus on dense Neural SLAM methods which support photorealistic 3D reconstruction, we include ORB-SLAM3 (Campos et al., 2021), a strong ORB-feature-based model, for reference (see Appendix Fig. D13). The primary evaluation metric is Absolute Trajectory Error (ATE) (Prokhorov et al., 2019), supplemented by PSNR, Depth L1 Loss, and qualitative comparisons on 3D mesh and rendered RGB-D for textured 3D quality evaluation. To ensure robustness, each experiment was repeated three times across eight scenes, averaging results over 24 trials per perturbation.

### 4.1 BENCHMARKING UNDER INDIVIDUAL PERTURBATIONS

**Performance under RGB imaging corruptions** (Table 2). **1)** *Clean vs. perturbed.* The models evaluated generally show robustness to most RGB imaging corruptions. Even for iMAP, which uses a simple fully-connected network to implicitly encode the scene, the trajectory estimation performance under most perturbations is comparable to that under clean conditions, though it still degrades. **2)** *Neural models' robustness partly stems from the local denoising property of neural representation.* For example, iMAP, with NeRF-based representation, exhibits an intriguing local denoising property under *Spatter Effect*, producing RGB images and textured mesh nearly free of spatter noise (please see Sec. D.3 for details).

**3)** *Comparison across perturbations categories.* Different perturbations impact performance to varying degrees, with environmental effects posing the most significant challenge, followed by sensor noise, while image post-processing perturbations like *JPEG Compression* have a minor influence. **4)** *Dynamic vs. static mode.* Dynamic perturbations are more challenging than static ones, complicating model deployment in dynamic environments. Static perturbations, with consistent intensity across frames, simplify the optimization of Neural methods, whereas dynamic perturba-

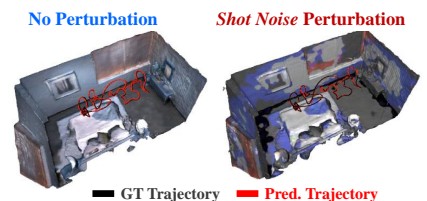

Figure 5: Effect of *Shot Noise* on 3D reconstruction of Nice-SLAM.

tions require cross-domain pose and map optimization. **5)** *Effect of depth.* By comparing GO-SLAM (Mono) and GO-SLAM (RGB-D), we find that adding depth information increases robustness to RGB imaging perturbations. While the monocular version GO-SLAM exhibits slightly lower error in clean settings, the RGB-D version outperforms it under perturbed conditions. **6)** *3D reconstruction quality.*

Table 2: Performance (ATE↓ (m)) under static (**Top**) and dynamic (**Bottom**) RGB imaging perturbations. The best-performing methods are highlighted with  first , second , and third .

| Method | Clean | Perturbed | | Blur Effect | | | | Noise Effect | | | | Environmental Interference | | | | Post-processing | | | |
|---|---|---|---|---|---|---|---|---|---|---|---|---|---|---|---|---|---|---|---|
| | | Mean | Max | Motion | Defocus | Gaussian | Glass | Gaussian | Shot | Impulse | Speckle | Fog | Frost | Snow | Spatter | Bright | Contrast | Jpeg | Pixelate |
| GO-SLAM (Mono) | 0.0039 | 0.0903 | 0.7207 | 0.0151 | 0.0052 | 0.0052 | 0.0089 | 0.0776 | 0.0456 | 0.0296 | 0.0190 | 0.2157 | 0.7207 | 0.1921 | 0.0859 | 0.0046 | 0.0047 | 0.0095 | 0.0046 |
| iMAP (RGB-D) | 0.1209 | 0.1568 | 0.3831 | 0.1424 | 0.1671 | 0.1811 | 0.0672 | 0.0278 | 0.0779 | 0.1710 | 0.1087 | 0.1913 | 0.1316 | 0.1665 | 0.1473 | 0.1903 | 0.3831 | 0.1884 | 0.1669 |
| Nice-SLAM (RGB-D) | 0.0147 | 0.0253 | 0.0654 | 0.0307 | 0.0151 | 0.0161 | 0.0188 | 0.0254 | 0.0377 | 0.0353 | 0.0151 | 0.0186 | 0.0160 | 0.0323 | 0.0320 | 0.0654 | 0.0161 | 0.0150 | 0.0145 |
| CO-SLAM (RGB-D) | 0.0090 | 0.0104 | 0.0125 | 0.0115 | 0.0096 | 0.0097 | 0.0097 | 0.0125 | 0.0101 | 0.0099 | 0.0105 | 0.0118 | 0.0113 | 0.0104 | 0.0098 | 0.0103 | 0.0112 | 0.0095 | 0.0094 |
| GO-SLAM (RGB-D) | 0.0046 | 0.0574 | 0.6271 | 0.0135 | 0.0052 | 0.0052 | 0.0090 | 0.0169 | 0.0140 | 0.0171 | 0.0100 | 0.1211 | 0.6271 | 0.0416 | 0.0164 | 0.0047 | 0.0054 | 0.0065 | 0.0050 |
| SplaTAM-S (RGB-D) | 0.0045 | 0.0062 | 0.0160 | 0.0160 | 0.0052 | 0.0049 | 0.0048 | 0.0054 | 0.0050 | 0.0044 | 0.0051 | 0.0085 | 0.0063 | 0.0048 | 0.0051 | 0.0038 | 0.0133 | 0.0044 | 0.0048 |
| GO-SLAM (Mono) | 0.0039 | 0.0933 | 0.7395 | 0.0155 | 0.0065 | 0.0060 | 0.0090 | 0.0509 | 0.0253 | 0.0306 | 0.0158 | 0.2668 | 0.7395 | 0.2254 | 0.0474 | 0.0066 | 0.0050 | 0.0298 | 0.0044 |
| iMAP (RGB-D) | 0.1209 | 0.1756 | 0.2873 | 0.1243 | 0.1042 | 0.2149 | 0.1221 | 0.1354 | 0.1170 | 0.1967 | 0.1576 | 0.2279 | 0.2873 | 0.2412 | 0.1528 | 0.2141 | 0.2576 | 0.1607 | 0.0955 |
| Nice-SLAM (RGB-D) | 0.0147 | 0.0214 | 0.0409 | 0.0157 | 0.0252 | 0.0359 | 0.0211 | 0.0288 | 0.0409 | 0.0146 | 0.0155 | 0.0167 | 0.0211 | 0.0197 | 0.0187 | 0.0206 | 0.0155 | 0.0146 | 0.0170 |
| CO-SLAM (RGB-D) | 0.0090 | 0.0105 | 0.0117 | 0.0107 | 0.0095 | 0.0115 | 0.0093 | 0.0106 | 0.0103 | 0.0102 | 0.0098 | 0.0117 | 0.0116 | 0.0111 | 0.0109 | 0.0106 | 0.0111 | 0.0095 | 0.0097 |
| GO-SLAM (RGB-D) | 0.0046 | 0.0363 | 0.2213 | 0.0130 | 0.0057 | 0.0055 | 0.0078 | 0.0185 | 0.0117 | 0.0139 | 0.0098 | 0.1685 | 0.2213 | 0.0637 | 0.0166 | 0.0051 | 0.0052 | 0.0092 | 0.0049 |
| SplaTAM-S (RGB-D) | 0.0045 | 0.0080 | 0.0450 | 0.0191 | 0.0053 | 0.0052 | 0.0050 | 0.0058 | 0.0072 | 0.0044 | 0.0067 | 0.0062 | 0.0062 | 0.0450 | 0.0041 | 0.0054 | 0.0096 | 0.0046 | 0.0045 |

Figure 6: Effect of depth perturbations.

Figure 7: Effect of fast motion.

Fig. 5 shows that trajectory estimation accuracy does not always correlate with 3D reconstruction quality. While Nice-SLAM accurately reconstructs 3D geometry and trajectory under *Shot Noise* RGB perturbation, the noise severely degrades color reconstruction. This highlights the need for ***restoration mechanisms to ensure noise-tolerant and noise-free 3D reconstruction***. Furthermore, Appendix **Theorem A** demonstrates that restoration mechanisms can reduce gradient variability from noisy inputs, supporting more robust optimization. **7)** ***Learning-based vs. heuristic representation.*** Learning-based models show strong robustness to RGB imaging perturbations for pose tracking (Table 2), while ORB-SLAM3, relying on ORB features, suffers severe tracking loss.

**Performance under depth missing or noises.** Fig. 6 illustrates the effect of depth perturbations. **1)** ***Depth missing effects.*** Dense Neural SLAM models exhibit minimal performance degradation when faced with partial depth missing. This robustness can be explained by Appendix **Theorem B**, which demonstrates that missing depth data does not introduce noise into the optimization process. By excluding missing depth from the gradient calculation, these models maintain stable pose updates, which prevents significant errors from propagating. **2)** ***Depth noise effects.*** Noise introduced to depth significantly impacts all models, directly interfering with the depth loss term and disrupting optimization. Appendix **Theorem C** demonstrates that noisy depth leads to biased gradients, causing misaligned pose updates. This instability results in substantial performance degradation, as evidenced by the considerable increase in ATE and the failure of depth reconstruction, as shown in the qualitative depth rendering results of SplaTAM-S in Fig. 6. **3)** ***RGB vs. RGB & noisy depth.*** GO-SLAM's ATE rises from 0.46 cm to 3.78 cm with depth noise, making RGBD-version GO-SLAM perform worse than its monocular counterpart. This highlights the need for selective disregard of noisy inputs.

**Performance under fast motion.** Fig. 7 shows significant performance degradation in most models under fast motion. **1)** ***Robust Neural models***. The Neural model GO-SLAM performs robustly under fast motion, thanks to global optimization techniques like bundle adjustment and loop closure. **2)** ***Fragile Neural models.*** In contrast, other models like Nice-SLAM, CO-SLAM, and SplaTAM-S, which rely on RGB-D reconstruction loss for differentiable pose and map optimization, struggle, leading to failed pose optimization and degraded map reconstruction (see Fig. 7). As Appendix **Theorem D** demonstrates, large pose changes amplify gradients, causing inefficient optimization and potential divergence for differentiable pose optimization. **3)** ***Neural vs. non-neural.*** Overall, pure NeRF-based and Gaussian-Splat-based models evaluated show lower robustness to rapid motion changes compared to the non-neural model ORB-SLAM3 (whose results are in Appendix Fig. D13d), which leverages ORB descriptors for tracking, making it more resilient to such effects. Thus, to

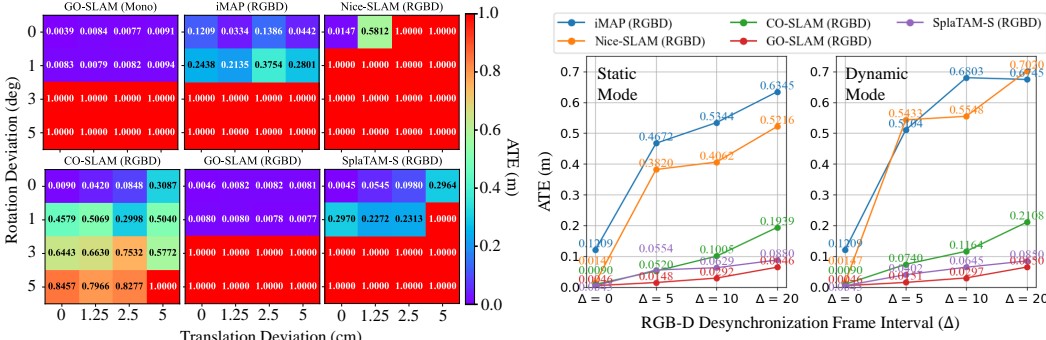

Figure 8: Effect of motion deviations.  Figure 9: Effect of RGB-D desynchronization.

improve robustness, combining the learnable components of Neural methods with ORB-SLAM3's resilience to fast motion could offer a more balanced approach.

**Performance under unstable sensor motion with deviations. 1)** *Small deviations cause great challenge.* As shown in Fig. 8, even minor pose deviations (*e.g.*, 2.5 cm in translation) significantly degrade trajectory estimation across most models. Combined translation and rotation deviations amplify errors, leading to complete pose tracking failure. **2)** *Failure due to invalid linear motion assumptions under dynamic motion.* By leveraging

Table 3: Effect of linear motion assumption on ATE↓ (cm)) of SplaTAM on original-speed (1×) clean videos. 'Iter': pose optimization iteration.

| Assump. | Iter. | O-0 | O-1 | O-2 | O-3 | O-4 | R-0 | R-1 | R-2 | Avg. |
|---|---|---|---|---|---|---|---|---|---|---|
| ✓ | 40 | **0.47** | **0.27** | **0.29** | **0.32** | **0.55** | **0.31** | **0.40** | **0.29** | **0.36** |
| ✗ | 40 | 3.59 | 2.42 | 2.48 | 0.58 | 2.39 | 3.30 | 25.82 | 4.83 | 5.68 |
| ✗ | 200 | 16.09 | 0.62 | 15.09 | 0.40 | 17.21 | 4.27 | 65.05 | 4.87 | 15.30 |

the assumption of smooth, linear sensor motion, models such as SplaTAM-S and CO-SLAM *perform well on standard datasets that exhibit this data bias*. However, this assumption does not hold in cases with abrupt movements. Table 3 demonstrates significant performance drops when this assumption is violated. Increasing pose optimization iterations further degrades performance, highlighting the sensitivity of these methods to the linear motion assumption. Appendix **Theorem E** shows that assuming linear motion under non-linear conditions leads to misaligned gradients and convergence failures in differentiable pose optimization, as the models fail to capture higher-order dynamics.

**Performance under poorly-synced RGB-D streams.** Fig. 9 shows that while increased desynchronization leads to performance drops, the degradation is less severe for methods like CO-SLAM, SplaTAM-S, and GO-SLAM. The relative performance ranking of top-performing methods remains consistent under both well-synced and desynchronized sensor settings, which suggests that enhancing performance in clean settings could potentially improve robustness under poorly-synced settings.

**Remark. 1)** *No universally robust model.* No single model handles all perturbations. The inconsistency between performance in clean and perturbed settings highlights the need for evaluating systems across diverse degraded conditions. **2)** *Synergy across methods.* Neural models excel in pose tracking under RGB imaging perturbations, while ORB-SLAM3 with feature matching is more robust to fast motion. This inspired our *CorrGS* method, marrying correspondence-based matching with neural representation. **3)** *Caution with model and dataset assumptions.* Assuming linear sensor motion can lead to overfitting to dataset biases, impairing the model's ability to generalize to diverse real-world scenarios. Our customizable noisy data generation pipeline helps reveal such model vulnerabilities by introducing varied perturbations, ensuring robustness beyond standard dataset biases.

## 4.2 FROM INDIVIDUAL TO MIXED PERTURBATIONS

We conduct case studies (see Table 4) on scene `Office-0` under static mode perturbations.

**Mixed perturbations generally lead to degraded performance, though not always.** We find that some perturbation combinations like *Snow Effect* and *Motion Blur* (**ii**) amplify errors, leading to higher ATE for GO-SLAM. However, other combinations, like adding *JPEG Compression* (**iv**), produce an effect similar to the single-perturbation setting and do not degrade performance.

Table 4: Mixed perturbation effect (ATE↓ (m)).

| Method | Clean | (i) | (ii) | (iii) | (iv) | (v) | (vi) |
|---|---|---|---|---|---|---|---|
| GO-SLAM | **0.004** | 0.095 | 0.512 | 0.620 | 0.356 | 0.397 | 0.627 |
| △ATE | 0.000 | +0.091 | **+0.417** | +0.108 | -0.264 | +0.041 | +0.230 |
| SplaTAM-S | **0.005** | 0.007 | 0.011 | 0.008 | 0.007 | 0.211 | 0.269 |
| △ATE | 0.000 | +0.002 | +0.004 | -0.003 | -0.001 | **+0.204** | +0.058 |

**Left to Right**: We progressively add: (i) *RGB Snow Effect*, (ii) *RGB Motion Blur*, (iii) *RGB Gaussian Noise*, (iv) *RGB JPEG*, & (v) *Depth Noise*, (vi) *RGB-D Desync*.

**Individual perturbations often yield similar performance degradation in isolated and mixed settings.** Introducing additional *Depth Noise* (**v**) significantly impacts SplaTAM-S, increasing ATE by 0.204 m and degrading rendered RGB quality (see Fig. 10). Likewise, coupled RGB corruptions (**ii**) strongly affect GO-SLAM. This suggests analyzing individual perturbations can expose model vulnerabilities in multi-perturbation scenarios.

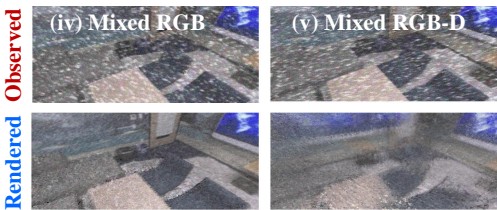

Figure 10: Rendering under mixed perturbations (same as Table 4) for SplaTAM-S.

### 4.3 Towards Streamlined Benchmarking via Perturbation Proxies

**Optimizing robustness benchmarking through correlation analysis.** Evaluating model robustness under large-scale perturbations can be resource-intensive, requiring numerous runs across varying perturbation types, severity levels, scenes, and models. However, by understanding correlations between different perturbations, we can streamline the model evaluation process.

**Streamlined benchmarking using representative perturbations.** We use Spearman's rank correlation (Spearman, 1961) to identify perturbations that closely mirror overall performance degradation, allowing efficient evaluation by focusing on representative perturbations. Fig. 11 shows that perturbations like *Gaussian Noise*, *Shot Noise*, *Impulse Noise*, and *Spatter* have high correlations with others, making them effective proxies for broader testing. While they may not capture every specific effect, they provide a rough estimate of performance under degraded RGB.

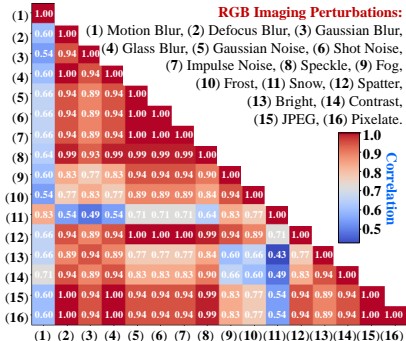

Figure 11: Correlation of performance (ATE) across static RGB perturbations.

## 5 CorrGS: Robust Learning under Noisy Video

Our benchmarking reveals that existing methods struggle with large motions and suffer color degradation in 3D reconstructions under RGB imaging perturbations. These insights motivate us to propose a more robust model, *i.e.*, Correspondence-guided Gaussian Splatting (CorrGS), to achieve ***reliable ego-motion estimation and noise-free dense 3D reconstruction, even from noisy, fast-motion videos***.

### 5.1 Method

**Overview.** CorrGS employs Gaussian Splatting combined with correspondence-guided differentiable optimization to deliver robust ego-motion estimation and photorealistic 3D reconstruction. Using the Gaussian-based map representation maintained to be noise-free, CorrGS renders RGB-D images at estimated poses. It then conducts differentiable optimization by calculating the differences between these rendered images and the observed noisy RGB-D frames. The integration of correspondence guidance is pivotal, as it enhances geometric alignment for accurate ego-motion estimation and facilitates appearance restoration for reconstructing noise-free, photorealistic 3D. Please see Appendix Sec. G for details on *CorrGS*, including schematic pipeline (Fig. G31) and pseudocode (Alg. 1).

**Correspondence-guided Pose Learning (CPL)** uses ***correspondence-initialized poses to improve the initialization for differentiable pose parameter learning***, enabling robust ego-motion estimation. **1)** We first calculate $N$ 3D correspondences, indexed as $i \in \{1, \dots, N\}$, by lifting 2D visual matches between rendered and observed RGB frames into 3D using the rendered and observed depth, respectively. This process establishes correspondences between observed noisy 3D points $\mathbf{p}_{o,i}$ and rendered 3D points $\mathbf{p}_{r,i}$. **2)** Then, we calculate the relative pose $\mathcal{P}_{rel} = \{\mathbf{R}_{rel}, \mathbf{t}_{rel}\}$ between rendered and observed points via Eq. 4. **3)** The pose initialization via correspondence is obtained by multiplying the previous pose estimate with the relative pose $\mathcal{P}_{rel}$.

$$\mathcal{P}_{rel}^* = \arg \min_{\mathbf{R}_{rel}, \mathbf{t}_{rel}} \sum_{i=1}^{N} \rho \left( \|\mathbf{R}_{rel}\mathbf{p}_{r,i} + \mathbf{t}_{rel} - \mathbf{p}_{o,i}\|^2 \right), \tag{4}$$

where $\| \cdot \|$ denotes the Euclidean norm, and $\rho(\cdot)$ is the soft L1 loss function. The ***computational complexity*** of this optimization process is bounded by the image resolution, enabling low time cost.

Table 5: Comparison (ATE↓ (cm)) on sparse-view video.

| Method | LC | O-0 | O-1 | O-2 | O-3 | O-4 | R-0 | R-1 | R-2 | Avg. |
|---|---|---|---|---|---|---|---|---|---|---|
| GO-SLAM | ✓ | 20.02 | 0.44 | **Fail** | 0.40 | 0.67 | 0.71 | 0.50 | 0.43 | **Fail** |
| CO-SLAM | | 99.38 | 60.77 | 176.1 | 162.02 | 223.5 | 170.5 | 140.4 | 155.15 | 148.48 |
| SplaTAM | | 115.55 | 63.13 | 76.55 | 196.95 | 367.00 | 126.66 | 150.66 | 1006.01 | 262.81 |
| **CorrGS (Default)** | | **1.80** | **2.41** | **1.96** | **0.99** | **1.36** | **0.84** | **17.73** | **1.72** | **3.60** |
| CO-SLAM-L | | 55.52 | 5.23 | 6.28 | 3.80 | 10.76 | 11.08 | 104.03 | 3.91 | 25.08 |
| Hero-SLAM-L | | 0.77 | 1.10 | 1.97 | 1.33 | 82.3 | 22.5 | 1.83 | 0.80 | 14.08 |
| SplaTAM-L | | 1.59 | 0.71 | 0.39 | 0.34 | 0.52 | 0.93 | 81.11 | 1.73 | 10.9 |
| **CorrGS-L (Default)** | | **0.39** | **0.13** | **0.22** | **0.33** | **0.49** | **0.76** | **0.59** | **0.71** | **0.45** |
| CorrGS-L (w/o PQV) | | 1.80 | 0.83 | 0.43 | 0.29 | 0.45 | 1.06 | 88.64 | 1.79 | 11.91 |

LC indicates using loop closure. '-L' indicates using the same pose optimization iteration (200).

**w/o CPL**    **w/ CPL**

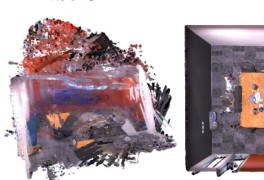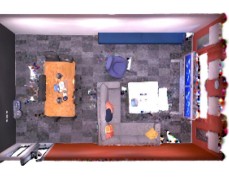

Figure 12: Ablation of CPL on 3D reconstruction from sparse-view video.

**4)** However, the initialization from Eq. 4 may fail when the two viewpoints are very close together. In such scenarios, it is challenging to accurately estimate the relative pose by optimizing the transformation between the point clouds of the two viewpoints, leading to higher RGB-D rendering error. To mitigate this, we propose a ***Pose Quality Verification*** step that rejects the initialization from Eq. 4 if it results in a higher rendering loss than a naive pose propagated from previous frames. **5)** The final pose is further refined using the differentiable rendering-based optimization.

**Correspondence-guided Appearance Restoration Learning (CARL).** **1)** CARL *mitigates color degradation by learning a restoration model* $f(\cdot;\theta)$ that maps noisy colors $\mathbf{C}_{o,i}$ of observed points to their clean counterparts $\mathbf{C}_{r,i}$, rendered from historical map which is maintained to be noise-free:

$$\theta^* = \arg\min_\theta \sum_{i=1}^{N} \|f(\mathbf{C}_{o,i};\theta) - \mathbf{C}_{r,i}\|^2 . \tag{5}$$

**2)** The learnt model is then applied to the observed image. It ensures 3D consistency and enhances pose tracking via a second round of CPL with the restored image.

**Theoretical insights on the benefits of *CorrGS*.** **1)** CPL enhances pose learning by generating more accurate pose estimates that are closer to the ground-truth. This accuracy reduces the magnitude of gradients and improves the conditioning of the Hessian matrix, thereby facilitating more stable and robust tracking. **2)** CARL employs a learned restoration mechanism to denoise perturbed color inputs, effectively mitigating noise-induced residual variability, which in turn stabilizes gradients and accelerates optimization convergence. See Appendix Sec. I for more details.

## 5.2 Pilot Study on Synthetic Noisy Sparse-view Video

**Metrics**. Performance is quantified using **1)** ATE for ego-motion accuracy, **2)** PSNR for simultaneous color restoration and reconstruction quality, and **3)** depth L1 loss for depth reconstruction accuracy.

**Noisy data synthesis setup.** We focus on illumination changes, leaving other perturbations for future work. **1)** To evaluate CPL's robustness under fast motion, we synthesize sparse-view videos (10× faster than (Straub et al., 2019)). **2)** To test CARL, we create sequences with partial brightness reduction, where the first half of each video is clean and the latter half has lower brightness.

**Implementation.** **1)** CPL uses the RGB-based matcher from (Sun et al., 2021) to produce 2D correspondence, which is lift to 3D correspondence via depth. **2)** CARL uses a linear model for restoration, optimized using the Adam optimizer over 100 iterations with a learning rate of 0.2. **3)** CPL and CARL each add a total of 0.1 seconds (for 1200×680 input images), resulting in ***minimal overhead while enhancing robustness***. **4)** Hyperparameters follow the baseline (Keetha et al., 2024).

**Effect of CPL: Accelerated convergence and enhanced robustness under dynamic motion.** **1)** As shown in Table 5, our CorrGS and CorrGS-L consistently achieve the lowest ATE across all sequences, outperforming top pure Neural models like SplaTAM, SplaTAM-L, and Hero-SLAM (Xin et al., 2024), which target extreme motion scenarios, while also demonstrating greater robustness compared to GO-SLAM, which utilizes an additional loop closure mechanism. **2)** CPL leads to notable acceleration in pose learning convergence (see Appendix Fig. H37), significantly improving trajectory estimation under fast motion. **3)** Pose Quality Verification (PQV) step safeguards against potential failures in CPL, as seen in the performance drop without PQV (CorrGS-L w/o PQV) in Table 5. **4)** Fig. 12 shows that CPL enables robust tracking, resulting in high-quality 3D reconstruction, whereas the baseline fails. **5)** Our coupled data-model approach—using benchmark-identified weaknesses to guide model refinements—advances robustness. Specifically, motion perturbation benchmarks exposed pose tracking issues, leading to the development of CPL to address these limitations.

Table 6: Comparisons on noisy sparse-view video. ✗: tracking failure. CARL-T&M is default CorrGS-L model.

| Setting | O-0 | O-1 | O-2 | O-3 | O-4 | R-0 | R-1 | R-2 | Avg. |
|---|---|---|---|---|---|---|---|---|---|
| **Trajectory Estimation Error (ATE↓ [cm])** | | | | | | | | | |
| Baseline | ✗ | ✗ | ✗ | ✗ | ✗ | 0.51 | ✗ | ✗ | – |
| CARL-T | ✗ | ✗ | 3.44 | 0.68 | 0.95 | 0.43 | ✗ | ✗ | – |
| CARL-M | ✗ | ✗ | ✗ | ✗ | ✗ | 5.73 | ✗ | ✗ | – |
| **CARL-T&M (Default)** | **0.30** | **0.12** | **0.22** | **0.31** | **0.41** | **0.35** | **1.08** | **1.55** | **0.54** |
| **RGB Restoration Quality (PSNR↑ [dB])** | | | | | | | | | |
| Baseline | ✗ | ✗ | ✗ | ✗ | ✗ | 12.67 | ✗ | ✗ | – |
| CARL-T | ✗ | ✗ | 12.82 | 12.53 | 20.14 | 11.92 | ✗ | ✗ | – |
| CARL-M | ✗ | ✗ | ✗ | ✗ | ✗ | 24.53 | ✗ | ✗ | – |
| **CARL-T&M (Default)** | **40.67** | **40.01** | **33.61** | **32.33** | **36.09** | **32.52** | **33.23** | **34.54** | **35.38** |
| **Depth Rendering Quality (Depth L1↓ [cm])** | | | | | | | | | |
| Baseline | ✗ | ✗ | ✗ | ✗ | ✗ | 0.76 | ✗ | ✗ | – |
| CARL-T | ✗ | ✗ | 1.40 | 1.10 | 1.48 | 0.74 | ✗ | ✗ | – |
| CARL-M | ✗ | ✗ | ✗ | ✗ | ✗ | 4.66 | ✗ | ✗ | – |
| **CARL-T&M (Default)** | **0.25** | **0.19** | **0.55** | **0.81** | **0.52** | **0.58** | **0.52** | **0.63** | **0.63** |

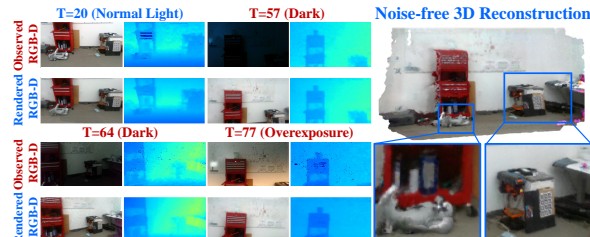

Figure 13: Effect of CARL on 3D reconstruction from sparse-view noisy video.

**Effect of CARL: Online restoration learning enables consistent noise-free 3D reconstruction.**
Table 6 quantitatively evaluates CARL's effect on ego-motion estimation, RGB restoration, and depth rendering quality under partial degraded illumination, with qualitative 3D reconstruction results in Fig. 13. **1)** CARL significantly improves pose accuracy and enables noise-free 3D reconstruction from noisy video, increasing tracking success from 1/8 (baseline) to 8/8, with an ATE of 0.54 cm, close to the noise-free value of 0.45 cm. **2)** Omitting the restored image in mapping (CARL-T) reduces tracking success from 8/8 to 4/8 and results in inconsistent maps with regional severe artifacts. **3)** Using CARL solely for mapping (CARL-M) improves RGB quality compared to the baseline (24.53 dB vs. 12.67 dB in PSNR on O-1), but loses track due to pose optimization relying on unrestored RGB-D. **4)** Thus, the full CARL method (CARL-T&M) is essential for robust pose tracking and noise-free 3D reconstruction, achieving a high average restored RGB PSNR of 35.48 dB.

### 5.3 PILOT STUDY ON REAL-WORLD NOISY SPARSE-VIEW VIDEO

To demonstrate *CorrGS*'s practical applicability, we conducted a pilot study using noisy, sparse-view video captured under real-world conditions, including extreme lighting variations (*e.g.*, darkness to overexposure), fast motion, and depth noise, recorded with a RealSense D435i sensor. Fig. 14 shows that *CorrGS* effectively handles sensing degradation from dynamic illumination and fast motion, rendering noise-free RGB and reconstructing a clean, photorealistic 3D map from noisy RGB-D. Noisy depth in some frames (see Appendix Fig. H44) suggests an opportunity to enhance the model with depth restoration alongside color restoration.

Figure 14: RGB-D rendering and 3D reconstruction by *CorrGS* on real, fast video with dynamic illumination.

## 6 CONCLUSION AND FUTURE WORK

**Conclusion.** We presented a structured approach to tackle robust ego-motion estimation and photorealistic 3D reconstruction from noisy video. Our contributions are threefold: **1)** We developed a scalable noisy data synthesis pipeline incorporating comprehensive RGB-D perturbations for mobile agents, enabling realistic and diverse noisy datasets. **2)** We introduced the *Robust-Ego3D* benchmark to systematically expose performance bottlenecks and guide targeted improvements. **3)** Building on insights from our benchmark analyses, we proposed *CorrGS*, which significantly enhances robustness in ego-motion estimation and 3D reconstruction from noisy, sparse-view video. This work sets a new standard for assessing and advancing the robustness of dense Neural SLAM.

**Future work**. We hope to inspire exploration to *embrace imperfect observations and turn them into clarity in 3D and ego-motion*, as the road ahead offers opportunities: **1)** refining the approach to adaptively select perturbations based on real-world conditions, ensuring efficiency and relevance; **2)** leveraging generative models to produce richer, more diverse perturbations that better reflect real-world challenges; **3)** expanding *Robust-Ego3D* to encompass outdoor and cross-domain environments, broadening the scope of model evaluation; and **4)** optimizing *CorrGS* by implementing adaptive processing for key frames, further enhancing its performance and efficiency.

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

TABLE OF CONTENTS OF THE APPENDIX

# A    MORE DETAILS ABOUT PERTURBATION TAXONOMY

Figure A1: **Taxonomy of perturbations for RGB-D sensing.** The *sources of perturbations* include: (**a**) sensor pose errors, (**b**) RGB and (**c**) depth imaging corruptions, and (**d**) RGB-D sensor synchronization errors. Dashed arrows illustrate the *propagation order* of individual perturbations.

As shown in Fig. A1, perturbations affecting RGB-D sensing systems for mobile agents can originate from sensor pose deviations, inaccuracies within the RGB-D imaging processes, and desynchronization issues between RGB and depth sensors. Within a RGB-D sensing system, the order (dashed arrows of Fig. A1) in which perturbations occur and interact follows the sensing and processing procedure.

### A.1   PERTURBATION ON SENSOR POSES

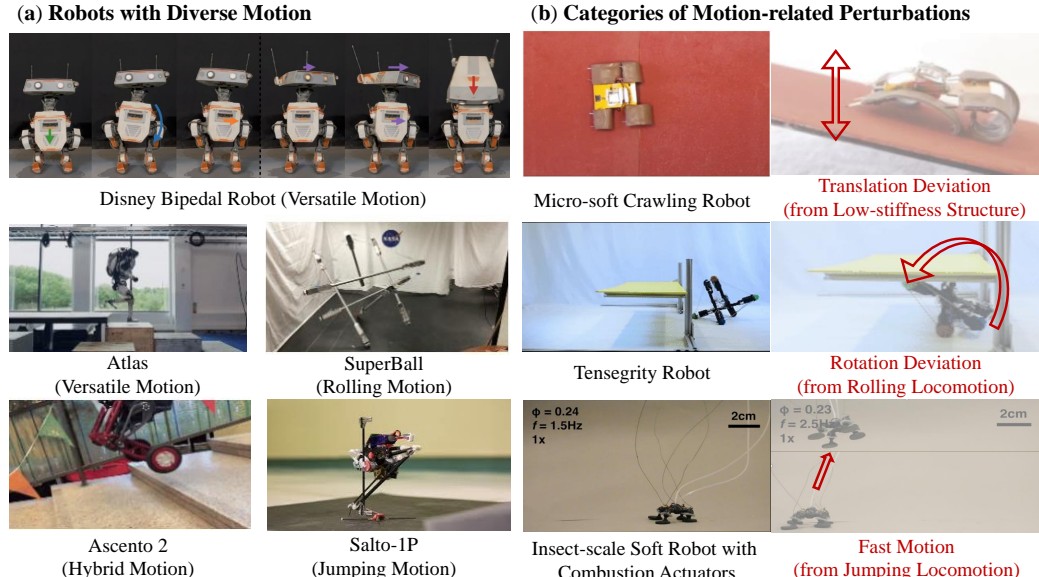

Figure A2: **Real-world mobile platforms can have versatile motion.**   (**a**) Autonomous robots exhibit highly dynamic and versatile motions, pushing the boundaries of traditional control systems. These diverse movement types challenge SLAM systems, which must operate in these dynamic contexts. (**b**) To address these challenges, we categorize motion perturbations into three key types: translation deviations (*e.g.*, caused by low-stiffness structures in soft robots (Patel et al., 2023a)), rotation deviations (*e.g.*, from rolling locomotion in tensegrity robots (Huang et al., 2022)), and fast-motion deviations (*e.g.*, arising from high-speed or jumping movements (Aubin et al., 2023)). These categories capture the full spectrum of dynamic disturbances that perception systems must handle for reliable operation.

**Motivation from real-world mobile agents.** Deploying sensing systems on autonomous robots requires addressing the diverse and dynamic motions these robots experience, as shown in Fig. A2a. Examples include the Disney bipedal robot (Grandia et al., 2024) and Atlas humanoid robot (Feng et al., 2014), both of which exhibit complex full-body control, as well as the SuperBall tensegrity robot (Sabelhaus et al., 2015), which rolls, and the hybrid jumping and wheel-based locomotion of Ascento (Klemm et al., 2019) and Salto-1P (Yim & Fearing, 2018), which excels in rapid jumping maneuvers. While these versatile motions allow for more advanced robot manipulation and locomotion, they present significant challenges when sensors are placed on the robots for dense SLAM. In real-world scenarios, autonomous agents frequently face external disturbances such as platform vibrations, uneven terrain, and dynamic movements, complicating sensor pose estimation and degrading SLAM performance. Existing benchmarks like Replica (Straub et al., 2019) and ShapeNet (Chang et al., 2015) typically assume smooth, stable sensor trajectories, which fail to capture the complexities and instabilities of real-world environments. For example, a visual SLAM system operating on a legged robot navigating rough terrain is prone to significant motion deviations and vibrations, negatively affecting both sensor pose estimation and the quality of visual data.

To better emulate real-world challenges, we introduce *rigid-body motion perturbations* that simulate disturbances by perturbing the sensor's trajectory in both rotational and translational axes. Grounded in rigid-body transformation theory, this method uses precise mathematical models to inject dynamic perturbations that mirror the unpredictable conditions encountered in realistic environments, providing a more rigorous test for SLAM systems.  As shown in Fig. A2b, we categorize these motion perturbations into three main types: translation deviations (*e.g.*, from low-stiffness structures in soft robots (Patel et al., 2023a)), rotation deviations (*e.g.*, from rolling locomotion in tensegrity robots (Huang et al., 2022)), and fast-motion deviations (*e.g.*, arising from high-speed movements or jumping (Aubin et al., 2023)). These categories capture the full spectrum of dynamic disturbances

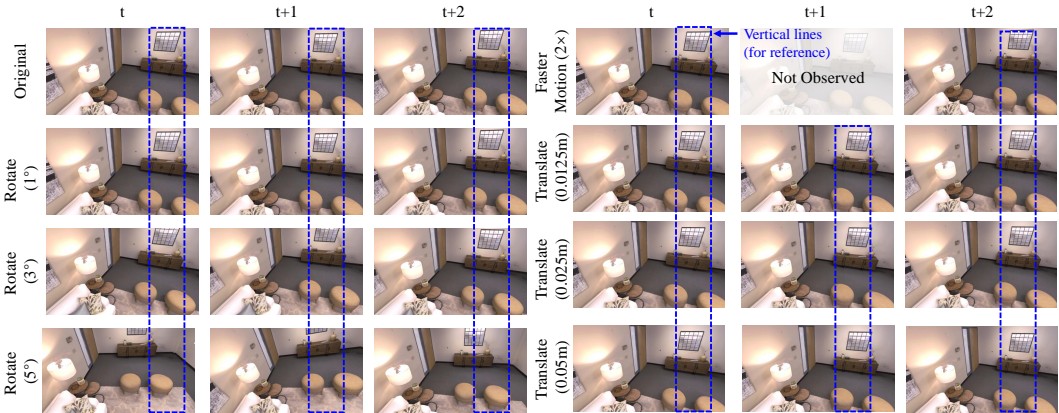

Figure A3: **Rendered RGB image streams under trajectory-level perturbations**, including translation deviations (Translate), rotation deviations (Rotate), and the faster motion effect.

that SLAM systems must handle, ensuring a more robust and comprehensive evaluation of their performance and reliability in real-world scenarios.

**(a) Motion deviations using rigid-body transformations.** A rigid-body transformation in 3D space consists of both a rotational and a translational component. For each sensor pose, represented by a rotation matrix $\mathbf{R} \in \boldsymbol{SO}(3)$ and a translation vector $\mathbf{t} \in \mathbb{R}^3$, we introduce perturbations that simulate physical disturbances in both orientation and position.

**Rotation deviations.** ($\Delta \mathbf{R} \in \boldsymbol{SO}(3)$): The rotation of the sensor is modeled by a 3x3 orthogonal matrix $\mathbf{R}$, which belongs to the special orthogonal group $SO(3)$. To simulate realistic rotational deviations, we introduce a perturbation $\Delta \mathbf{R}$, also a member of $SO(3)$, which represents a small rotational offset. The perturbed orientation $\mathbf{R}'$ is then given by:

$$\mathbf{R}' = \mathbf{R}\Delta \mathbf{R} \tag{A1}$$

where $\Delta \mathbf{R}$ is a rotation matrix corresponding to small angular deviations around the principal axes (x, y, z). These deviations are sampled from a zero-mean multivariate Gaussian distribution $\mathcal{N}(0, \Sigma_R)$, with covariance $\Sigma_R$ governing the severity of the perturbation. Specifically, $\Delta \mathbf{R}$ can be computed as:

$$\Delta \mathbf{R} = \exp\left([\omega]_\times\right) \tag{A2}$$

where $[\omega]_\times$ is the skew-symmetric matrix representation of the angular velocity vector $\omega = [\omega_x, \omega_y, \omega_z]^\top$, and $\omega$ is sampled from $\mathcal{N}(0, \Sigma_R)$. This formulation ensures the perturbed rotation matrix remains a valid element of $SO(3)$, preserving the properties of rigid-body rotations.

**Translation deviations.** ($\Delta \mathbf{t} \in \mathbb{R}^3$): The translational position of the sensor is represented by a vector $\mathbf{t}$, which defines the sensor's position in the world frame. To model translational perturbations, we add a random perturbation $\Delta \mathbf{t}$, sampled from a zero-mean Gaussian distribution $\mathcal{N}(0, \Sigma_t)$, resulting in the perturbed position:

$$\mathbf{t}' = \mathbf{t} + \Delta \mathbf{t} \tag{A3}$$

where $\Sigma_t$ is the covariance matrix that controls the magnitude and directionality of the perturbations. These translational deviations reflect small shifts caused by external physical factors such as terrain-induced vibrations or mechanical imperfections in the robot platform.

Together, the perturbed sensor pose $(\mathbf{R}', \mathbf{t}')$ models the real-world instability experienced by mobile platforms, where both rotational and translational movements are affected by external disturbances.

In Fig. A3, we present the rendered sensor streams under varying severity levels of trajectory-level perturbations, encompassing translational deviations, rotational deviations, and faster motion effects. Although the rotational and translational deviations we examined result in minor changes in observations between adjacent frames, these perturbations lead to significant performance degradation across the majority of benchmarking SLAM models. As depicted in Fig. A4, even slight trajectory-level deviations can have a substantial impact on trajectory estimation performance.

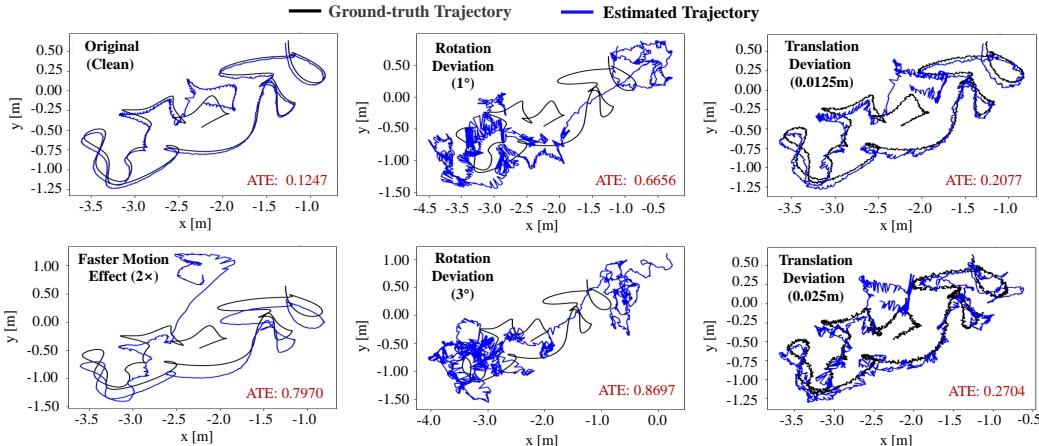

Figure A4: **Illustrations of motion deviations and the faster motion effect.** We present the synthesized ground-truth trajectories (in black) and the estimated trajectories (in blue) obtained using the CO-SLAM (Wang et al., 2023) model, which shows that slight trajectory deviations can have a significant impact on the trajectory estimation performance.

**(b) Faster motion effect.** In addition to introducing pose perturbations, we evaluate the robustness of sensing systems by simulating higher-speed motion scenarios. To do this, we down-sample the sensor's observation stream along the time axis, effectively increasing the perceived motion between consecutive frames. Let the original sensor stream have a time step $t$, and we introduce a faster motion effect by down-sampling at intervals $t' = kt$, where $k > 1$ is the down-sampling factor. This creates larger inter-frame motion, challenging SLAM algorithms to maintain accurate pose estimation despite fewer temporal observations.

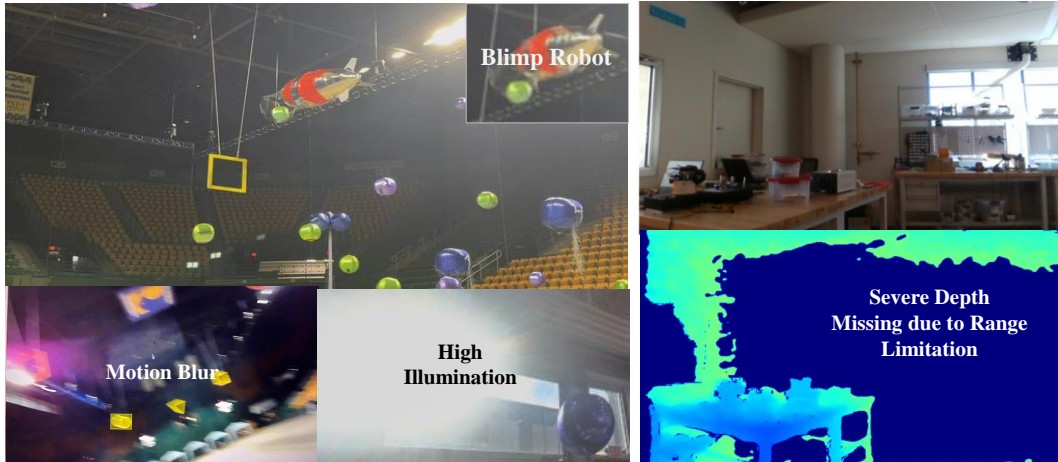

Figure A5: **Corruption in RGB-D imaging is a frequent issue in real-world data collection.** Challenges like motion blur and extreme illumination are common in the RGB sensor stream collected by mobile platforms (**Left**), while depth data loss due to range limitations significantly affects modeling (**Right**)

## A.2   PERTURBATION ON RGB SENSOR IMAGING

**Imaging corruptions is common in real-world data collected by mobile systems.** In real-world deployments, the perception model of autonomous systems frequently encounters imaging corruptions that degrade sensor data quality. These include motion blur, high illumination conditions, and depth data loss due to range limitations, as seen in the examples in Fig. A5. For instance, the Blimp robot experiences motion blur and high illumination during operation, while the depth collection via RealSense D435 RGB-D sensor could suffer from severe depth missing due to depth perception limitation, highlighting the challenges sensing systems face in maintaining accurate sensor pose estimation and environmental mapping under such conditions.

The perturbations introduced for RGB imaging are designed to model potential error sources across the entire image formation and processing pipeline, spanning from the physical 3D scene to the final 2D image. These perturbation sources encompass environmental interferences affecting light propagation, lens-related optical distortions, sensor noise due to imperfections in image acquisition hardware, and post-processing effects applied after image capture. This comprehensive approach simulates real-world image degradations in a physically grounded manner to test the robustness of perception systems under diverse visual conditions.

**(a) Environmental Interference**   Environmental interference, such as adverse weather conditions, affects light transmission through scattering, absorption, and occlusion. To simulate these effects, we employ *alpha blending techniques* that mix a layer representing the environmental interference with the clean image, generating a composite that simulates the perturbed conditions. This method aims at approximating real-world phenomena where environmental factors interfere with direct light paths.

- **Snow Effect**: Snow introduces random, localized variations in light intensity due to snowflakes scattering light. To model this, we generate a random noise layer with white spots simulating snow and blend it with the original image (Von Bernuth et al., 2019).

- **Frost Effect**: Frost simulates a translucent overlay that partially occludes the scene. This effect is modeled by applying a semi-transparent whitening filter to the image using a weighted combination of the clean image and a whitened version of it (Michaelis et al., 2019).

- **Fog Effect**: Fog scatters light uniformly, reducing contrast and blurring distant objects. We simulate fog by linearly interpolating between the original image and a constant gray-value image to simulate the effect of reduced visibility (Hendrycks & Dietterich, 2019).

- **Spatter Effect**: Spatter simulates droplets or streaks on the camera lens, obstructing part of the image. This effect is modeled using a semi-transparent mask applied to the image, with regions randomly designated as perturbed (spattered) or transparent. The spatter mask controls the distribution and severity of the distortion.

**(b) Lens-Related Distortions (Blur)**   Lens imperfections or motion during image capture can introduce blurring effects, which we simulate by convolving the input image with specific blur kernels. These kernels approximate various physical processes responsible for image blur, such as defocusing or rapid motion during capture.

- **Defocus Blur**: Caused by the camera lens being out of focus, this effect is simulated by convolving the image with a circular disc kernel (bokeh) to model the blurring of regions outside the camera's depth of field.

- **Glass Blur**: This effect emulates viewing an image through textured or distorted glass. We model this by convolving the image with an irregular blur kernel, simulating the scattering of light by the glass surface.

- **Motion Blur**: When the camera or objects in the scene move rapidly during image capture, it results in motion blur. We simulate this effect by convolving the image with a linear kernel oriented in the direction of motion to model the smearing caused by the movement.

- **Gaussian Blur**: To approximate image smoothing caused by lens imperfections, we apply Gaussian blur, which convolves the image with a Gaussian kernel. The degree of blurring is controlled by the standard deviation of the Gaussian function, simulating varying levels of softening.

**(c) Sensor Noise**   RGB image sensors are inherently imperfect, introducing noise during image acquisition that affects image quality. We model several types of sensor noise, each representing a different physical process that affects image formation.

- **Gaussian Noise**: Simulated by adding random pixel-wise noise sampled from a Gaussian distribution with zero mean, this models sensor imperfections where pixel intensities are corrupted by random fluctuations.

- **Shot Noise**: Shot noise arises from the quantum nature of light, caused by the random arrival of photons at the sensor. It is modeled using a Poisson distribution (Hasinoff, 2014) to account for the discrete nature of photon interactions during image capture.

- **Impulse Noise**: Also known as salt-and-pepper noise, this effect randomly corrupts individual pixels. For each pixel, a random value is sampled, and if it falls within a specified range, the pixel is either set to the minimum or maximum intensity value, while other pixels remain unchanged.

- **Speckle Noise**: Speckle noise is modeled as multiplicative noise, commonly seen in coherent imaging systems like radar and ultrasound. The intensity of each pixel in the image is perturbed as follows:

$$I'(x, y) = I(x, y) \times (1 + \rho \times \eta) \tag{A4}$$

  where $\rho$ controls the noise intensity, and $\eta$ is a Gaussian noise term.

**(d) Post-Processing Perturbations**   Post-processing effects modify the image after it has been captured, often degrading quality through compression or global adjustments.

- **Brightness**: Adjusting the brightness of the image involves adding a constant value to every pixel. This models exposure adjustments or lighting variations that affect the overall intensity of the image.

- **Contrast**: Contrast changes affect the variance in pixel intensities. We model contrast adjustments by linearly scaling pixel intensities about the mean intensity $\mathcal{J}$:

$$I'(x, y) = \beta \times (I(x, y) - \mathcal{J}) + \mathcal{J} \tag{A5}$$

  where $\beta$ controls the level of contrast.

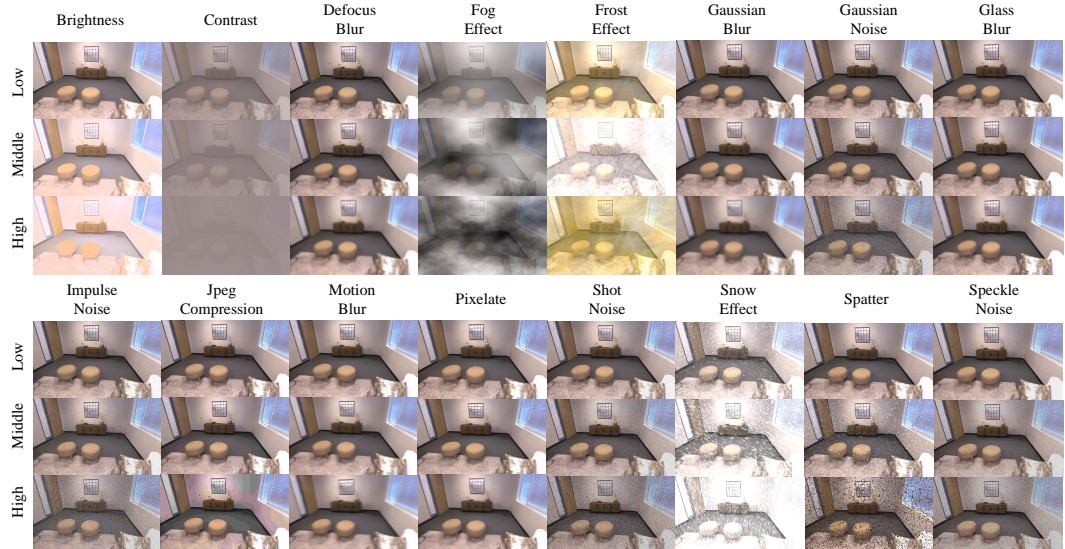

Figure A6: **Illustration of RGB imaging perturbations under different severity levels**. We consider **16** common image corruption types (Hendrycks & Dietterich, 2019) from **4** main categories of perturbations for robustness evaluation: **(1) noise-based distortions**: *Gaussian Noise*, *Shot Noise*, *Impulse Noise*, and *Speckle Noise*; **(2) blur-based effects**: *Defocus Blur*, *Glass Blur*, *Motion Blur*, and *Gaussian Blur*; **(3) environmental interferences**: *Snow Effect*, *Frost Effect*, *Fog Effect*, and *Spatter Effect*. **(4) post-processing manipulations**: *Brightness*, *Contrast*, *Pixelate*, and *JPEG Compression*. Each perturbation type is further split into **3** severity levels (low, middle, and high) for our benchmarking evaluation, which corresponds to Level 1, Level 3, and Level 5 of Table A1.

- **JPEG Compression**: JPEG compression introduces artifacts during lossy image compression. These artifacts, such as blocking effects and loss of detail, are simulated by reducing the image quality and reintroducing compression-induced distortions.

- **Pixelation**: This effect reduces image resolution by dividing the image into pixel blocks and replacing each block with its average value, simulating a drop in resolution commonly seen in low-quality video feeds.

**Implementation of RGB imaging perturbations.** We illustrate RGB imaging perturbations under different severity levels in Fig. A6. As shown in Table A1, we define five severity levels for each type of RGB imaging perturbation, following established robustness evaluation literature (Hendrycks & Dietterich, 2019).

Table A1: Specific configurations of the RGB imaging perturbations.

| Perturbation | Parameter | Level 1 | Level 2 | Level 3 | Level 4 | Level 5 |
|---|---|---|---|---|---|---|
| Snow Effect | (Mean, std, scale, threshold, blur radius, blur std, blending ratio) | 0.1, 0.3, 3.0, 0.5, 10.0, 4.0, 0.8 | 0.2, 0.3, 2, 0.5, 12, 4, 0.7 | 0.55, 0.3, 4, 0.9, 12, 8, 0.7 | 0.55, 0.3, 4.5, 0.85, 12, 8, 0.65 | 0.55, 0.3, 2.5, 0.85, 12, 12, 0.55 |
| Frost Effect | (Frost intensity, texture influence) | (1.00, 0.40) | (0.80, 0.60) | (0.70, 0.70) | (0.65, 0.70) | (0.60, 0.75) |
| Fog Effect | (Thickness, smoothness) | (1.5, 2.0) | (2.0, 2.0) | (2.5, 1.7) | (2.5, 1.5) | (3.0, 1.4) |
| Spatter Effect | (mean, standard deviation, sigma, threshold, scaling factor, complexity of effect) | (0.65, 0.3, 4, 0.69, 0.6, 0) | (0.65, 0.3, 3, 0.68, 0.6, 0) | (0.65, 0.3, 2, 0.68, 0.5, 0) | (0.65, 0.3, 1, 0.65, 1.5, 1) | (0.67, 0.4, 1, 0.65, 1.5, 1) |
| Defocus Blur | (Kernel radius, alias blur) | (3.0, 0.1) | (4.0, 0.5) | (6.0, 0.5) | (8.0, 0.5) | (10.0, 0.5) |
| Glass Blur | (Sigma, max delta, iterations) | (0.7, 1.0, 2.0) | (0.9, 2.0, 1.0) | (1.0, 2.0, 3.0) | (1.1, 3.0, 2.0) | (1.5, 4.0, 2.0) |
| Motion Blur | (Radius, sigma) | (10, 3) | (15, 5) | (15, 8) | (15, 12) | (20, 15) |
| Gaussian Blur | Sigma | 1 | 2 | 3 | 4 | 6 |
| Gaussian Noise | Noise scale | 0.08 | 0.12 | 0.18 | 0.26 | 0.38 |
| Shot Noise | Photon number | 60 | 25 | 12 | 5 | 3 |
| Impulse Noise | Noise amount | 0.03 | 0.06 | 0.09 | 0.17 | 0.27 |
| Speckle Noise | Noise scale | 0.15 | 0.2 | 0.35 | 0.45 | 0.6 |
| Brightness Increase | Adjustment ratio | 0.1 | 0.2 | 0.3 | 0.4 | 0.5 |
| Contrast Decrease | Adjustment of pixel mean | 0.40 | 0.30 | 0.20 | 0.10 | 0.05 |
| JPEG Compression | Compression quality | 25 | 18 | 15 | 10 | 7 |
| Pixelate | Resize factor | 0.60 | 0.50 | 0.40 | 0.30 | 0.25 |

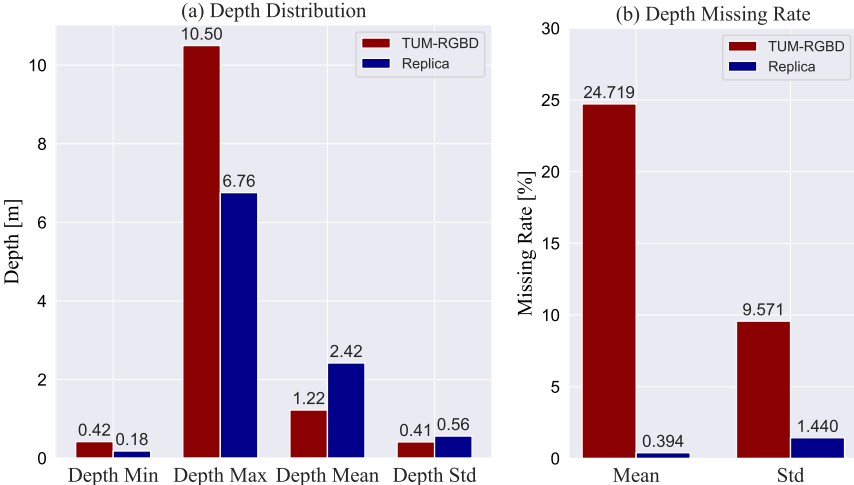

Figure A7: **Discrepancy in depth characteristics between simulated Replica (Straub et al., 2019) and real TUM-RGBD (Sturm et al., 2012) datasets.**.

### A.3 PERTURBATION ON DEPTH SENSOR IMAGING

**Motivation.** As illustrated in Fig. A7, there exists a noticeable disparity between simulated clean depth data, such as those from the Replica (Straub et al., 2019) SLAM benchmark, and real-world noisy depth data obtained from the TUM-RGBD (Schubert et al., 2018) SLAM benchmark. For example, while the Replica dataset has a minimum depth of approximately 0.18 m, the TUM-RGBD data exhibits a minimum depth of 0.4 m, reflecting the range limitations typical of real-world depth sensors. Additionally, real-world depth sensors tend to have a significantly higher rate of missing data, with TUM-RGBD showing approximately 25% missing depth values compared to less than 0.4% in Replica. These discrepancies highlight the need to introduce perturbations that model the limitations of real depth sensors, bridging the gap between simulated and real-world data.

In this section, we propose four key depth perturbations that emulate the sensing limitations and environmental constraints faced by real-world depth sensors, thereby offering a more realistic evaluation of SLAM systems.

**(a) Gaussian Noise**   Real-world depth sensors, such as time-of-flight (ToF) and structured light systems, are subject to random measurement noise due to factors like photon counting errors and sensor precision limits. This noise is often modeled as Gaussian-distributed errors across the depth map. To simulate this, we perturb each pixel $(x, y)$ in the depth map $D$ by adding Gaussian noise $\eta \sim \mathcal{N}(0, \sigma^2)$, where $\sigma$ is the standard deviation of the noise distribution, dependent on the sensor's characteristics:

$$D'(x, y) = D(x, y) + \eta \tag{A6}$$

This models the stochastic fluctuations in depth measurements, capturing the inherent noise that arises during the depth sensing process.

**(b) Edge Erosion**   Depth sensors, particularly ToF systems, often suffer from multi-path interference and sensor limitations near depth discontinuities, such as object edges. This results in inaccurate depth estimates along these regions. We simulate this by performing edge detection to identify the boundary pixels in the depth map and subsequently eroding a fraction of these edge pixels, representing measurement failures near complex geometries:

$$D'(x, y) = \begin{cases} \text{VOID} & \text{if } (x, y) \in \mathcal{P} \\ D(x, y) & \text{otherwise} \end{cases} \tag{A7}$$

where $\mathcal{P}$ represents the set of detected edge pixels, and VOID signifies missing data. This emulates the failure of depth sensors to capture accurate data near edges due to signal interference.

Table A2: Specific configurations of the depth imaging perturbations.

| Perturbation | Parameter | Level 1 | Level 2 | Level 3 | Level 4 | Level 5 |
|---|---|---|---|---|---|---|
| Gaussian Noise | Noise scale | 0.1 | 0.2 | 0.3 | 0.4 | 0.5 |
| Edge Erosion | Erosion rate | 0.015 | 0.020 | 0.025 | 0.03 | 0.035 |
| Random missing depth data | Missing rate (%) | 10 | 15 | 20 | 25 | 30 |
| Range clipping | (Min depth, Max depth) | (0.2, 4.4) | (0.3, 4.2) | (0.4, 4.0) | (0.5, 3.8) | (0.6, 3.6) |

**(c) Random Missing Depth Data**   Real-world depth sensors frequently encounter occlusions, reflective surfaces, or environmental conditions that block the sensor's line of sight, leading to missing or incomplete depth measurements. We simulate this phenomenon by applying a binary mask $M$ to the depth map, randomly occluding portions of the image to reflect areas where depth data is not captured:

$$D'(x,y) = D(x,y) \cdot M(x,y) \tag{A8}$$

where $M(x,y)$ is a binary mask generated by randomly sampling rectangular patches. Pixels within the occluded regions are set to a VOID value, representing missing data due to sensor limitations.

**(d) Range Clipping**   Depth sensors, especially consumer-grade systems like Kinect or RealSense, have limited depth ranges, with measurements becoming unreliable beyond a specific maximum distance or too noisy below a minimum distance. We simulate this range limitation by clipping depth values outside a predefined range $[D_{min}, D_{max}]$. Depth values that fall outside this range are replaced with a VOID value:

$$D'(x,y) = \begin{cases} \text{VOID} & \text{if } D(x,y) < D_{min} \text{ or } D(x,y) > D_{max} \\ D(x,y) & \text{otherwise} \end{cases} \tag{A9}$$

This perturbation reflects the operational range limitations of real-world depth sensors, where objects too far or too close to the sensor result in unreliable or missing depth data.

**Implementation of Depth Imaging Perturbations**   As shown in Table A2, to better reflect real-world depth sensing conditions, we apply varying severity levels for each of the proposed perturbations, inspired by the real depth distribution of datasets such as TUM-RGBD (Schubert et al., 2018). The specific implementation of these depth perturbations is available through our Depth Imaging Perturbation Synthesis Toolbox. The toolbox defines five severity levels for each perturbation, ensuring a comprehensive evaluation of SLAM systems across different conditions.

A.4    PERTURBATION ON MULTI-SENSOR SYNCHRONIZATION

**Motivation.** While integrated RGB-D sensors like RealSense, Kinect, and ZED minimize desynchronization issues due to their tightly coupled design, many real-world applications require more flexibility in sensor selection. In this work, we address a more generalized scenario where RGB and depth sensors operate independently, rather than assuming they are synchronized. This approach provides greater adaptability for sensor configurations in autonomous systems. For instance, lightweight ToF-based depth sensors, such as the Arducam ToF, paired with compact RGB cameras, offer advantages in weight and energy efficiency—critical factors for SLAM systems deployed on platforms with strict payload and power constraints, such as micro UAVs. However, using distributed RGB and depth sensors introduces potential inconsistencies between data streams. De-synchronization, caused by differences in signal frequencies, can lead to temporal misalignment, resulting in inaccuracies in the combined RGB-D data. Addressing these challenges is crucial for enabling robust performance in resource-constrained environments.

To emulate sensor delays in cases where multiple sensors within an RGB-D sensing system are not synchronized, we introduce temporal misalignment between sensor streams (see Fig. 2d of the main paper). Consider two initially synchronized sensor streams, denoted as $\mathbf{S}_1(t)$ and $\mathbf{S}_2(t)$. We simulate a delay in the second stream by shifting its sensor sequence by a frame interval $\Delta$. This creates perturbed streams $\mathbf{S}'_1(t) = \mathbf{S}_1(t)$ and $\mathbf{S}'_2(t) = \mathbf{S}_2(t + \Delta)$. While one sensor stream is shifted, the poses associated with each sensor reading remain unchanged. This ensures the system is operating on data grounded in the past, reflecting the real-world scenario of misaligned sensor information.

# B  MORE DETAILS ABOUT *Robust-Ego3D* BENCHMARK

## B.1  ASSUMPTIONS FOR BENCHMARKING SETUP

While our customizable noisy data synthesis pipeline allows for noisy data generation from any clean 3D scene, incorporating any composition of perturbation types, with any number and configuration of RGB and depth sensors, we initialize the *Robust-Ego3D* benchmark for model robustness evaluation under the following assumptions:

**Task.** We focus on the standard (passive) SLAM setting, assuming the absence of active decision-making processes.

**Model.** Our analysis is centered on vision-oriented SLAM scenarios, specifically targeting monocular and RGB-D settings. We assume the use of dense depth representation as opposed to sparse depth data obtained from a LiDAR scanner. In addition, the SLAM system is presumed to have known motion and observation models.

**Perturbation.** Although our noisy data synthesis pipeline is capable of generating SLAM benchmarks with multiple heterogeneous perturbations, we concentrate on investigating the performance degradation caused by individual sensor or trajectory perturbations. This focused approach is designed to dissect the system's response to isolated perturbations, allowing precise quantification of their specific impacts on SLAM performance. By analyzing the degradation induced by individual perturbations, we can effectively assess the system's robustness in a controlled manner and identify the root causes of performance degradation. This knowledge is crucial for developing targeted mitigation strategies that address the most vulnerable aspects, *i.e.*, *Achilles' Heel*, of the whole SLAM system. Also, we model these perturbations using simplified linear models (*e.g.*, Gaussian noise assumptions), in line with precedent set by established literature (Hendrycks & Dietterich, 2019; Wang et al., 2020; Chattopadhyay et al., 2021). While these simplified perturbations may not fully capture the complexity of real-world scenarios, they offer interpretability and facilitate analysis across different perturbation types.

**3D scene**. We assume that the environment is static, meaning there are no moving or dynamically changing objects within the scene. Also, the scene is bounded, typically referring to an indoor setting with predefined boundaries or limits.

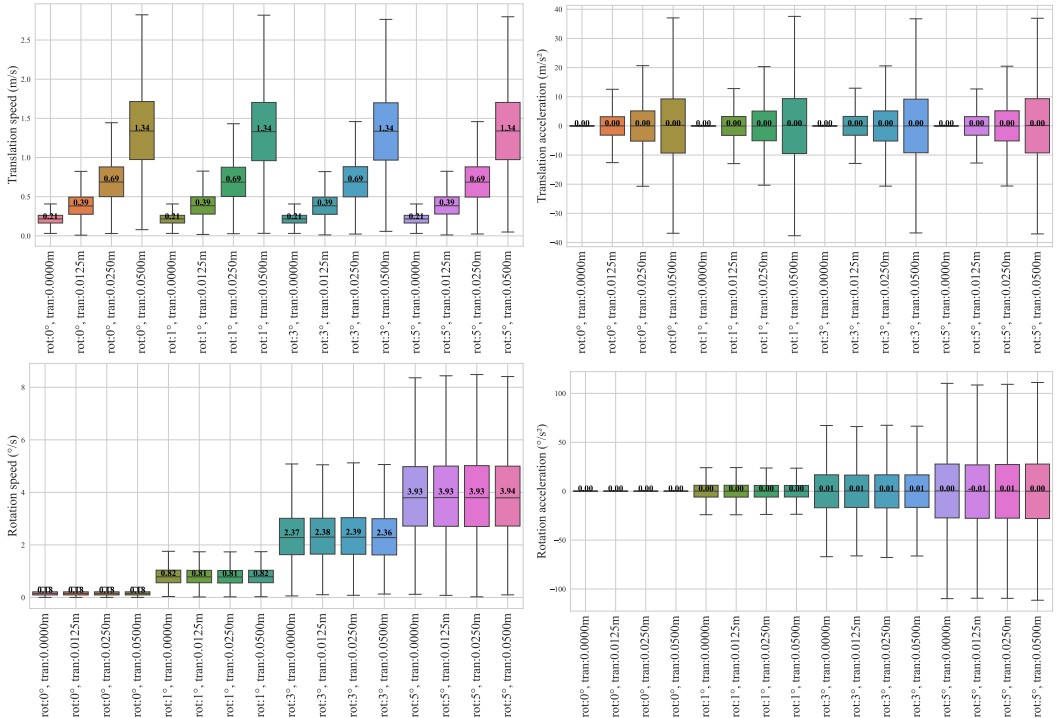

Figure B8: **Motion statistics of trajectory distribution under varying combinations of translation and rotation deviations**. Assuming a frame rate of 20 frames per second for the SLAM system, *i.e.*, a time interval of 0.05 seconds between neighboring pose frames, we present the motion distribution of perturbed trajectories in the proposed *Robust-Ego3D* benchmark. The figures show the distribution of translation speed (**Top Left**), translation acceleration (**Top Right**), rotation speed (**Bottom Left**), and rotation acceleration (**Bottom Right**). We report the mean value of each setting.

### B.2 *Robust-Ego3D* BENCHMARK SETUP DETAILS AND STATISTICS

**Data source for rendering**. We render the RGB-D sensor streams using 3D scene models sourced from the *Replica* dataset (Straub et al., 2019), which comprises real 3D scans of indoor scenes. We select the same set of eight rooms and offices as the (clean) Replica-SLAM dataset (Sucar et al., 2021) for consistent comparison. Each sequence has 2,000 frames at 1200×680 resolution.

**Benchmark sequence number distribution.** Using our established taxonomy of perturbations for SLAM and the noisy data synthesis pipeline, we have created a large-scale SLAM robustness benchmark called *Robust-Ego3D* to evaluate the robustness of monocular and RGB-D SLAM methods by incorporating various perturbations that mimic real-world sensor and motion effects.

Each perturbed setting of our benchmark is rendered in eight scenes from the 3D indoor scan dataset Replica (Straub et al., 2019). This process generates eight sequences from a single trajectory. For each perturbed setting, we calculate the average result from 24 experimental data points (eight sequences, each repeated three times). Then, we report the averaged result for each perturbed setting. Specifically, for the RGB imaging perturbation, we present the averaged result across three severity levels, under both static and dynamic perturbation modes.

We provide details about the specific distribution and setup of each perturbed setting as follows:

- **8 original clean sequences:** These sequences replicate the quality and the sequence number of the original Replica SLAM dataset (Sucar et al., 2021).

- **768 sequences with RGB imaging perturbations:** We apply 16 different types of image-level perturbations at 3 severity levels (Level 1, Level3, and Level 5 of Table A1), both under static and dynamic conditions.

- **32 sequences with depth imaging perturbations:** This category consists of 4 types of perturbations. For the depth noise, we adopt the hyperparameters of the Gaussian noise distribution as specified in previous literature (Hendrycks & Dietterich, 2019). Moreover, we set the depth missing rate to 20% and establish the depth clipping range based on the real-world depth distribution of the TUM-RGBD dataset (Sturm et al., 2012). The severity strength of each depth perturbation is shown in the Level 3 column of Table A2.

- **24 sequences with faster motion effects:** These sequences involve faster speed than the original sequences, with variations of two, four, and eight times the original speed.

- **120 sequences with motion deviations:** This category includes pure rotation deviation, pure translation deviation, and combined transformation matrix deviation. We define three severity levels for both rotation and translation deviations, and sample the deviation from a Gaussian distribution. Specifically, for rotation deviation, we introduce random deviations in rotation around the x, y, and z axes, with mean values of zero and standard deviations of 1, 3, and 5 degrees at each pose frame. For translation deviation, we introduce random deviations in the x, y, and z axes, with mean values of zero and standard deviations of 0.0125, 0.025, and 0.05 meters at each pose frame. In Fig. B8, we show the motion statistics of the perturbed trajectory sequences under varying combinations of translation and rotation deviations. This category of trajectory-deviated sequences encompasses a broad spectrum of motion speeds and accelerations, enabling a progressive evaluation of the robustness of SLAM models against increasingly challenging motion types. These insights are especially valuable for evaluating the implementation of SLAM systems in high-speed scenarios or on agile robot platforms exposed to significant vibrations.

- **48 sequences with RGB-D sensor de-synchronization:** We consider both static and dynamic perturbation models for multi-sensor misalignment. In the static mode, a constant time delay is synthesized between the two sensor streams, while in the dynamic perturbation model, there is a varying time delay between the streams. Specifically, the multi-sensor misalignment perturbation sequences consist of 24 sequences with a fixed cross-sensor frame delay interval ($\Delta$) of 5, 10, and 20 frames, as well as 24 sequences with dynamic perturbation where $\Delta$ deviates by 1 frame from the fixed intervals of 5, 10, and 20 frames.

Overall, this benchmark dataset enables a comprehensive evaluation of existing SLAM algorithms under simulated perturbations, providing a thorough assessment of the robustness of multi-modal SLAM systems in a wide range of challenges.

Table B3: RGB-D SLAM methods for evaluation.

| Method | Type | Modality Mono/RGB-D | Map Representation | Loop Closure | External Data | Speed | Processing | Year |
|--------|------|---------------------|--------------------|--------------|---------------|-------|------------|------|
| ORB-SLAM3 | Classical, Sparse | ✓ / ✓ | Keyframe+ORB, Explicit | ✓ | ✗ | Real-time | CPU | 2020 |
| iMAP | Neural, Dense | ✗ / ✓ | NeRF-based[1], Implicit | ✗ | ✗ | Quasi Real-time | CPU+GPU | 2021 |
| Nice-SLAM | Neural, Dense | ✗ / ✓ | NeRF-based, Implicit | ✗ | ✗ | Quasi Real-time | CPU+GPU | 2022 |
| CO-SLAM | Neural, Dense | ✗ / ✓ | NeRF-based, Implicit | ✗ | ✗ | Real-time | CPU+GPU | 2023 |
| GO-SLAM | Neural, Dense | ✓ / ✓ | NeRF-based, Implicit | ✓ | ✓[2] | Quasi Real-time | CPU+GPU | 2023 |
| SplaTAM-S | Neural, Dense | ✗ / ✓ | Gaussian Splat, Explicit | ✗ | ✗ | Quasi Real-time | CPU+GPU | 2024 |

(1) 'NeRF-based' indicates methods that leverage implicit neural networks to encode the 3D scene, following the philosophy of NeRF (Rosinol et al., 2022).
(2) GO-SLAM initializes the model parameters from the DROID-SLAM model (Teed & Deng, 2021) which leverages external data for model pre-training.

## B.3 MORE DETAILS ABOUT BASELINE MODELS FOR BENCHMARKING

**Additional descriptions about benchmarking models.** While previous SLAM robustness evaluations primarily focused on classical methods (Wang et al., 2020; Bujanca et al., 2021), our benchmark encompasses both classical and learning-based SLAM systems. As shown in Table B3, in addition to the classical SLAM model ORB-SLAM3 (Campos et al., 2021), whose results are used for reference, we *mainly evaluate top-performing dense Neural SLAM models* on standard benchmarks with diverse architecture design. iMAP(Sucar et al., 2021) and Nice-SLAM(Zhu et al., 2022) are representative methods with implicit neural representations. GO-SLAM(Zhang et al., 2023) focuses on global pose optimization, while CO-SLAM(Wang et al., 2023) utilizes hybrid neural encoding. SplaTAM-S (Keetha et al., 2024) employs Gaussian-based (Kerbl et al., 2023) map representation. The hyperparameters are set based on the recommendations given in the original papers or use default settings otherwise. Below, we offer additional descriptions of these models.

- **ORB-SLAM3** (Campos et al., 2021): An extension of ORB-SLAM2 (Mur-Artal & Tardós, 2017) that incorporates a multi-map system and visual-inertial odometry, enhancing robustness and performance.

- **iMAP** (Sucar et al., 2021): A neural RGB-D SLAM system that utilizes the MLP representation to achieve joint tracking and mapping.

- **Nice-SLAM** (Zhu et al., 2022): A neural RGB-D SLAM model that employs a multi-level feature grid for scene representation, reducing computational overhead and improving scalability.

- **CO-SLAM** (Wang et al., 2023): An advanced neural RGB-D SLAM system with a hybrid representation, enabling robust camera tracking and high-fidelity surface reconstruction in real time.

- **GO-SLAM** (Zhang et al., 2023): A neural visual SLAM framework for real-time optimization of poses and 3D reconstruction. It supports both monocular and RGB-D input settings.

- **SplaTAM** (Keetha et al., 2024): A neural RGB-D SLAM model that follows Gaussian Splatting (Kerbl et al., 2023) to construct an adaptive map representation based on Gaussian kernels. Due to time and computational constraints, we evaluate the relatively more efficient SplaTAM-S model variant in our benchmark.

**Remark: Some dense Neural SLAM models are inherently 'trained' on the testing distribution.** Neural SLAM models are optimized (*i.e.*, 'trained') on perturbed RGB-D observations, enabling continuous adaptation and updating of internal representations based on incoming data at each timestamp. This inherently includes 'training with introduced perturbations', as the models adjust to variations during online operation. Neural SLAM methods like Nice-SLAM (Zhu et al., 2022), CO-SLAM (Wang et al., 2023), and GO-SLAM (Zhang et al., 2023) leverage Neural Radiance Field (NeRF) as the map representation, updating the parameters of NeRF network and the parameters of poses for each frame when new observations arrive during testing; SplaTAM (Keetha et al., 2024) leverages explicit Gaussian Splats (Kerbl et al., 2023) as the map representation, updating the parameters of Gaussian kernels as well as the parameters of poses for each frame during testing. We find that this test-time online learning mechanism provides better robustness compared to non-Neural SLAM methods without adaptation capabilities, allowing Neural SLAM models to be robust to static RGBD imaging perturbations by continuously refining their environment understanding for optimal performance.

### B.4 DETAILS ABOUT HARDWARE SETUP FOR BENCHMARKING EXPERIMENTS

Our experiments were primarily conducted on a GPU server equipped with two NVIDIA A6000 GPUs, each featuring 48 GB of memory. These resources were utilized for synthesizing perturbed noisy data and evaluating the robustness of RGB-D SLAM models. The operating system used was Ubuntu 22.04. Additionally, we tested the compatibility of our benchmarking code on a GPU server with four NVIDIA RTX6000 Ada GPUs, each with 48 GB of memory, and on a GPU server with two NVIDIA A100 GPUS, each with 40 GB of memory.

It is important to note that the memory requirements of different SLAM methods vary based on the complexity of the perturbed RGB-D video sequences used for evaluation and the specific memory cost of each method. For instance, the CO-SLAM (Wang et al., 2023) model can run on a GPU with 12GB of memory. Meanwhile, only a GPU is required for all the SLAM methods evaluated in our study under each perturbed setting.

### B.5 UNIQUENESS AND ADVANTAGES OF OUR *Robust-Ego3D* BENCHMARK

Our instantiated benchmark *Robust-Ego3D* for RGB-D SLAM robustness evaluation offer several distinct advantages that can advance scalable SLAM evaluation and more robust SLAM model:

**Unparalleled diversity and controllability.** With 124 perturbation settings and an extensive dataset comprising 1,000 long video sequences and nearly 2 million image-depth pairs, our tool offers unmatched diversity and controllability. Researchers can create highly customized and challenging test scenarios, exploring a wide range of real-world conditions and pushing the boundaries of SLAM algorithms. This extensive collection of perturbations allows for a comprehensive assessment of SLAM systems under diverse environmental conditions and sensor noise profiles.

**Scalability and fair comparison.** The large size of our dataset enables statistically significant evaluations and fair comparisons between different SLAM algorithms under diverse conditions. This scalability is crucial for robust benchmarking and identifying the strengths and weaknesses of various approaches. By providing a large and diverse testing ground, our tool facilitates unbiased comparisons and promotes the development of more reliable and generalizable SLAM solutions.

**Decoupled perturbation study.** Our pipeline facilitates the decoupled study of individual and mixed perturbations, providing valuable insights into the isolated and combined effects of various noise sources. This granular understanding is essential for developing targeted strategies to enhance SLAM robustness in complex environments. By disentangling the impact of individual noise sources, researchers can gain a deeper understanding of their specific effects on SLAM performance and design algorithms resilient to specific types of perturbations.

**Standardization.** Our pipeline introduces a systematic and standardized approach to generating noisy environments, ensuring consistency and reproducibility across different studies. This standardization is crucial for facilitating meaningful comparisons and advancing the field of SLAM research. By establishing a common framework for generating and evaluating SLAM datasets with perturbations, our tool promotes collaboration and accelerates the progress of the entire research community.

## B.6 Uniqueness of Synthetic Datasets Compared to Real-World Data

Synthesized datasets offer unique and substantial advantages over real-world datasets, particularly when it comes to SLAM evaluation under noisy or complex conditions. While real-world datasets like TUM-RGBD (Sturm et al., 2012) capture certain real-world complexities, such as environmental noise and dynamic motion, the flexibility, scalability, and precision provided by synthesized datasets make them invaluable for comprehensive dense Neural SLAM evaluation and beyond.

**Accurate and complete 3D ground-truth for photorealistic mapping evaluation.** Real-world datasets, though offering realism, often lack precise ground-truth 3D scans needed for detailed, quantitative mapping evaluation. Specifically, the absence of high-fidelity 3D geometry and appearance data limits the ability to assess mapping quality in depth. For example, datasets such as SubT (Ebadi et al., 2023) and TUM-RGBD (Sturm et al., 2012) provide odometry data but fail to deliver accurate 3D scans required for photorealistic mapping evaluations. In contrast, synthesized datasets, such as those rendered from high-quality 3D assets like Replica (Straub et al., 2019) and ScanNet (Dai et al., 2017a), provide nearly perfect 3D ground-truth data for both odometry and geometry, making them indispensable for evaluating mapping performance.

**Scalability and diversity for evaluating (and training) generalizable models.** Real-world SLAM datasets are limited in size and diversity due to the challenges of data collection, such as manual setup and difficulty capturing complex scenarios like sensor noise, motion blur, or dynamic motion. As a result, these datasets are often insufficient for training robust, generalizable SLAM models. In contrast, synthesized datasets offer a ***scalable solution*** by generating large, diverse data under controlled conditions. This allows exploration of rare and costly scenarios, such as dynamic lighting changes and sensor degradation. Synthesized data not only serves as a comprehensive evaluation platform but also helps SLAM models generalize across real-world environments. The ability to generate unlimited data under varied conditions is key to developing SLAM systems that perform reliably in dynamic settings. ***We argue that scalable synthesized datasets are essential for training models that generalize to real-world applications.***

**Customization and control over perturbations.** One of the most significant advantages of synthesized datasets is their ability to inject specific, customizable perturbations in a controlled manner. Real-world datasets are inherently limited by the conditions present during data collection, which restricts the types of perturbations they can capture. Factors such as sensor noise, environmental variation, and lighting conditions are often incidental, making it difficult to systematically test how different perturbations affect SLAM performance. Synthesized datasets, however, offer ***greater flexibility in customizing and injecting perturbations***, such as noise, motion artifacts, or sensor degradation, in a controlled and systematic way. This allows researchers to isolate specific effects and systematically evaluate model performance under a wide variety of challenging conditions. Such flexibility is invaluable for stress-testing models and understanding their limitations, especially in real-world conditions where similar noise patterns might occur but in less predictable ways.

**Bias management and enhanced explainability.** While all datasets, real or synthetic, have inherent biases, the controllability of synthesized datasets makes it easier to identify, manage, and mitigate these biases. Real-world datasets are often subject to hidden biases introduced by uncontrolled environmental factors, sensor characteristics, or specific collection methodologies, which can skew model performance in unanticipated ways. In contrast, ***synthesized datasets allow for greater transparency and control over biases***, as perturbations can be systematically introduced and analyzed. This not only enhances the explainability of model behavior but also allows for better tuning and optimization, leading to models that are more robust to bias-related challenges in real-world deployment.

**Bridging the gap between noise-free and real-world conditions.** Finally, synthesized datasets play a critical role in bridging the gap between existing noise-free SLAM datasets and the complexities of real-world environments where noise, imperfect sensing, and challenging agent motions are common. While current real-world datasets offer valuable insights, the level of control and precision offered by synthesized datasets, particularly when simulating degraded sensor data and complex motion scenarios, makes them an essential tool for SLAM evaluation. As such, **our *Robust-Ego3D* benchmark** synthesizes realistic noise and perturbations under imperfect sensing and motion conditions, offering a more comprehensive evaluation platform that brings SLAM closer to real-world robustness.

## C DATASHEET FOR *Robust-Ego3D* BENCHMARK

We document the necessary information about the proposed datasets and benchmarks following the guidelines of Gebru *et al.* (Gebru et al., 2021).

### C.1 MOTIVATION

Q1 **For what purpose was the dataset created?** Was there a specific task in mind? Was there a specific gap that needed to be filled? Please provide a description.

- Our benchmark was created to holistically evaluate the robustness of dense Neural SLAM models under diverse perturbations. Prior to our work, dense Neural SLAM models were typically evaluated under noise-free settings. With our customizable perturbation synthesis pipeline, we evaluate the models across a wide range of RGB-D perturbations crucial for real-world deployment, including RGB imaging perturbations, depth imaging perturbations, motion-related perturbations, and RGB-D desynchronization.

Q2 **Who created the dataset (e.g., which team, research group) and on behalf of which entity (e.g., company, institution, organization)?**

- This benchmark is presented by [Anonymous Author][2]. Our aim is to advance the benchmarking, development, and deployment of more reliable and robust ego-motion estimation and photo realistic 3D reconstruction.

Q3 **Who funded the creation of the dataset?** If there is an associated grant, please provide the name of the grantor and the grant name and number.

- This work was partially supported by [Anonymous Author].

Q4 **Any other comments?**

- No.

### C.2 COMPOSITION

Q5 **What do the instances that comprise the dataset represent (e.g., documents, photos, people, countries)?** *Are there multiple types of instances (e.g., movies, users, and ratings; people and interactions between them; nodes and edges)? Please provide a description.*

- Our initialized *Robust-Ego3D* benchmark includes RGB-D video sequences rendered from scanned 3D scenes, the 6D trajectory at each timestamp, and the 3D scene point cloud.

Q6 **How many instances are there in total (of each type, if appropriate)?**

- The *Robust-Ego3D* benchmark includes 1,000 perturbed settings, each with 2,000 RGB-D video sequences. Detailed statistics for each scenario are provided in Sec. B.2 of the Appendix.

Q7 **Does the dataset contain all possible instances or is it a sample (not necessarily random) of instances from a larger set?** *If the dataset is a sample, what is the larger set? Is the sample representative of the larger set (e.g., geographic coverage)? If so, please describe how this representativeness was validated/verified. If it is not representative of the larger set, please describe why not (e.g., to cover a more diverse range of instances, because instances were withheld or unavailable).*

- The 3D scenes in our benchmark are sourced from the existing 3D scanned indoor scene dataset Replica (Straub et al., 2019), and we use all possible instances from these datasets.

Q8 **What data does each instance consist of?** *"Raw" data (e.g., unprocessed text or images) or features? In either case, please provide a description.*

- Each instance includes RGB-D images, trajectories, perturbation categories, and ground-truth 3D scene meshes.

---

[2]Author and repository information have been anonymized in compliance with the double-blind policy and will be provided upon acceptance.

Q9 **Is there a label or target associated with each instance?** *If so, please provide a description.*

- RGB-D video sequences for each perturbed setting are rendered from 3D scans in the Replica dataset (Straub et al., 2019).

Q10 **Is any information missing from individual instances?** *If so, please provide a description, explaining why this information is missing (e.g., because it was unavailable). This does not include intentionally removed information, but might include, e.g., redacted text.*

- No.

Q11 **Are relationships between individual instances made explicit (e.g., users' movie ratings, social network links)?** *If so, please describe how these relationships are made explicit.*

- Each RGB-D video sequence is explicitly linked to a specific 3D scene in the Replica dataset, defined by a unique trajectory and a set of perturbations.

Q12 **Are there recommended data splits (e.g., training, development/validation, testing)?** *If so, please provide a description of these splits, explaining the rationale behind them.*

- No, our benchmark is primarily intended for evaluation, as RGB-D SLAM models use online optimization/adaptation and do not require a separate training stage. However, it can be extended for model training if needed.

Q13 **Are there any errors, sources of noise, or redundancies in the dataset?** *If so, please provide a description.*

- No.

Q14 **Is the dataset self-contained, or does it link to or otherwise rely on external resources (e.g., websites, tweets, other datasets)?** *If it links to or relies on external resources, a) are there guarantees that they will exist, and remain constant, over time; b) are there official archival versions of the complete dataset (i.e., including the external resources as they existed at the time the dataset was created); c) are there any restrictions (e.g., licenses, fees) associated with any of the external resources that might apply to a future user? Please provide descriptions of all external resources and any restrictions associated with them, as well as links or other access points, as appropriate.*

- The benchmark is self-contained. We provide all the details and instructions at [Anonymous Repo].

Q15 **Does the dataset contain data that might be considered confidential (e.g., data that is protected by legal privilege or by doctor–patient confidentiality, data that includes the content of individuals' non-public communications)?** *If so, please provide a description.*

- No. The 3D scans used in our *Robust-Ego3D* benchmark are sourced from existing open-source datasets.

Q16 **Does the dataset contain data that, if viewed directly, might be offensive, insulting, threatening, or might otherwise cause anxiety?** *If so, please describe why.*

- No.

Q17 **Does the dataset relate to people?** *If not, you may skip the remaining questions in this section.*

- No. This dataset does not relate to people.

Q18 **Does the dataset identify any subpopulations (e.g., by age, gender)?**

- N/A.

Q19 **Is it possible to identify individuals (i.e., one or more natural persons), either directly or indirectly (i.e., in combination with other data) from the dataset?** *If so, please describe how.*

- N/A.

Q20 **Does the dataset contain data that might be considered sensitive in any way (e.g., data that reveals racial or ethnic origins, sexual orientations, religious beliefs, political opinions or union memberships, or locations; financial or health data; biometric or genetic data; forms of government identification, such as social security numbers; criminal history)?** *If so, please provide a description.*

- No.

Q21 **Any other comments?**

- We caution discretion on behalf of the user and call for responsible usage of the benchmark for research purposes only.

## C.3 COLLECTION PROCESS

Q22 **How was the data associated with each instance acquired?** *Was the data directly observable (e.g., raw text, movie ratings), reported by subjects (e.g., survey responses), or indirectly inferred/derived from other data (e.g., part-of-speech tags, model-based guesses for age or language)? If data was reported by subjects or indirectly inferred/derived from other data, was the data validated/verified? If so, please describe how.*

- The 3D scenes used for SLAM data generation in our benchmark are sourced from the existing open-source dataset Replica (Straub et al., 2019). The details of the benchmark construction are provided in the Experiment section of the main paper, with further details in Sec. B of the Appendix.

Q23 **What mechanisms or procedures were used to collect the data (e.g., hardware apparatus or sensor, manual human curation, software program, software API)?** *How were these mechanisms or procedures validated?*

- No additional raw data was collected. Our contribution is the development of a perturbation taxonomy and toolbox to transform existing clean SLAM datasets and 3D scenes into noisy datasets for robustness evaluation.

Q24 **If the dataset is a sample from a larger set, what was the sampling strategy (e.g., deterministic, probabilistic with specific sampling probabilities)?**

- No new raw data was collected. RGB-D sensor streams were rendered using 3D scene models from the Replica dataset (Straub et al., 2019), consisting of real indoor scene scans. We used the same eight rooms and offices as the (clean) Replica-SLAM dataset (Sucar et al., 2021) to ensure consistency in comparisons.

Q25 **Who was involved in the data collection process (e.g., students, crowdworkers, contractors) and how were they compensated (e.g., how much were crowdworkers paid)?**

- N/A. Our benchmark does not include new raw data collection.

Q26 **Over what timeframe was the data collected? Does this timeframe match the creation timeframe of the data associated with the instances (e.g., recent crawl of old news articles)?** *If not, please describe the timeframe in which the data associated with the instances was created.*

- N/A. Our benchmark does not include new raw data collection.

Q27 **Were any ethical review processes conducted (e.g., by an institutional review board)?** *If so, please provide a description of these review processes, including the outcomes, as well as a link or other access point to any supporting documentation.*

- N/A. Our benchmark does not include new raw data collection.

Q28 **Does the dataset relate to people?** *If not, you may skip the remaining questions in this section.*

- No.

Q29 **Did you collect the data from the individuals in question directly, or obtain it via third parties or other sources (e.g., websites)?**

- N/A. Our dataset does not relate to people.

Q30 **Were the individuals in question notified about the data collection?** *If so, please describe (or show with screenshots or other information) how notice was provided, and provide a link or other access point to, or otherwise reproduce, the exact language of the notification itself.*

- N/A. Our dataset does not relate to people.

Q31 **Did the individuals in question consent to the collection and use of their data?** *If so, please describe (or show with screenshots or other information) how consent was requested and provided, and provide a link or other access point to, or otherwise reproduce, the exact language to which the individuals consented.*

- N/A. Our dataset does not relate to people.

Q32 **If consent was obtained, were the consenting individuals provided with a mechanism to revoke their consent in the future or for certain uses?** *If so, please provide a description, as well as a link or other access point to the mechanism (if appropriate).*

- N/A. Our dataset does not relate to people.

Q33 **Has an analysis of the potential impact of the dataset and its use on data subjects (e.g., a data protection impact analysis) been conducted?** *If so, please provide a description of this analysis, including the outcomes, as well as a link or other access point to any supporting documentation.*

- We discuss the limitations of our current work in the Conclusion and Future Work section of the main paper, and we plan to further investigate and analyze the impact of our benchmark in future work. We acknowledge the potential data biases and limitations of our initial benchmark and have detailed the assumptions made during benchmark construction in Sec. B.1 of the Appendix.

Q34 **Any other comments?**

- No.

### C.4 PREPROCESSING, CLEANING, AND/OR LABELING

Q35 **Was any preprocessing/cleaning/labeling of the data done (e.g., discretization or bucketing, tokenization, part-of-speech tagging, SIFT feature extraction, removal of instances, processing of missing values)?** *If so, please provide a description. If not, you may skip the remainder of the questions in this section.*

- No preprocessing or labeling was performed.

Q36 **Was the "raw" data saved in addition to the preprocessed/cleaned/labeled data (e.g., to support unanticipated future uses)?** *If so, please provide a link or other access point to the "raw" data.*

- N/A. No preprocessing or labeling was performed for creating the scenarios.

Q37 **Is the software used to preprocess/clean/label the instances available?** *If so, please provide a link or other access point.*

- N/A. No preprocessing or labeling was performed for creating the scenarios.

Q38 **Any other comments?**

- No.

### C.5 USES

Q39 **Has the dataset been used for any tasks already?** *If so, please provide a description.*

- Not yet. We present a new benchmark.

Q40 **Is there a repository that links to any or all papers or systems that use the dataset?** *If so, please provide a link or other access point.*

- We will provide links to works that use our benchmark at [Anonymous Repo].

Q41 **What (other) tasks could the dataset be used for?**

- Our benchmark primarily studies the robustness of dense dense Neural SLAM models under perturbations.
- Its customizable nature allows for future research on the robustness of various SLAM systems and other RGB-D sensing-related tasks.

Q42 **Is there anything about the composition of the dataset or the way it was collected and preprocessed/cleaned/labeled that might impact future uses?** *For example, is there anything that a future user might need to know to avoid uses that could result in unfair treatment of individuals or groups (e.g., stereotyping, quality of service issues) or other undesirable harms (e.g., financial harms, legal risks)? If so, please provide a description. Is there anything a future user could do to mitigate these undesirable harms?*

   • No.

Q43 **Are there tasks for which the dataset should not be used?** *If so, please provide a description.*

   • No.

Q44 **Any other comments?**

   • No.

### C.6 DISTRIBUTION AND LICENSE

Q45 **Will the dataset be distributed to third parties outside of the entity (e.g., company, institution, organization) on behalf of which the dataset was created?** *If so, please provide a description.*

   • Yes, this benchmark has been open-sourced.

Q46 **How will the dataset be distributed (e.g., tarball on website, API, GitHub)?** *Does the dataset have a digital object identifier (DOI)?*

   • Our benchmark and the code used for evaluation are available at [Anonymous Repo].

Q47 **When will the dataset be distributed?**

   • Sep, 2024, and onward.

Q48 **Will the dataset be distributed under a copyright or other intellectual property (IP) license, and/or under applicable terms of use (ToU)?** *If so, please describe this license and/or ToU, and provide a link or other access point to, or otherwise reproduce, any relevant licensing terms or ToU, as well as any fees associated with these restrictions.*

   • The 3D scene dataset Replica and the benchmarking methods used in our benchmark are sourced from existing open-source repositories, as illustrated in Sec. O. The license associated with them is followed accordingly.
   • Our code is released under the Apache-2.0 license.

Q49 **Have any third parties imposed IP-based or other restrictions on the data associated with the instances?** *If so, please describe these restrictions, and provide a link or other access point to, or otherwise reproduce, any relevant licensing terms, as well as any fees associated with these restrictions.*

   • We release it under the Apache-2.0 license.
   • Copyright for the original 3D scenes used for rendering is not owned by us.

Q50 **Do any export controls or other regulatory restrictions apply to the dataset or to individual instances?** *If so, please describe these restrictions, and provide a link or other access point to, or otherwise reproduce, any supporting documentation.*

   • No.

Q51 **Any other comments?**

   • No.

### C.7 MAINTENANCE

Q52 **Who will be supporting/hosting/maintaining the dataset?**

   • Anonymous Author

Q53 **How can the owner/curator/manager of the dataset be contacted (e.g., email address)?**

   • Anonymous Author

Q54 **Is there an erratum?** *If so, please provide a link or other access point.*

- There is no erratum for our initial release. Errata will be documented as future releases on the benchmark website.

Q55 **Will the dataset be updated (e.g., to correct labeling errors, add new instances, delete instances)?** *If so, please describe how often, by whom, and how updates will be communicated to users (e.g., mailing list, GitHub)?*

- Yes, our benchmark will be updated. We plan to expand scenarios, metrics, and models to be evaluated.

Q56 **If the dataset relates to people, are there applicable limits on the retention of the data associated with the instances (e.g., were individuals in question told that their data would be retained for a fixed period of time and then deleted)?** *If so, please describe these limits and explain how they will be enforced.*

- N/A. Our dataset does not relate to people.

Q57 **Will older versions of the dataset continue to be supported/hosted/maintained?** *If so, please describe how. If not, please describe how its obsolescence will be communicated to users.*

- We will host other versions.

Q58 **If others want to extend/augment/build on/contribute to the dataset, is there a mechanism for them to do so?** *If so, please provide a description. Will these contributions be validated/verified? If so, please describe how. If not, why not? Is there a process for communicating/distributing these contributions to other users? If so, please provide a description.*

- Users may contact us by reporting an issue on our benchmark GitHub page or directly contacting the author of this project ([Anonymous Author]) to request adding new scenarios, metrics, or models.

Q59 **Any other comments?**

- No.

# D ADDITIONAL RESULTS AND ANALYSES ON *Robust-Ego3D* BENCHMARK

## D.1 MORE BENCHMARKING ANALYSES

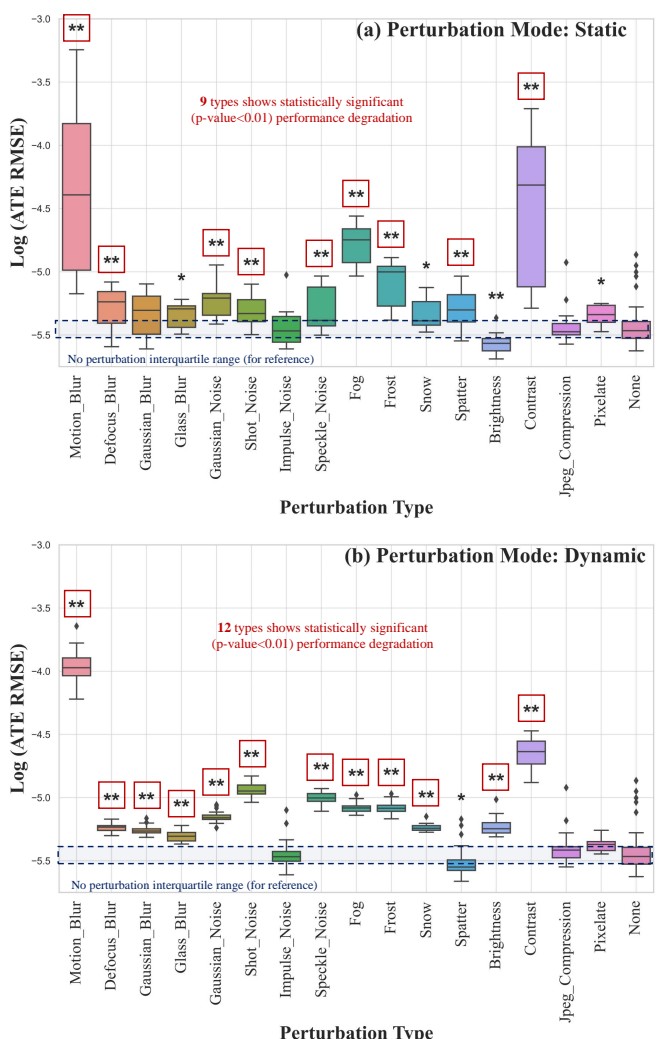

Figure D9: Effect of each RGB imaging perturbation type on the trajectory estimation performance of SplaTAM-S (Keetha et al., 2024) model, which shows the best overall performance under RGB imaging perturbations among all the benchmark methods. The t-test (Kim, 2015) is performed to compare the performance distribution between each perturbed setting and the perturbation-free setting (which is denoted as *None* in the last column of each sub-figure). ** and * indicate a significant distribution difference between the pair at the 0.01 and 0.05 significance level, respectively.

**More analyses on the most robust SLAM model under RGB imaging perturbation, *i.e.*, SplaTAM-S.** Even the top-performing SplaTAM-S model, which achieves the best average performance under RGB imaging perturbations, experiences a more substantial decrease in trajectory estimation accuracy under dynamic conditions, with statistically significant differences observed for most of the tested perturbation types (see Fig. D9). Interestingly, increased brightness, while slightly beneficial under static conditions, leads to significant errors under dynamic conditions. In Fig. D10, we examine the impact of perturbation strength on SplaTAM-S. While the trajectory estimation error and geometry rendering quality (measured by Depth L1) remain mostly stable, the rendered RGB image quality (measured by PSNR, MS-SSIM, and LPIPS) degrades significantly as perturbation severity increases.

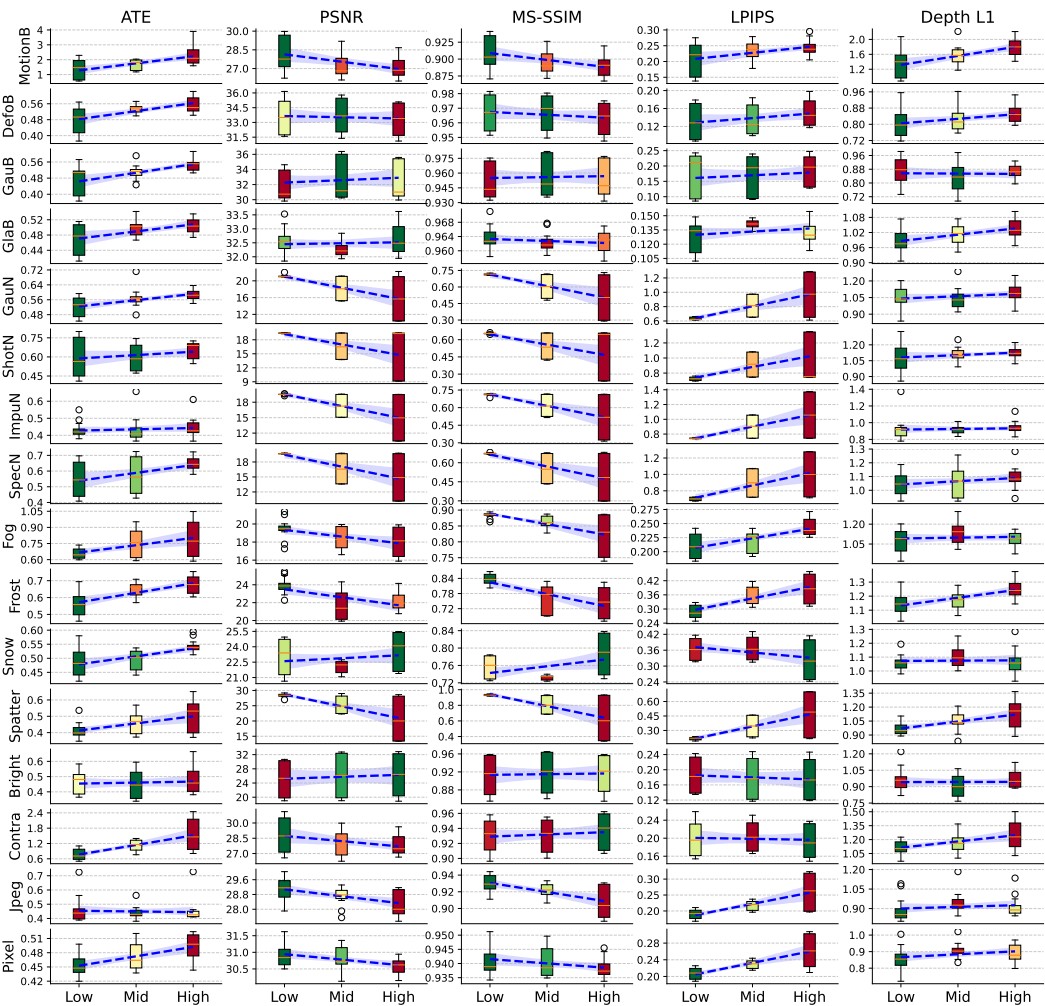

Figure D10: Effect of varied perturbation severity of static RGB imaging perturbations on performance of trajectory estimation (measured by ATE↓ (cm)), RGB rendering quality (measured by PSNR↑, MS-SSIM↑, and LPIPS↓), and depth rendering quality (measured by Depth L1 loss↓ (cm)) for SplaTAM-S. The fitted trend lines are illustrated in blue.

Table D4: **3D metrics for evaluation of mesh reconstruction quality** of the reconstructed 3D mesh $P$ when given the ground-truth 3D mesh $Q$ (in the scale of meter [m]). We follows the 3D reconstruction metrics defined in the CO-SLAM (Wang et al., 2023) paper.

| 3D Reconstruction Metric | Definition |
|---|---|
| Accuracy (ACC) | $\frac{1}{|P|} \sum_{p \in P} \left( \min_{q \in Q} ||p - q||^2 \right)$ |
| Completion (Comp.) | $\frac{1}{|Q|} \sum_{q \in Q} \left( \min_{p \in P} ||p - q||^2 \right)$ |
| Completion Ratio (Comp. R.) | $\frac{1}{|Q|} \sum_{q \in Q} \left( \min_{p \in P} ||p - q||^2 \leq 0.05 \right)$ |

Table D5: Effects of RGB imaging perturbation on mapping for CO-SLAM.

| Metrics | Clean Mean | Low Severity | Middle Severity | High Severity | Perturb. Mean |
|---|---|---|---|---|---|
| ACC↓ [cm] | **2.08** | 2.11 | 2.12 | 2.39 | 2.21 |
| Comp.↓ [cm] | **2.17** | 2.19 | 2.20 | 2.89 | 2.43 |
| Comp. R.↑ [%] | **93.13** | 93.07 | 93.04 | 92.34 | 92.82 |

**Effect of RGB imaging perturbations on the geometry quality of mapping.** We follow the mapping quality evaluation protocol in (Wang et al., 2023) to assess 3D reconstruction using Accuracy (ACC) [cm], Completion (Comp.) [cm], and Completion Ratio (Comp. R.) [%] with a 5 cm threshold. Table D4 details the definition for each of these metrics. Note that only certain dense SLAM models can produce 3D reconstruction results for further evaluation of mapping quality. In Table D5, we evaluate the impact of image perturbations on the mapping quality of the CO-SLAM (Wang et al., 2023) model, which shows a strong robustness to most of the image-level corruptions. The results reveal a direct correlation between perturbation severity and both 3D reconstruction error and completion error. Specifically, the clean setting achieves the highest accuracy (2.08 cm) and the lowest completeness score (2.17 cm), while the high perturbation severity setting exhibits the highest errors in ACC (2.39 cm) and completion (2.89 cm). Overall, our analysis shows that increasing severity levels of perturbation lead to larger errors in the reconstructed 3D map.

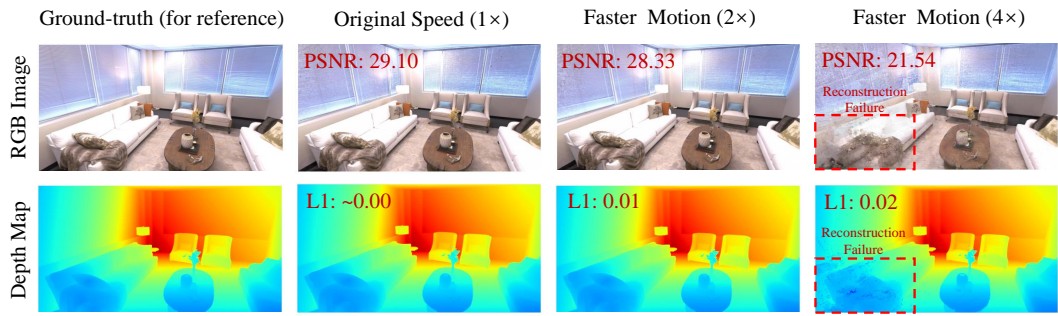

Figure D11: Effect of faster motion on the 2D reconstruction losses of RGB images (**Left**) and depth maps (**Right**), which are measured via PSNR and Depth L1 loss, for SplaTAM-S (Keetha et al., 2024).

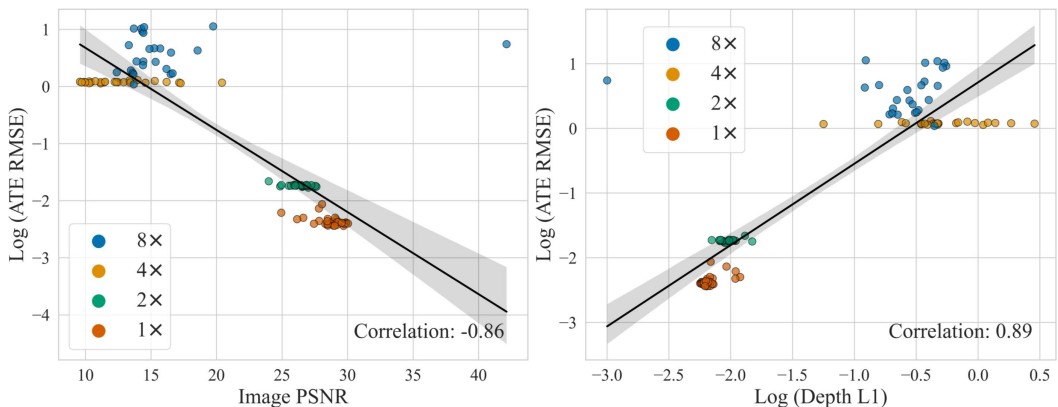

Figure D12: Correlation between ATE (logarithm form) and the 2D reconstruction losses of RGB images (**Left**) and depth maps (**Right**), which are used for model optimization, under faster motion effects for SplaTAM-S (Keetha et al., 2024). Pearson correlation coefficient (Cohen et al., 2009) is reported in the bottom-right corner.

**There exists SLAM models that can perceive perturbed observations.** We conduct a case study to explore the ability of SplaTAM-S model to perceive the severity of perturbations. Notice that SplaTAM-S optimizes the map and 3D pose by minimizing the 2D reconstruction loss for the RGB and depth maps during inference. In Fig. D11 and Fig. D12, we observe a strong correlation between the accuracy of the final trajectory estimation and the RGB-D reconstruction loss. This indicates that when the model produces a larger reconstruction loss for a certain sensor stream, it is likely that the trajectory estimation is also inaccurate. While this doesn't provide exact localization, it serves as a valuable indicator of potential observation degradation and model failure. This suggests SLAM systems could self-monitor performance using internal indicators, enabling real-time failure mitigation in safety-critical applications.

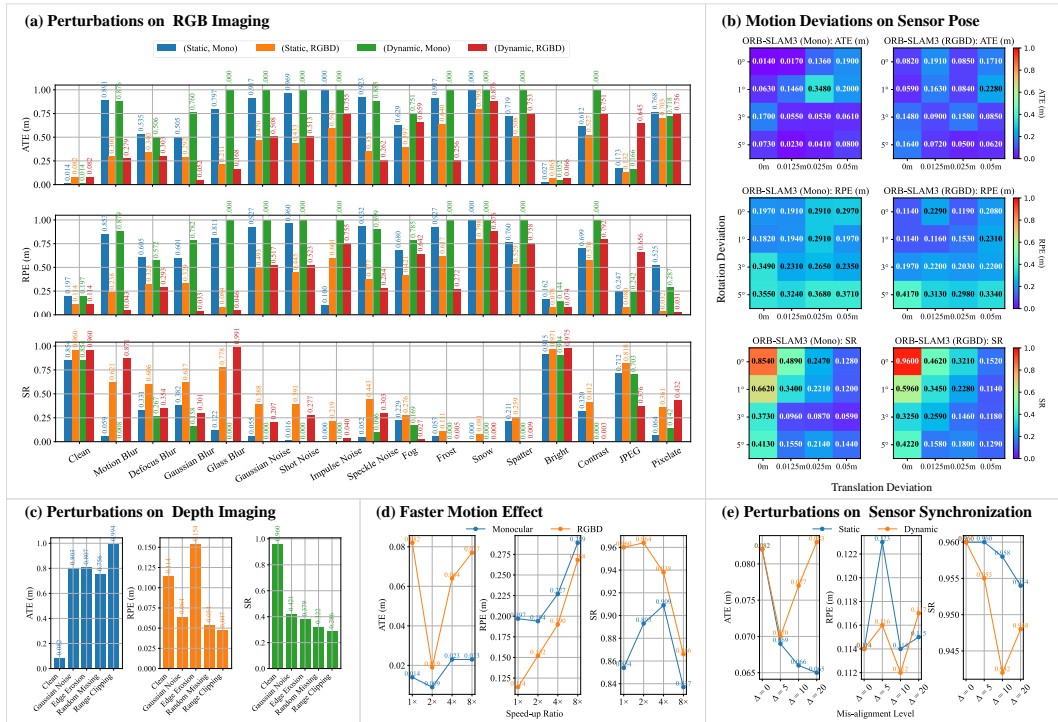

Figure D13: Performance (measured by ATE↓ (m), RPE↓ (m), and SR↑) of ORB-SLAM3 (Campos et al., 2021) under diverse perturbations. For visualization, sequences resulting in failure are assigned an ATE/RPE value of 1.0 and a Success Rate of 0.

## D.2 CLASSICAL SLAM ORB-SLAM3 RESULTS (FOR REFERENCE)

To better understand the strengths and weaknesses of learning-based dense SLAM models, we also include results from the classical SLAM model ORB-SLAM3 (Campos et al., 2021) for reference and comparison. We evaluate its performance using Absolute Trajectory Error (ATE), Relative Pose Error (RPE), and Success Rate (SR). SR is defined as the ratio of the cumulative sum of Euclidean distances between consecutive estimated poses to the cumulative sum of Euclidean distances between consecutive ground truth poses. Formally, the SR metric is defined as:

$$SR = \frac{\sum_{i=1}^{N} ||p_i^{\text{est}} - p_{i-1}^{\text{est}}||}{\sum_{i=1}^{N} ||p_i^{\text{gt}} - p_{i-1}^{\text{gt}}||} \tag{D10}$$

where $|| \cdot ||$ denotes the Euclidean distance.

Lower ATE&RPE (↓) and higher SR (↑) indicate better performance, with ORB-SLAM3's results under various perturbations presented in Fig D13. Since ORB-SLAM3 lacks photorealistic 3D mapping capability, it is excluded from the main comparison, which focuses on dense Neural SLAM models designed for high-fidelity reconstructions.

### D.3 MORE QUALITATIVE ANALYSES OF MODEL UNDER PERTURBATIONS

**Nice-SLAM excels in 3D geometry reconstruction under shot noise but fails in color detail reconstruction and loses tracking under rapid motion. 1**) Fig. D14 showcases the 3D reconstruction and trajectory estimation results of the Nice-SLAM (Zhu et al., 2022) model under varying levels of shot noise perturbation on the RGB image. Nice-SLAM consistently produces high-quality geometry reconstructions even when subjected to high severity levels of shot noise, which we attribute to the nearly error-free, unperturbed depth map aiding geometry reconstruction. However, we observe that the model struggles to accurately predict and reconstruct appearance details. Consequently, as the noise in the RGB images intensifies, color reconstruction quality diminishes. **2**) Fig. D15, Fig. D16, Fig. D17, and Fig. D18 demonstrates the complete failure of the Nice-SLAM model in reconstructing 3D geometry and maintaining tracking under rapid motion.

**SplaTAM-S shows strong robustness in 3D reconstruction under motion blur but fails under severe contrast reduction, losing tracking and reconstruction capabilities. 1**) Fig. D19 presents the qualitative results of the SplaTAM-S (Keetha et al., 2024) model under different severity levels of motion blur image-level perturbations. The trajectory estimation reveals that, in the absence of perturbation or with low levels of motion blur, the model produces smooth trajectories. However, as perturbation severity increases to a moderate or high level, the predicted trajectory exhibits more deviations. Notably, the 3D reconstruction consistently maintains high quality despite the increasing blurring caused by observation degradation. **2**) Fig. D20 and Fig. D21 depict failure instances of the SplaTAM-S (Keetha et al., 2024) model. These failures occur when subjected to varying levels of contrast decrease image-level perturbation under both static and dynamic perturbation modes. Higher severity levels of perturbation result in complete tracking loss and reconstruction failure.

**NeRF as a locally noise-tolerant representation (for certain perturbations like *Spatter Effect*) for iMAP and Nice-SLAM. 1**) The RGB-D rendering and 3D reconstruction results depicted in Fig. D22, Fig. D24, Fig. D23, and Fig. D25 demonstrate the impressive local de-noising capabilities of iMAP (Sucar et al., 2021) and Nice-SLAM (Zhu et al., 2022) under severe spatter noise in RGB imaging. Both methods generate RGB images and reconstruct 3D meshes that are nearly free of spatter, showcasing their robustness in handling local disturbances. **2**) iMAP, as a NeRF-based SLAM system, leverages the power of neural representations to effectively capture scene features despite significant noise. Its feed-forward network (FFN) learns compact and noise-robust representations, allowing adaptive refinement and selective artifact suppression. This capability highlights the importance of neural representation in achieving effective local de-noising. **3**) Similarly, Nice-SLAM also shows strong local noise-tolerant capabilities, benefiting from the strengths of the iMAP model. Both methods exemplify the role of neural representations in maintaining clean visual data under challenging conditions.

**ORB-SLAM3's reliance on handcrafted ORB feature detection makes it highly sensitive to image perturbations, leading to increased trajectory estimation error (ATE) compared to the more robust, learned features used by Neural SLAM methods.** While ORB-SLAM3 demonstrates resilience to certain types of image corruption such as brightness changes and defocus blur (Fig.D27), its performance significantly degrades under perturbations that impair ORB feature detection. As shown in Fig. D28, increasing levels of Gaussian blur result in a noticeable reduction in the number of detected ORB features. This reduction correlates with a rise in ATE, as depicted in Fig. D29, indicating that the degradation of feature descriptors directly impairs trajectory estimation accuracy. Noise-related perturbations have an even more pronounced effect, often causing complete tracking failure due to the susceptibility of handcrafted keypoints to noise (Fig. D30). In contrast, dense Neural SLAM methods employ learned features that adapt better to noise and maintain robustness under challenging visual conditions, highlighting a key limitation of traditional feature-based SLAM models like ORB-SLAM3.

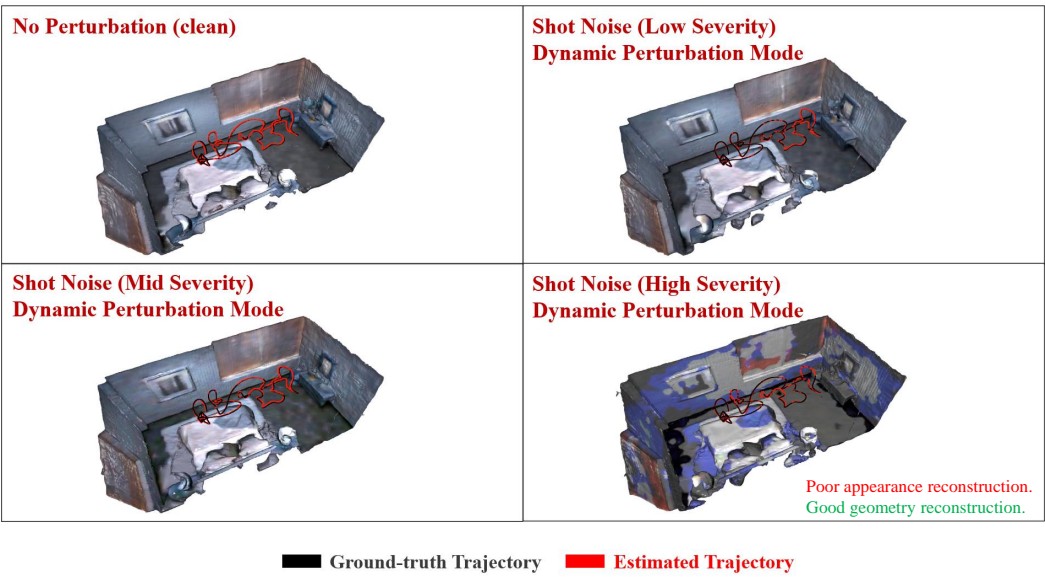

Figure D14: **Nice-SLAM (Zhu et al., 2022) excels in 3D geometry reconstruction under *Shot Noise* RGB imaging perturbations but fails in color detail reconstruction under high-severity of perturbations.**

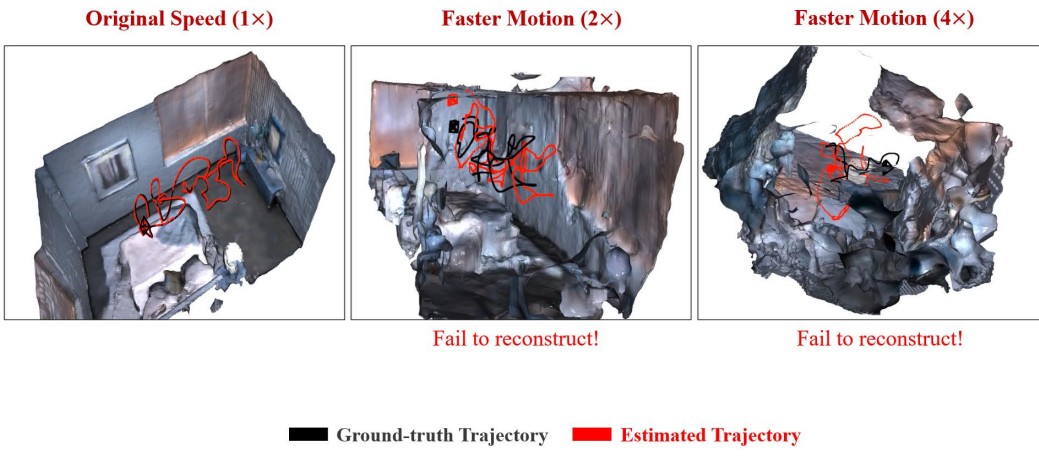

Figure D15: **Qualitative failure results of Nice-SLAM (Zhu et al., 2022) under fast motion.**

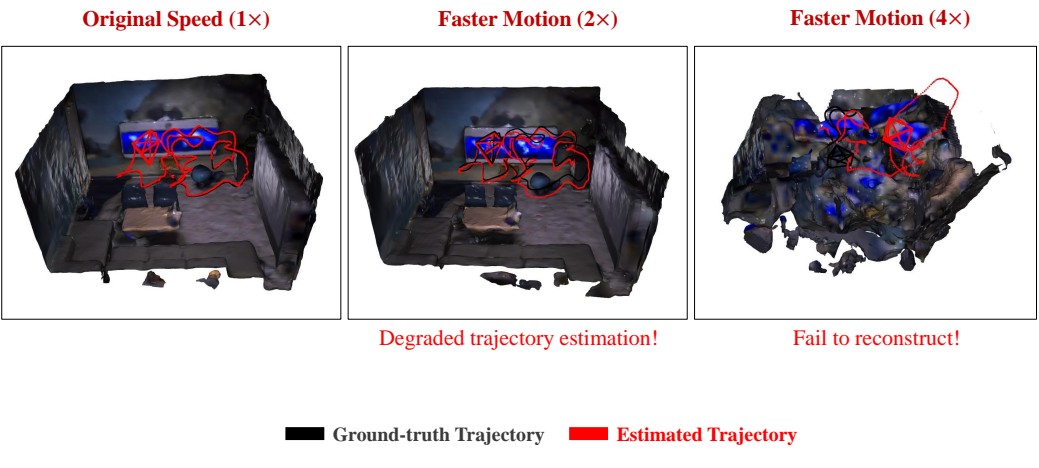

Figure D16: **Qualitative failure results of Nice-SLAM (Zhu et al., 2022) under fast motion.**

**Original Speed (1×)**   **Faster Motion (2×)**   **Faster Motion (4×)**

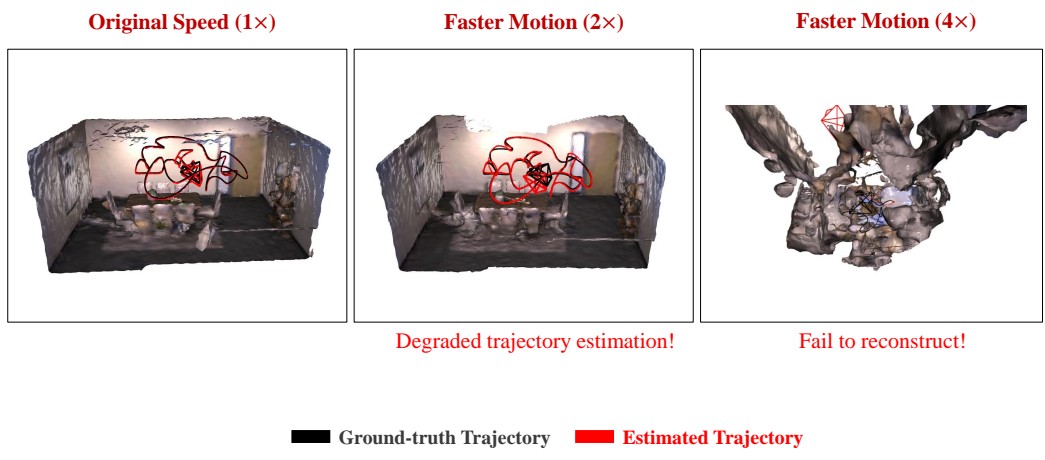

Degraded trajectory estimation!   Fail to reconstruct!

■ Ground-truth Trajectory  ■ Estimated Trajectory

Figure D17: **Qualitative failure results of Nice-SLAM (Zhu et al., 2022) under fast motion.**

**Original Speed (1×)**   **Faster Motion (2×)**   **Faster Motion (4×)**

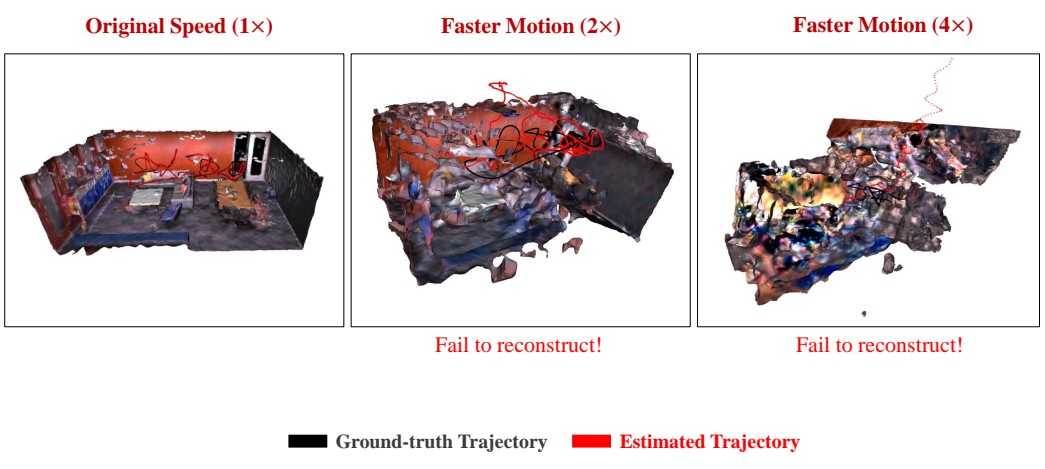

Fail to reconstruct!   Fail to reconstruct!

■ Ground-truth Trajectory  ■ Estimated Trajectory

Figure D18: **Qualitative failure results of Nice-SLAM (Zhu et al., 2022) under fast motion.**

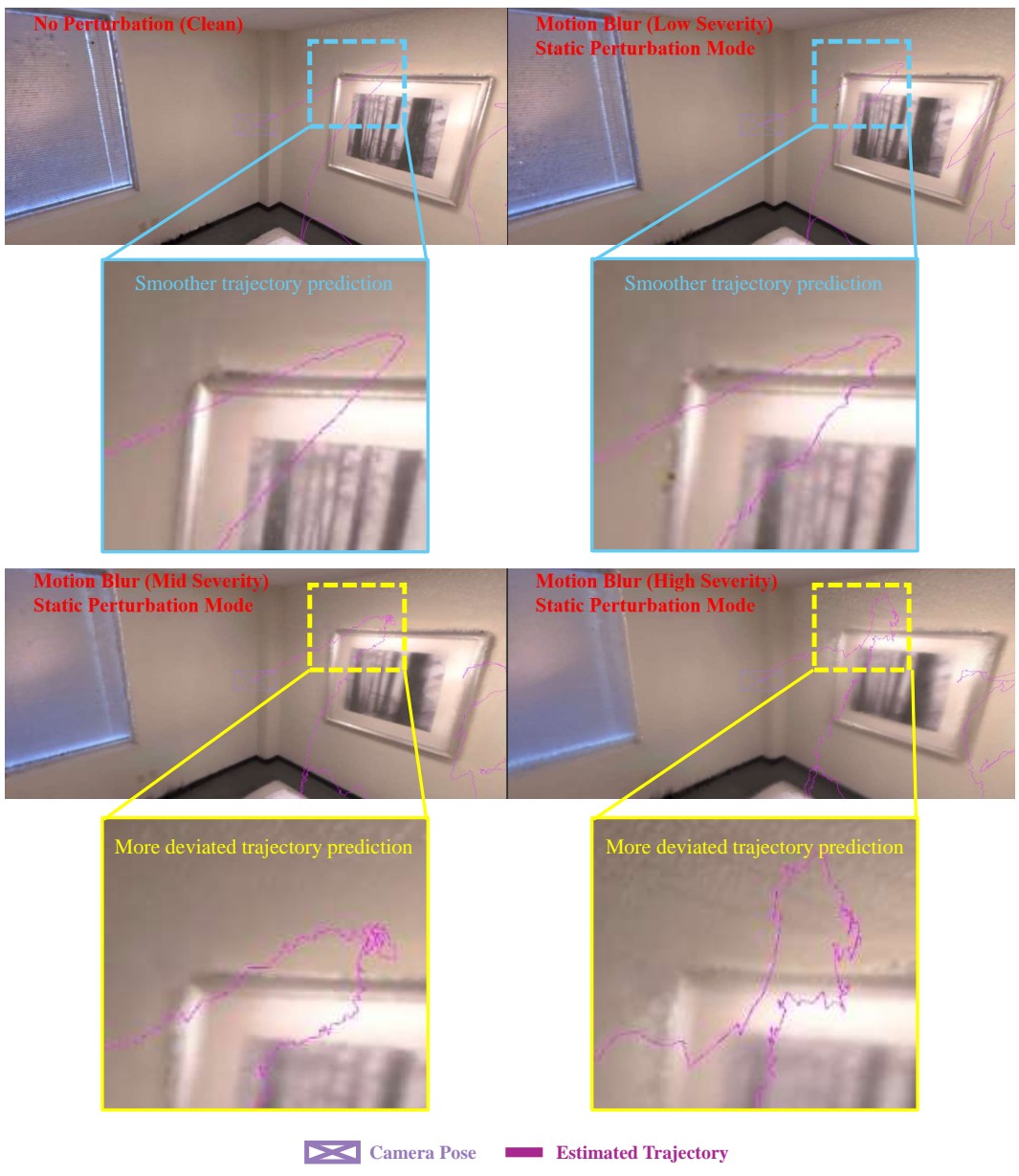

Figure D19: **Qualitative results of successful cases of SplaTAM-S model** (Keetha et al., 2024) **with RGB-D input.**

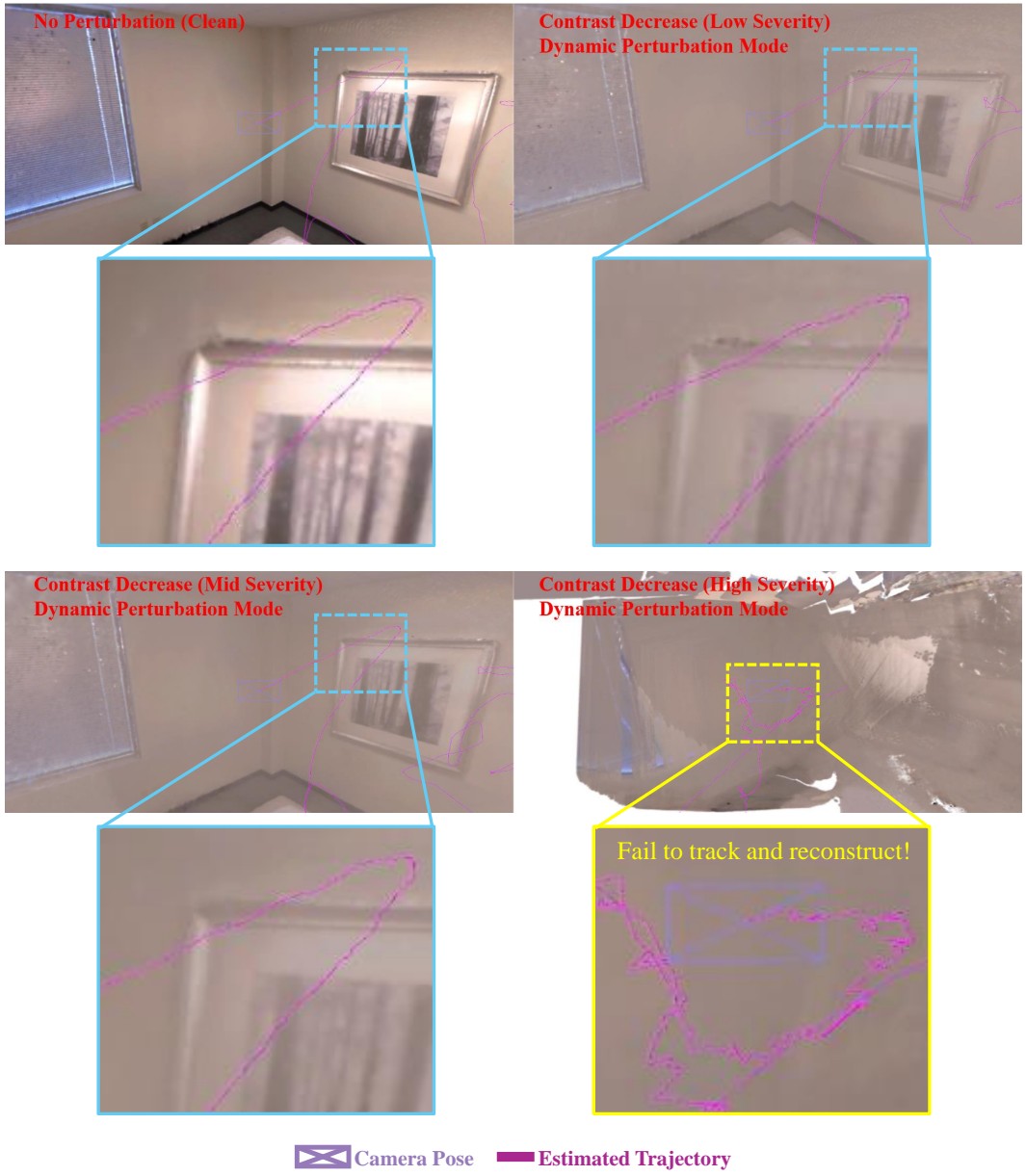

Figure D20: **Qualitative results of the failure cases of SplaTAM-S model** (Keetha et al., 2024) **with RGB-D input.**

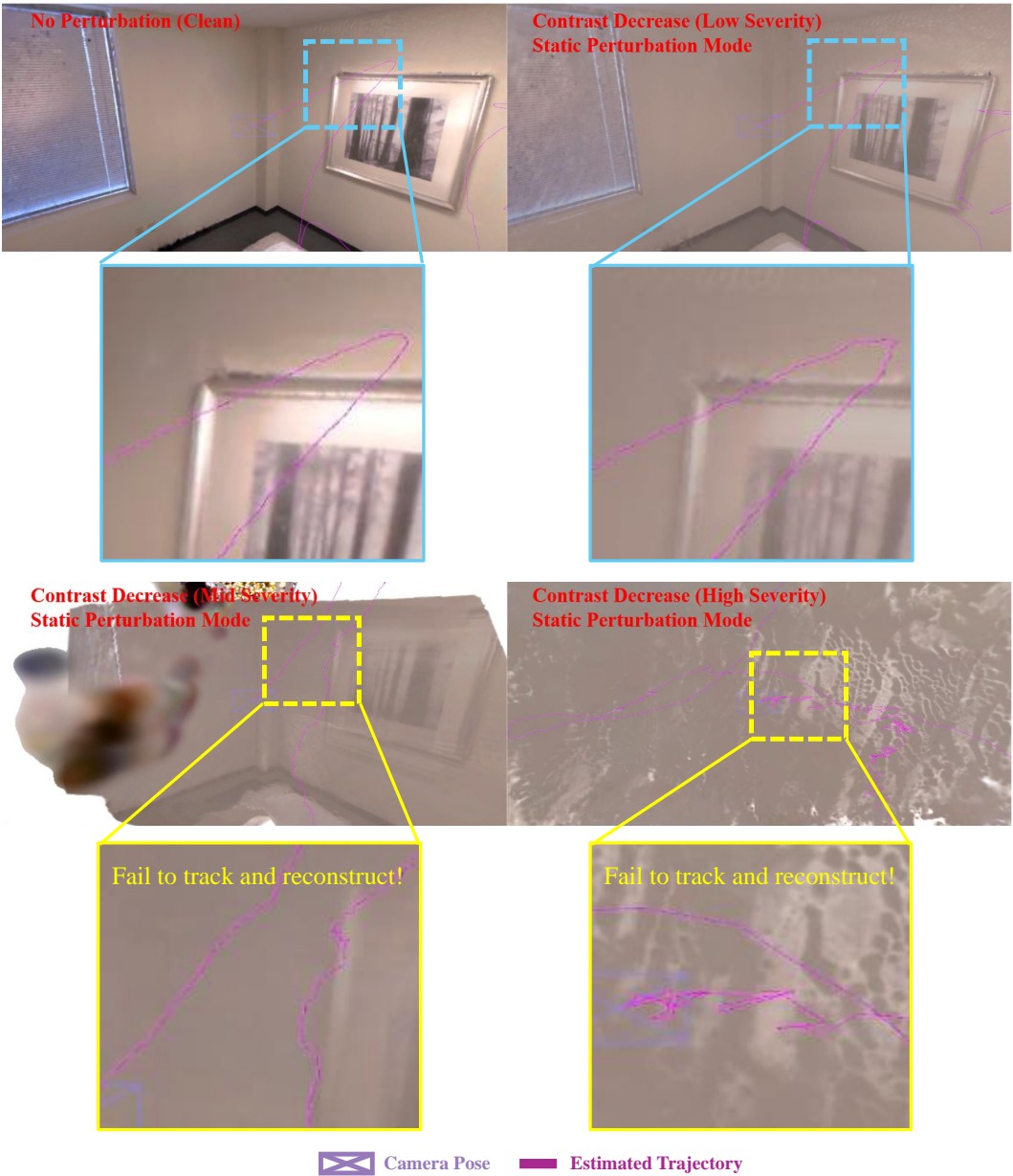

Figure D21: **Qualitative results of the failure cases of SplaTAM-S model (**Keetha et al., 2024**) with RGB-D input.**

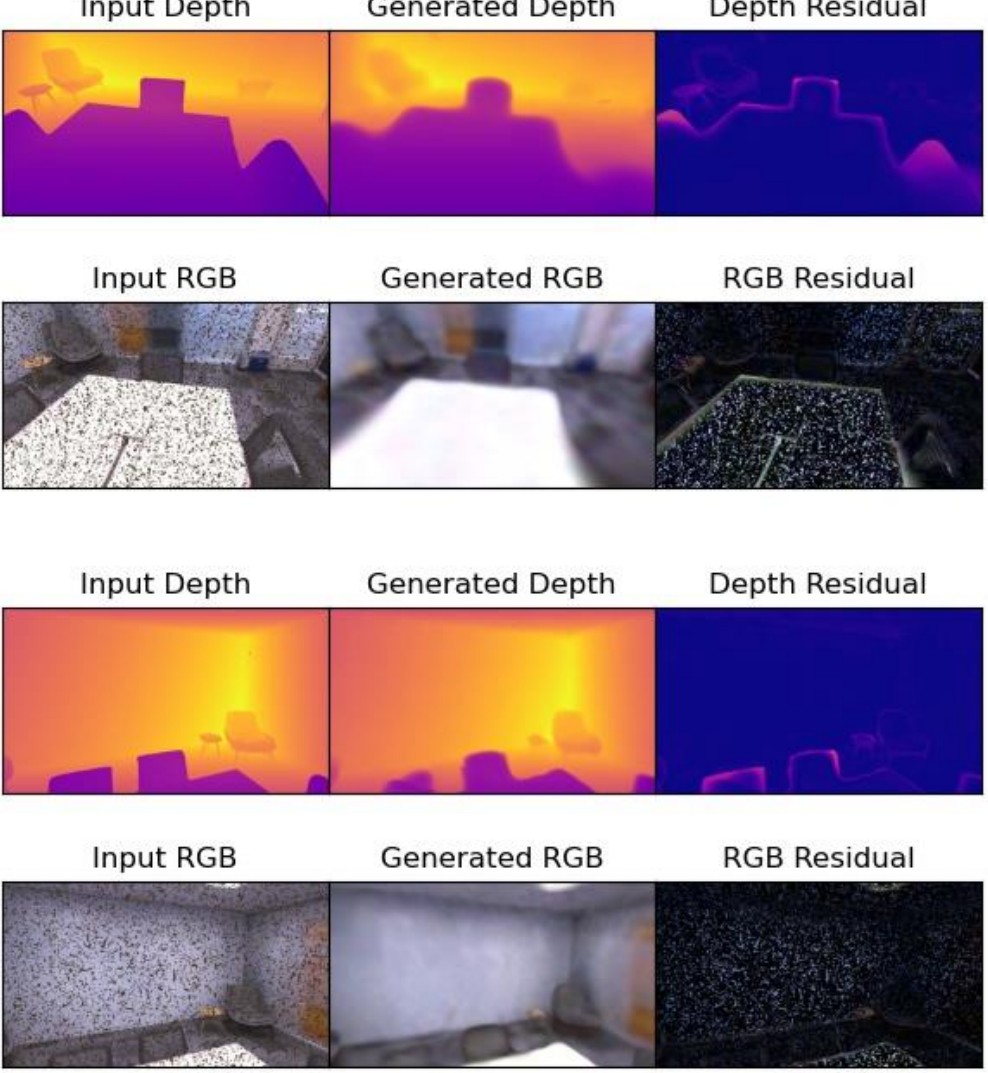

Figure D22: **Intriguing local de-noising capability of iMAP** (Sucar et al., 2021) under severe Spatter effect perturbations in RGB imaging. The generated RGB, rendered from the implicit representation, is nearly free of spatter noise.

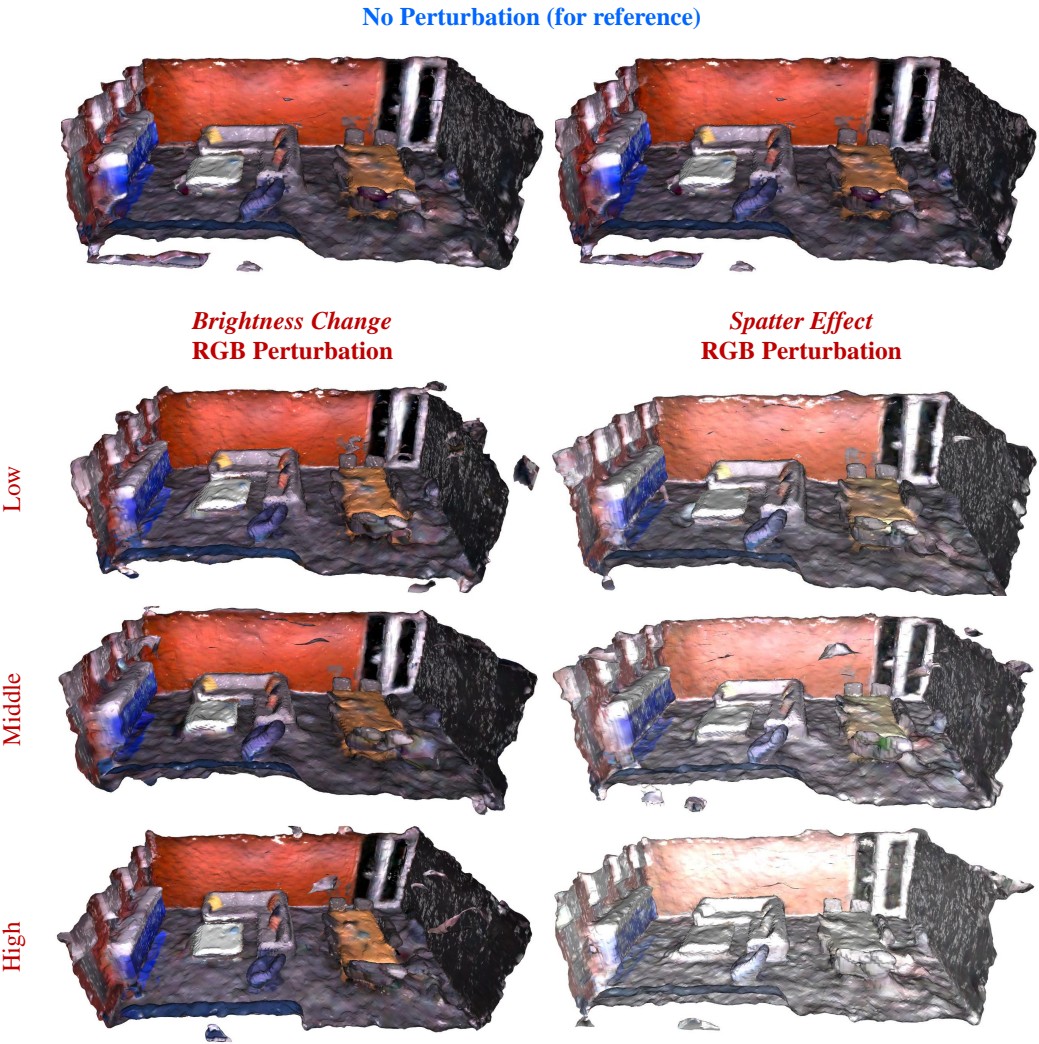

Figure D23: **Local de-noising capability of iMAP (Sucar et al., 2021) under different severities of** *Spatter Effect* **perturbations.** The reconstruction is nearly spatter-free (**Left**), but global perturbations like brightness changes (**Right**) cause color shifts in the reconstructed 3D mesh.

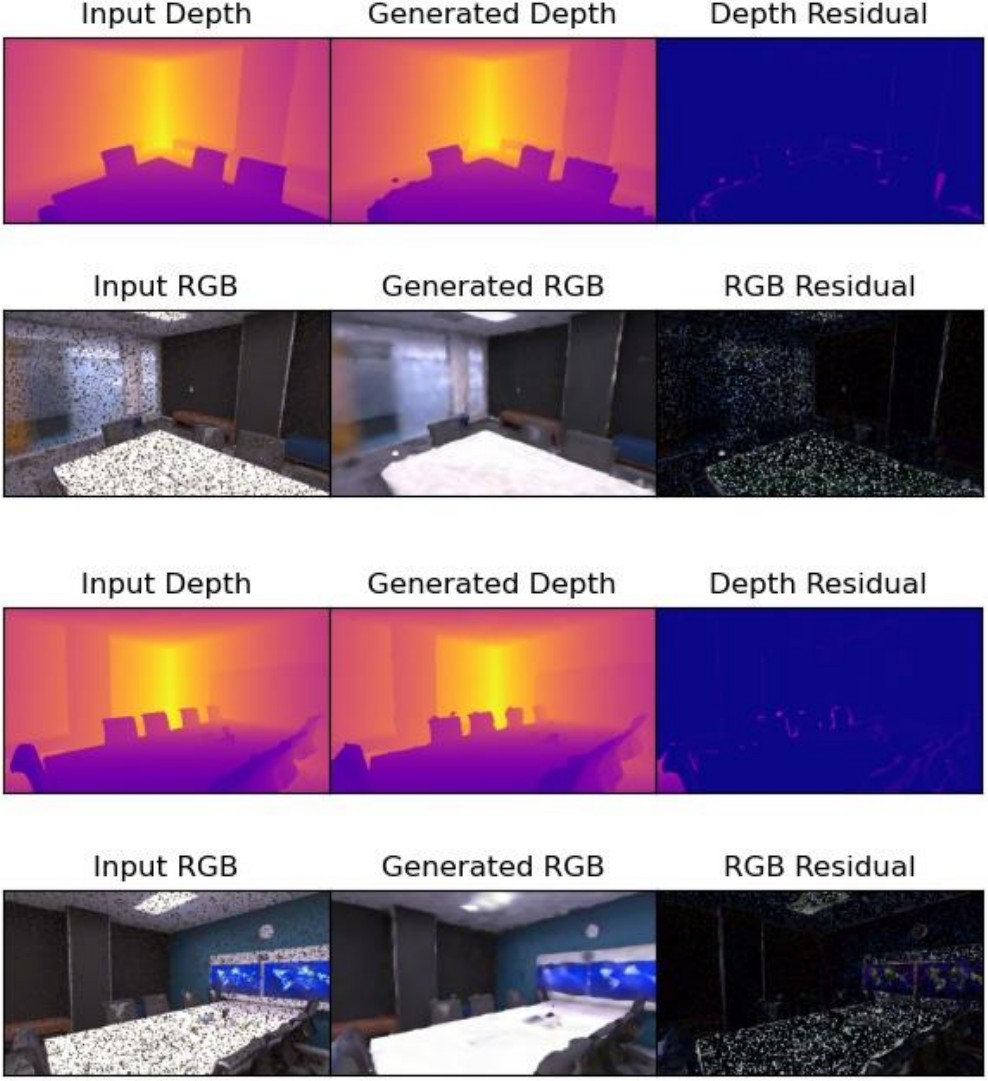

Figure D24: **Intriguing local de-noising capability of Nice-SLAM** (Zhu et al., 2022) (similar to iMAP) under severe Spatter effect perturbations in RGB imaging. The generated RGB, rendered from the implicit representation, is nearly free of spatter noise.

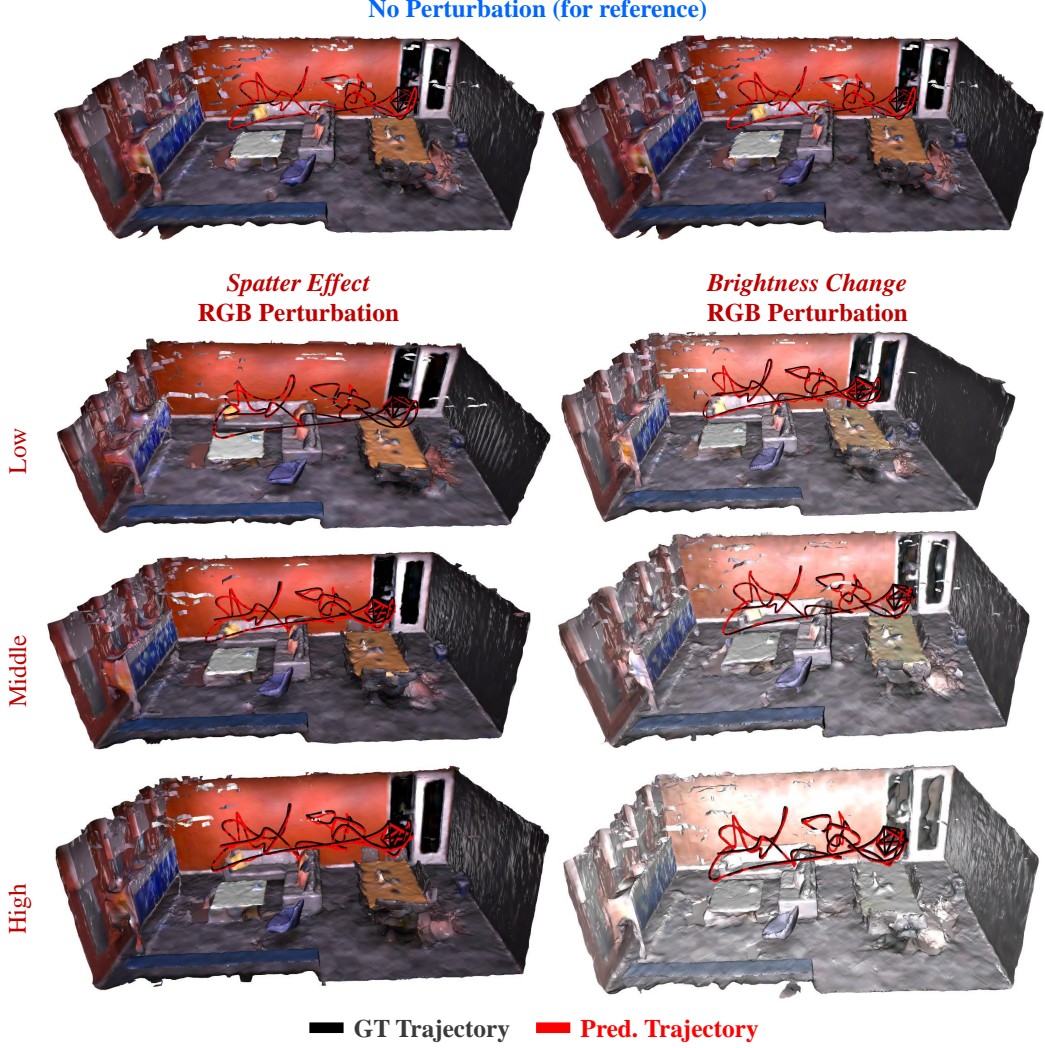

Figure D25: **Local de-noising capability of Nice-SLAM** (Zhu et al., 2022) (similar to iMAP) under different severities of *Spatter Effect* perturbations. The reconstruction is nearly spatter-free (**Left**), but global perturbations like brightness changes (**Right**) cause color shifts in the reconstructed 3D mesh.

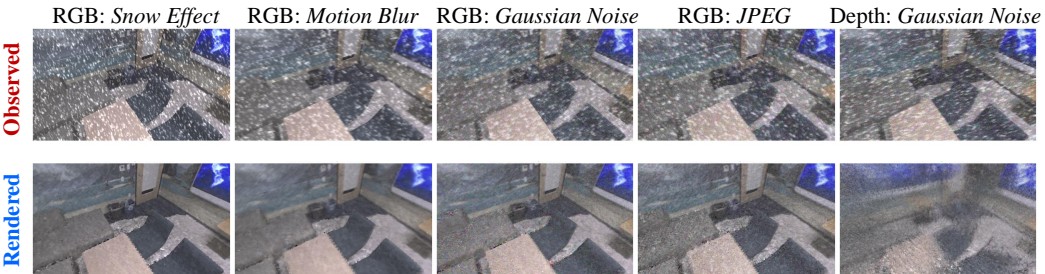

Figure D26: **Intriguing local de-noising capability of SplaTAM-S** (Keetha et al., 2024) under mixed RGB and depth imaging perturbations (composed from left to right). The RGB frames rendered from the explicit neural representation, *i.e.*, 3D Gaussian Splatting (Kerbl et al., 2023), is with less noise than the original noisy observations.

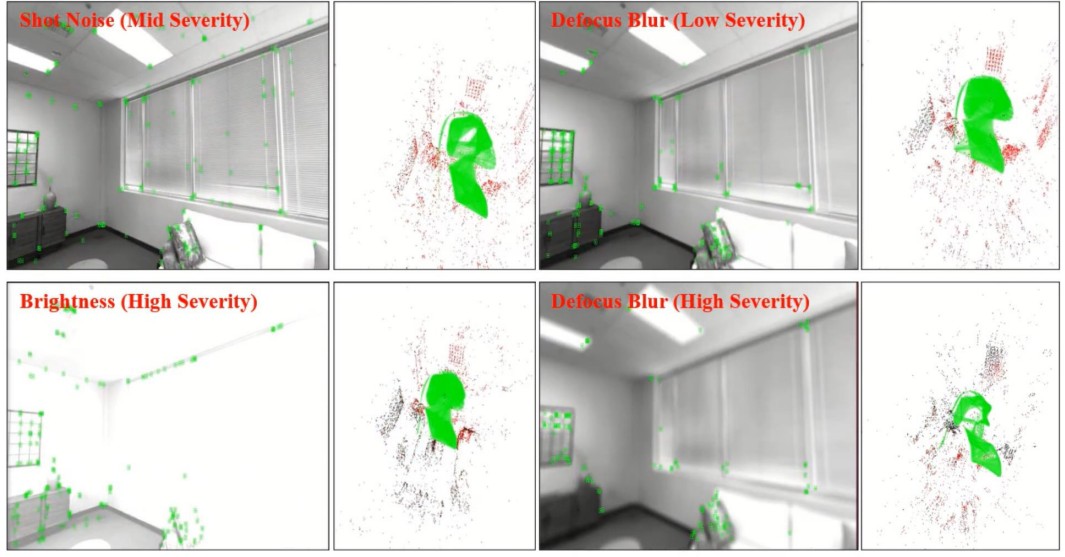

Figure D27: **Qualitative results of the successful cases of ORB-SLAM3 model (Campos et al., 2021) with RGB-D input.**

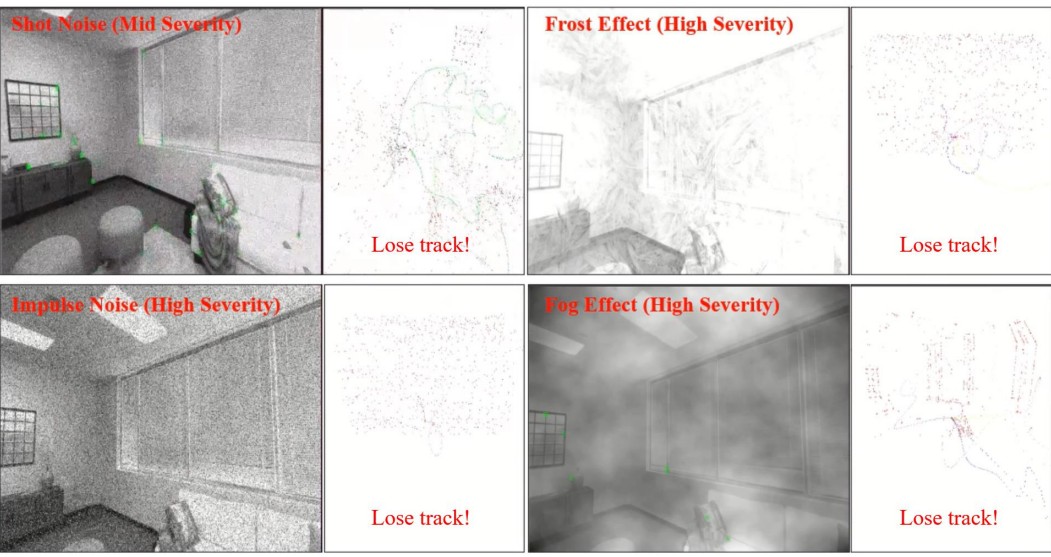

Figure D28: **Sensitivity of ORB features under perturbations in ORB-SLAM3 (Campos et al., 2021),** which can lead to complete tracking loss and failure.

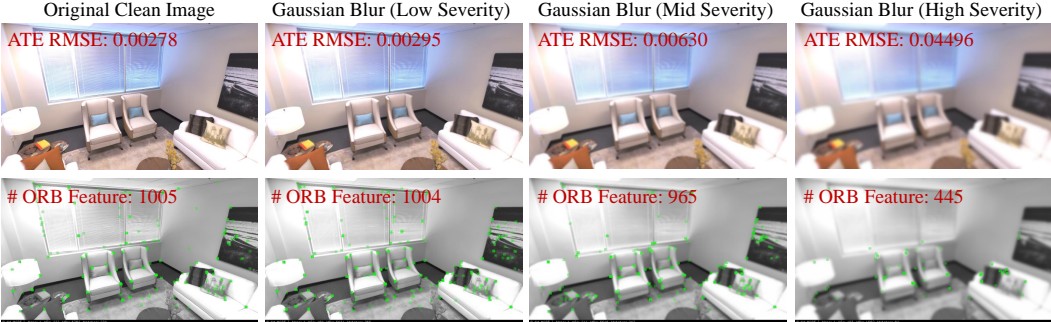

Figure D29: **Effect of *Gaussian Blur* image-level perturbation under different severity (Top) on the quality of detected ORB features (Bottom)**, which are marked as green dots, for the classical SLAM model ORB-SLAM3 (Campos et al., 2021). We report the average trajectory accuracy via ATE RMSE and the average number of ORB features detected in various perturbed settings.

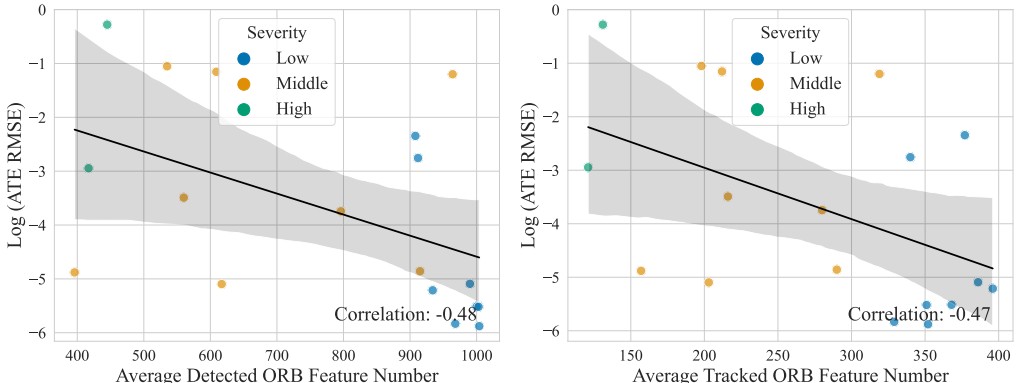

Figure D30: **Correlation between trajectory estimation accuracy and the average number of detected (Left) and tracked (Right) ORB features for ORB-SLAM3** (Campos et al., 2021) model (RGB-D setting) under different severity level of Gaussian Blur image-level perturbation. We report the Pearson correlation coefficient (Cohen et al., 2009) at the bottom right corner. While the correlation coefficient does not indicate a significant linear correlation, there is a noticeable trend of increased trajectory estimation when fewer ORB features are detected or tracked.

# E   THEORETICAL INSIGHTS ON *Robust-Ego3D* BENCHMARKING RESULTS

## E.1   POTENTIAL BENEFITS OF IMAGE RESTORATION ON OPTIMIZATION

We first analyze how restoring noisy RGB images affects the optimization process in differentiable rendering-based SLAM.

**Theorem A.** *Restoring noisy RGB images reduces the variability in the residuals used in differentiable rendering-based SLAM, leading to more stable gradient computations and more efficient convergence.*

*Proof.* Let $\mathcal{P}_t = \{R_t, \mathbf{t}_t\}$ represent the camera pose at time step $t$. The photometric loss for the RGB image is:

$$\mathcal{L}_C(\mathcal{P}_t) = |\hat{\mathbf{I}}_t - \mathbf{I}_t|^2, \tag{E11}$$

where $\hat{\mathbf{I}}_t$ is the rendered image from the map $\mathcal{M}$ given the pose $\mathcal{P}_t$, and $\mathbf{I}_t$ is the observed image.

Assume the observed RGB image is corrupted by additive Gaussian noise:

$$\mathbf{I}_t = \mathbf{I}_t^{\text{true}} + \mathbf{N}_t, \quad \mathbf{N}_t \sim \mathcal{N}(0, \sigma^2 \mathbf{I}), \tag{E12}$$

where $\mathbf{I}_t^{\text{true}}$ is the true image without noise, and $\sigma^2$ is the variance of the noise.

The gradient of the RGB loss with respect to the pose is:

$$\nabla_{\mathcal{P}_t} \mathcal{L}_C = 2(\hat{\mathbf{I}}_t - \mathbf{I}_t)^\top \nabla_{\mathcal{P}_t} \hat{\mathbf{I}}_t. \tag{E13}$$

The noise $\mathbf{N}_t$ affects the residual $(\hat{\mathbf{I}}_t - \mathbf{I}_t)$, introducing variability into the gradient, which can hinder optimization due to the stochastic nature of the noise.

By applying a restoration mechanism to obtain a denoised image $\tilde{\mathbf{I}}_t$, we reduce the noise component:

$$\tilde{\mathbf{I}}_t \approx \mathbf{I}_t^{\text{true}}. \tag{E14}$$

Using the denoised image, the gradient becomes:

$$\nabla \mathcal{P}_t \mathcal{L}_C = 2(\hat{\mathbf{I}}_t - \tilde{\mathbf{I}}_t)^\top \nabla \mathcal{P}_t \hat{\mathbf{I}}_t. \tag{E15}$$

Since $(\hat{\mathbf{I}}_t - \tilde{\mathbf{I}}_t)$ has reduced noise, the gradient is more stable and accurately reflects the discrepancy between the rendered and true images. This leads to more efficient convergence during optimization. □

## E.2   EFFECT OF MISSING AND NOISY DEPTH DATA

Next, we examine how missing and noisy depth data affect the gradient computation and the stability of the optimization.

**Theorem B.** *In differentiable rendering-based optimization for RGB-D SLAM, missing depth data does not introduce noise into the gradient computation and thus does not lead to instability in the optimization process.*

*Proof.* Let $\Omega$ be the set of pixels where depth measurements are available. The depth loss is computed over $\Omega$:

$$\mathcal{L}_D(\mathcal{P}_t) = \frac{1}{|\Omega|} \sum_{i \in \Omega} (\hat{\mathbf{D}}_{t,i} - \mathbf{D}_{t,i})^2, \tag{E16}$$

where $\hat{\mathbf{D}}_{t,i}$ is the rendered depth at pixel $i$, and $\mathbf{D}_{t,i}$ is the observed depth.

The gradient of the depth loss with respect to the pose is:

$$\nabla_{\mathcal{P}_t} \mathcal{L}_D = \frac{2}{|\Omega|} \sum_{i \in \Omega} (\hat{\mathbf{D}}_{t,i} - \mathbf{D}_{t,i}) \nabla_{\mathcal{P}_t} \hat{\mathbf{D}}_{t,i}. \tag{E17}$$

Since missing depth data is excluded from $\Omega$, it does not contribute to the loss or its gradient. Therefore, the absence of depth measurements does not introduce any noise or instability into the gradient computation, allowing the optimization process to remain stable. □

**Theorem C.** *In differentiable rendering-based optimization for RGB-D SLAM, noisy depth inputs introduce variability into the residuals used for gradient computation, which can lead to misaligned pose updates and instability in the optimization process.*

*Proof.* Assume the observed depth map is corrupted by additive Gaussian noise:

$$\mathbf{D}_t = \mathbf{D}_t^{\text{true}} + \mathbf{N}_t, \quad \mathbf{N}_t \sim \mathcal{N}(0, \sigma_D^2 \mathbf{I}), \tag{E18}$$

where $\mathbf{D}_t^{\text{true}}$ is the true depth map, and $\sigma_D^2$ is the variance of the depth noise.

The gradient of the depth loss with respect to the pose is:

$$\nabla_{\mathcal{P}_t} \mathcal{L}_D = 2(\hat{\mathbf{D}}_t - \mathbf{D}_t)^\top \nabla \mathcal{P}_t \hat{\mathbf{D}}_t. \tag{E19}$$

The noise $\mathbf{N}_t$ affects the residual $(\hat{\mathbf{D}}_t - \mathbf{D}_t)$, introducing variability into the gradient. This noisy gradient can misalign pose updates because the optimization is guided by inaccurate depth discrepancies.

When $\lambda_D$ is large, the optimization relies heavily on the depth loss. The variability in the gradient due to noisy depth can lead to incorrect pose updates, causing instability and possibly divergence in the optimization process. $\square$

### E.3 EFFECT OF LARGE POSE CHANGES ON OPTIMIZATION

We analyze how large pose changes impact the efficiency and convergence of gradient-based optimization in differentiable rendering.

**Theorem D.** *Large pose changes in differentiable rendering-based optimization can lead to large residuals and gradients, causing inefficiencies in gradient-based optimization and increasing the risk of divergence.*

*Proof.* Consider a camera moving with a large pose change $\Delta \mathcal{P} = \{\Delta \mathbf{R}, \Delta \mathbf{t}\}$. The rendered images $\hat{\mathbf{I}}$ and $\hat{\mathbf{D}}$ depend on the pose $\mathcal{P}$ and the map $\mathcal{M}$:

$$\hat{\mathbf{I}} = f_{\text{render}}(\mathcal{M}, \mathcal{P}), \quad \hat{\mathbf{D}} = g_{\text{render}}(\mathcal{M}, \mathcal{P}). \tag{E20}$$

With large pose changes, the projections of the Gaussians onto the image plane can change drastically, leading to large discrepancies between the rendered and observed images. This results in large residuals in the loss function:

$$\mathcal{L}(\mathcal{P}) = \lambda_C |\hat{\mathbf{I}} - \mathbf{I}|^2 + \lambda_D |\hat{\mathbf{D}} - \mathbf{D}|^2. \tag{E21}$$

Consequently, the gradients of the loss with respect to the pose parameters become large:

$$\nabla_{\mathcal{P}} \mathcal{L} = 2 \left[ \lambda_C (\hat{\mathbf{I}} - \mathbf{I})^\top \nabla_{\mathcal{P}} \hat{\mathbf{I}} + \lambda_D (\hat{\mathbf{D}} - \mathbf{D})^\top \nabla_{\mathcal{P}} \hat{\mathbf{D}} \right]. \tag{E22}$$

If the optimization uses a fixed learning rate $\eta$, large gradients can cause the pose update step to overshoot the minimum:

$$\mathcal{P}_{k+1} = \mathcal{P}_k - \eta \nabla_{\mathcal{P}} \mathcal{L}(\mathcal{P}_k). \tag{E23}$$

This overshooting can lead to divergence or oscillations in the optimization process. To handle large pose changes efficiently, adaptive step sizes or robust optimization methods (e.g., using line search or second-order methods) are required to ensure convergence. $\square$

### E.4 INVALID LINEAR MOTION ASSUMPTION UNDER COMPLEX MOTION

We now introduce a theorem to support the observation that differentiable rendering optimization can fail to tackle dynamic, non-linear motion due to invalid linear motion assumptions.

**Theorem E.** *Differentiable rendering-based optimization methods that rely on linear motion assumptions can fail to converge under dynamic, non-linear motion due to the inability of linear models to accurately capture higher-order motion dynamics, leading to large residuals and misaligned gradients.*

*Proof.* In differentiable rendering-based SLAM, many methods assume that the camera motion between consecutive frames is approximately linear and smooth. This assumption simplifies the motion model and is reasonable for slow and steady movements. The pose at time $t$ is often predicted using a first-order approximation:

$$\mathcal{P}_t = \mathcal{P}_{t-1} \oplus \Delta\mathcal{P}, \tag{E24}$$

where $\oplus$ denotes the pose composition, and $\Delta\mathcal{P}$ is a small incremental motion estimated under the linear motion assumption.

Under dynamic, non-linear motion, the actual motion $\Delta\mathcal{P}_{\text{true}}$ includes higher-order dynamics such as acceleration and jerk, which are not captured by the linear model:

$$\Delta\mathcal{P}_{\text{true}} = \Delta\mathcal{P}_{\text{linear}} + \Delta\mathcal{P}_{\text{nonlinear}}, \tag{E25}$$

where $\Delta\mathcal{P}_{\text{nonlinear}}$ represents higher-order motion components.

The linear motion assumption leads to an inaccurate initial pose estimate:

$$\mathcal{P}_t^{\text{est}} = \mathcal{P}_{t-1} \oplus \Delta\mathcal{P}_{\text{linear}}. \tag{E26}$$

The discrepancy between the estimated pose $\mathcal{P}_t^{\text{est}}$ and the true pose $\mathcal{P}_t^{\text{true}}$ introduces large residuals in the loss function:

$$\mathcal{L}(\mathcal{P}_t^{\text{est}}) = \lambda_C |\hat{\mathbf{I}}(\mathcal{P}_t^{\text{est}}) - \mathbf{I}_t|^2 + \lambda_D |\hat{\mathbf{D}}(\mathcal{P}_t^{\text{est}}) - \mathbf{D}_t|^2. \tag{E27}$$

These large residuals result in large gradients:

$$\nabla_{\mathcal{P}_t}\mathcal{L} = 2\left[\lambda_C \nabla_{\mathcal{P}_t}\hat{\mathbf{I}}^\top(\hat{\mathbf{I}} - \mathbf{I}_t) + \lambda_D \nabla_{\mathcal{P}_t}\hat{\mathbf{D}}^\top(\hat{\mathbf{D}} - \mathbf{D}_t)\right]. \tag{E28}$$

Due to the significant pose discrepancy, the gradients may point in incorrect directions because the linear approximation does not align with the true motion trajectory. The optimization process, relying on these misaligned gradients, can fail to converge or even diverge.

Therefore, under dynamic, non-linear motion, differentiable rendering-based optimization methods that rely on linear motion assumptions are prone to failure because they cannot accurately capture higher-order motion dynamics, leading to ineffective optimization. $\square$

# F    DETAILED TABLES OF RESULTS ON *Robust-Ego3D* BENCHMARK

To mitigate potential randomness, for each perturbed setting, we conduct each experiment three times on eight 3D scenes (totaling 24 experiments per perturbed result) and report the averaged results. Specifically, for the RGB imaging perturbation, we present the averaged result across three severity levels, under both static and dynamic perturbation modes. This approach reduces the impact of randomness while maintaining computational efficiency, striking a balance between mitigating randomness and ensuring feasibility.

To facilitate future quantitative comparison on our benchmark, we have provided additional detailed benchmarking tables of RGB-D SLAM methods in this section, which includes:

- Neural SLAM methods under depth imaging perturbation (Table F6).
- Neural SLAM methods under the faster motion effect (Table F7).
- Neural SLAM methods under motion deviations (Table F8).
- Neural SLAM methods under RGB-D desynchronization (Table F9).
- ORB-SLAM3 under RGB imaging perturbation (Table F10,Table F11, Table F12).
- ORB-SLAM3 under depth imaging perturbation (Table F13, Table F14, Table F15).
- ORB-SLAM3 under the faster motion effect (Table F16, Table F17, Table F18).
- ORB-SLAM3 under the motion deviations (Table F19, Table F20, Table F21 ).
- ORB-SLAM3 under RGB-D desynchronization (Table F22, Table F23, Table F24).

Furthermore, we have noticed large performance deviations with each perturbed setting for the ORB-SLAM3 model. To reflect potential randomness, we have included the performance standard deviation of this model in our experiment results. We acknowledge these potential performance deviations, which indicate low model robustness – a phenomenon our work aims to highlight to encourage the SLAM community to develop more robust and stable SLAM systems.

Table F6: Performance (ATE [m]) of dense Neural SLAM models under depth imaging perturbation.

| Method | Clean | Gaussian Noise | Edge Erosion | Random Missing | Range Clipping |
|---|---|---|---|---|---|
| iMAP (RGB-D) (Sucar et al., 2021) | 0.1209 | ✗ | 0.0307 | 0.1083 | 0.2438 |
| Nice-SLAM (RGB-D) (Zhu et al., 2022) | 0.0147 | ✗ | 0.0149 | 0.0154 | 0.1183 |
| CO-SLAM (RGB-D) (Wang et al., 2023) | 0.0090 | 0.5794 | 0.0096 | 0.0094 | 0.0122 |
| GO-SLAM (RGB-D) (Zhang et al., 2023) | 0.0046 | 0.0378 | 0.0046 | 0.0046 | 0.0045 |
| SplaTAM-S (RGB-D) (Keetha et al., 2024) | 0.0045 | ✗ | 0.0042 | 0.0042 | 0.0048 |

✗ indicates completely unacceptable, *i.e.*, performance (ATE $\geq$ 1.0 [m])

Table F7: Performance (ATE [m]) of dense Neural SLAM under faster motion.

| Speed-up Ratio | $1\times$ | $2\times$ | $4\times$ | $8\times$ |
|---|---|---|---|---|
| GO-SLAM (Mono) (Zhang et al., 2023) | 0.0039 | 0.0042 | 0.0046 | 0.0048 |
| iMAP (RGB-D) (Sucar et al., 2021) | 0.1209 | 0.4675 | 0.9445 | 1.0000 |
| Nice-SLAM (RGB-D) (Zhu et al., 2022) | 0.0147 | 0.1702 | 1.0000 | 1.0000 |
| CO-SLAM (RGB-D) (Wang et al., 2023) | 0.0090 | 0.1062 | 0.9510 | 1.0000 |
| GO-SLAM (RGB-D) (Zhang et al., 2023) | 0.0046 | 0.0046 | 0.0046 | 0.0050 |
| SplaTAM-S (RGB-D) (Keetha et al., 2024) | 0.0045 | 0.0184 | 1.0000 | 1.0000 |

Table F8: Performance (ATE [m]) of dense Neural SLAM models under motion deviations.

| Rotate [deg] | Clean | 0 | | | 1 | | | | 3 | | | | 5 | | | |
|---|---|---|---|---|---|---|---|---|---|---|---|---|---|---|---|---|
| Translate [m] | | 0.0125 | 0.025 | 0.05 | 0 | 0.0125 | 0.025 | 0.05 | 0 | 0.0125 | 0.025 | 0.05 | 0 | 0.0125 | 0.025 | 0.05 |
| GO-SLAM (Mono) | 0.0039 | 0.0084 | 0.0077 | 0.0091 | 0.0083 | 0.0079 | 0.0082 | 0.0094 | ✗ | ✗ | ✗ | ✗ | ✗ | ✗ | ✗ | ✗ |
| iMAP (RGB-D) | 0.1209 | 0.0334 | 0.1386 | 0.0442 | 0.2438 | 0.2135 | 0.3754 | 0.2801 | ✗ | ✗ | ✗ | ✗ | ✗ | ✗ | ✗ | ✗ |
| Nice-SLAM (RGB-D) | 0.0147 | 0.5812 | ✗ | ✗ | ✗ | ✗ | ✗ | ✗ | ✗ | ✗ | ✗ | ✗ | ✗ | ✗ | ✗ | ✗ |
| CO-SLAM (RGB-D) | 0.0090 | 0.0420 | 0.0848 | 0.3087 | 0.4579 | 0.5069 | 0.2998 | 0.5040 | 0.6443 | 0.6630 | 0.7532 | 0.5772 | 0.8457 | 0.7966 | 0.8277 | ✗ |
| GO-SLAM (RGB-D) | 0.0046 | 0.0082 | 0.0082 | 0.0081 | 0.0080 | 0.0080 | 0.0078 | 0.0077 | ✗ | ✗ | ✗ | ✗ | ✗ | ✗ | ✗ | ✗ |
| SplaTAM-S (RGB-D) | 0.0045 | 0.0545 | 0.0980 | 0.2964 | 0.297$F$ | 0.2272 | 0.2313 | ✗ | ✗ | ✗ | ✗ | ✗ | ✗ | ✗ | ✗ | ✗ |

Notation $F$ represents settings that include failure sequences where no final trajectory is generated due to tracking loss. The number in front of $F$ represents the average ATE as failure sequences are set as a value of 1.0. Notation ✗ indicates completely unacceptable trajectory estimation performance, *i.e.*, ATE $\geq$ 1.0 [m].

Table F9: Performance (ATE [m]) of dense Neural SLAM models under sensor de-synchronization.

| Method | Clean | Static Mode | | | Dynamic Mode | | |
|---|---|---|---|---|---|---|---|
| | $\Delta = 0$ | $\Delta = 5$ | $\Delta = 10$ | $\Delta = 20$ | $\Delta = 5$ | $\Delta = 10$ | $\Delta = 20$ |
| iMAP (RGB-D) (Sucar et al., 2021) | 0.1209 | 0.4672 | 0.5344 | 0.6345 | 0.5104 | 0.6803 | 0.6745 |
| Nice-SLAM (RGB-D) (Zhu et al., 2022) | 0.0147 | 0.3820 | 0.4062 | 0.5216 | 0.5433 | 0.5548 | 0.7020 |
| CO-SLAM (RGB-D) (Wang et al., 2023) | 0.0090 | 0.0520 | 0.1005 | 0.1939 | 0.0740 | 0.1164 | 0.2108 |
| GO-SLAM (RGB-D) (Zhang et al., 2023) | 0.0046 | 0.0148 | 0.0292 | 0.0646 | 0.0151 | 0.0297 | 0.0650 |
| SplaTAM-S (RGB-D) (Keetha et al., 2024) | 0.0045 | 0.0554 | 0.0629 | 0.0880 | 0.0402 | 0.0645 | 0.0850 |

$\Delta$ denotes the misaligned frame interval between RGB and depth streams.

Table F10: ATE [m] metric of ORB-SLAM3 (Campos et al., 2021) under RGB imaging perturbations.

| Perturb. Mode | Input Modality | Clean | Blur Effect | | | | Noise Effect | | | | Environmental Interference | | | | Post-processing Effect | | | |
|---|---|---|---|---|---|---|---|---|---|---|---|---|---|---|---|---|---|---|
| | | | Motion | Defocus | Gaussian | Glass | Gaussian | Shot | Impulse | Speckle | Fog | Frost | Snow | Spatter | Bright | Contra. | JPEG | Pixelate |
| Static | Mono | 0.014 ±0.028 | 0.891 ±0.285 | 0.535 ±0.487 | 0.505 ±0.506 | 0.797 ±0.404 | 0.917 ±0.281 | 0.969 ±0.152 | 1.000 ±0.000 | 0.923 ±0.261 | 0.629 ±0.490 | 0.917 ±0.280 | 1.000 ±0.000 | 0.719 ±0.449 | 0.027 ±0.059 | 0.612 ±0.478 | 0.173 ±0.325 | 0.768 ±0.410 |
| Static | RGB-D | 0.082 ±0.179 | 0.300 ±0.365 | 0.340 ±0.408 | 0.292 ±0.425 | 0.211 ±0.335 | 0.470 ±0.498 | 0.433 ±0.490 | 0.591 ±0.494 | 0.351 ±0.469 | 0.397 ±0.480 | 0.640 ±0.474 | 0.795 ±0.409 | 0.508 ±0.503 | 0.065 ±0.142 | 0.527 ±0.492 | 0.132 ±0.191 | 0.703 ±0.354 |
| Dynamic | Mono | 0.014 ±0.028 | 0.876 ±0.351 | 0.506 ±0.528 | 0.760 ±0.445 | 1.000 ±0.000 | 1.000 ±0.000 | 1.000 ±0.000 | 1.000 ±0.000 | 0.885 ±0.325 | 0.751 ±0.461 | 1.000 ±0.000 | 1.000 ±0.000 | 1.000 ±0.000 | 0.052 ±0.092 | 1.000 ±0.000 | 0.166 ±0.341 | 0.718 ±0.412 |
| Dynamic | RGB-D | 0.082 ±0.179 | 0.279 ±0.330 | 0.303 ±0.435 | 0.052 ±0.080 | 0.168 ±0.245 | 0.508 ±0.526 | 0.513 ±0.520 | 0.755 ±0.453 | 0.262 ±0.455 | 0.659 ±0.476 | 0.256 ±0.459 | 0.876 ±0.352 | 0.753 ±0.458 | 0.066 ±0.145 | 0.751 ±0.462 | 0.645 ±0.491 | 0.756 ±0.294 |

Table F11: RPE [m] metric of ORB-SLAM3 (Campos et al., 2021) under RGB imaging perturbations.

| Perturb. Mode | Input Modality | Clean | Blur Effect | | | | Noise Effect | | | | Environmental Interference | | | | Post-processing Effect | | | |
|---|---|---|---|---|---|---|---|---|---|---|---|---|---|---|---|---|---|---|
| | | | Motion | Defocus | Gaussian | Glass | Gaussian | Shot | Impulse | Speckle | Fog | Frost | Snow | Spatter | Bright | Contra. | JPEG | Pixelate |
| Static | Mono | 0.197 ±0.030 | 0.853 ±0.335 | 0.605 ±0.408 | 0.601 ±0.410 | 0.811 ±0.376 | 0.927 ±0.247 | 0.960 ±0.194 | 0.100 ±0.000 | 0.932 ±0.230 | 0.680 ±0.423 | 0.927 ±0.246 | 1.000 ±0.000 | 0.760 ±0.383 | 0.162 ±0.028 | 0.699 ±0.373 | 0.247 ±0.294 | 0.525 ±0.485 |
| Static | RGB-D | 0.114 ±0.023 | 0.238 ±0.350 | 0.323 ±0.400 | 0.329 ±0.397 | 0.084 ±0.036 | 0.493 ±0.477 | 0.445 ±0.479 | 0.601 ±0.482 | 0.377 ±0.450 | 0.421 ±0.459 | 0.617 ±0.470 | 0.798 ±0.403 | 0.403 ±0.481 | 0.078 ±0.022 | 0.570 ±0.442 | 0.080 ±0.028 | 0.032 ±0.026 |
| Dynamic | Mono | 0.197 ±0.030 | 0.879 ±0.342 | 0.572 ±0.459 | 0.782 ±0.403 | 1.000 ±0.000 | 1.000 ±0.000 | 1.000 ±0.000 | 1.000 ±0.000 | 0.899 ±0.284 | 0.785 ±0.398 | 1.000 ±0.000 | 1.000 ±0.000 | 1.000 ±0.000 | 0.144 ±0.034 | 1.000 ±0.000 | 0.242 ±0.307 | 0.287 ±0.440 |
| Dynamic | RGB-D | 0.114 ±0.023 | 0.043 ±0.024 | 0.293 ±0.437 | 0.033 ±0.027 | 0.046 ±0.021 | 0.517 ±0.517 | 0.523 ±0.511 | 0.755 ±0.453 | 0.284 ±0.443 | 0.642 ±0.494 | 0.272 ±0.450 | 0.876 ±0.350 | 0.758 ±0.448 | 0.074 ±0.016 | 0.792 ±0.393 | 0.656 ±0.474 | 0.031 ±0.011 |

Table F12: Success rate (SR) of ORB-SLAM3 (Campos et al., 2021) under RGB imaging perturbations.

| Perturb. Mode | Input Modality | Clean | Blur Effect | | | | Noise Effect | | | | Environmental Interference | | | | Post-processing Effect | | | |
|---|---|---|---|---|---|---|---|---|---|---|---|---|---|---|---|---|---|---|
| | | | Motion | Defocus | Gaussian | Glass | Gaussian | Shot | Impulse | Speckle | Fog | Frost | Snow | Spatter | Bright | Contra. | JPEG | Pixelate |
| Static | Mono | 0.854 ±0.149 | 0.059 ±0.152 | 0.331 ±0.385 | 0.382 ±0.415 | 0.122 ±0.307 | 0.055 ±0.186 | 0.016 ±0.076 | 0.000 ±0.000 | 0.052 ±0.183 | 0.229 ±0.362 | 0.057 ±0.197 | 0.000 ±0.000 | 0.211 ±0.352 | 0.915 ±0.142 | 0.320 ±0.405 | 0.712 ±0.349 | 0.064 ±0.089 |
| Static | RGB-D | 0.960 ±0.046 | 0.621 ±0.423 | 0.606 ±0.443 | 0.617 ±0.430 | 0.778 ±0.319 | 0.388 ±0.468 | 0.391 ±0.475 | 0.219 ±0.409 | 0.443 ±0.457 | 0.276 ±0.421 | 0.111 ±0.282 | 0.081 ±0.236 | 0.259 ±0.405 | 0.971 ±0.030 | 0.412 ±0.461 | 0.818 ±0.284 | 0.361 ±0.232 |
| Dynamic | Mono | 0.854 ±0.149 | 0.008 ±0.270 | 0.267 ±0.000 | 0.158 ±0.022 | 0.000 ±0.000 | 0.000 ±0.000 | 0.000 ±0.000 | 0.000 ±0.000 | 0.096 ±0.273 | 0.169 ±0.315 | 0.000 ±0.000 | 0.000 ±0.000 | 0.000 ±0.000 | 0.904 ±0.046 | 0.000 ±0.000 | 0.703 ±0.328 | 0.142 ±0.168 |
| Dynamic | RGB-D | 0.960 ±0.046 | 0.871 ±0.364 | 0.354 ±0.429 | 0.301 ±0.437 | 0.991 ±0.081 | 0.207 ±0.378 | 0.277 ±0.473 | 0.040 ±0.084 | 0.303 ±0.441 | 0.027 ±0.045 | 0.005 ±0.009 | 0.000 ±0.000 | 0.009 ±0.026 | 0.975 ±0.027 | 0.003 ±0.007 | 0.376 ±0.519 | 0.432 ±0.218 |

Table F13: ATE metric [m] of ORB-SLAM3 (Campos et al., 2021) with RGB-D input under depth perturbations.

| Clean | Gaussian Noise | Edge Erosion | Random Missing | Range Clipping |
|---|---|---|---|---|
| 0.082 ± 0.179 | 0.803 ± 0.301 | 0.807 ± 0.365 | 0.756 ± 0.397 | 0.994 ± 0.283 |

Table F14: RPE metric [m] of ORB-SLAM3 (Campos et al., 2021) with RGB-D input under depth perturbations.

| Clean | Gaussian Noise | Edge Erosion | Random Missing | Range Clipping |
|---|---|---|---|---|
| 0.114 ± 0.023 | 0.064 ± 0.018 | 0.154 ± 0.342 | 0.054 ± 0.022 | 0.047 ± 0.017 |

Table F15: Success rate (SR) of ORB-SLAM3 (Campos et al., 2021) with RGB-D input under depth perturbations.

| Clean | Gaussian Noise | Edge Erosion | Random Missing | Range Clipping |
|---|---|---|---|---|
| 0.960 ± 0.046 | 0.421 ± 0.331 | 0.379 ± 0.281 | 0.322 ± 0.354 | 0.286 ± 0.181 |

Table F16: ATE metric [m] of ORB-SLAM3 (Campos et al., 2021) under fast motion.

| Speed-up Ratio | 1× | 2× | 4× | 8× |
|---|---|---|---|---|
| Monocular | 0.014 ± 0.028 | 0.009 ± 0.009 | 0.023 ± 0.051 | 0.023 ± 0.053 |
| RGB-D | 0.082 ± 0.179 | 0.019 ± 0.023 | 0.064 ± 0.156 | 0.077 ± 0.125 |

Table F17: RPE metric [m] of ORB-SLAM3 (Campos et al., 2021) under fast motion.

| Speed-up Ratio | 1× | 2× | 4× | 8× |
|---|---|---|---|---|
| Monocular | 0.197 ± 0.030 | 0.194 ± 0.035 | 0.227 ± 0.035 | 0.289 ± 0.064 |
| RGB-D | 0.114 ± 0.023 | 0.152 ± 0.032 | 0.190 ± 0.030 | 0.268 ± 0.059 |

Table F18: Success rate (SR) of ORB-SLAM3 (Campos et al., 2021) under fast motion.

| Speed-up Ratio | 1× | 2× | 4× | 8× |
|---|---|---|---|---|
| Monocular | 0.854 ± 0.149 | 0.893 ± 0.081 | 0.909 ± 0.049 | 0.837 ± 0.115 |
| RGB-D | 0.960 ± 0.046 | 0.964 ± 0.012 | 0.938 ± 0.029 | 0.866 ± 0.129 |

Table F19: ATE metric [m] of ORB-SLAM3 (Campos et al., 2021) under motion deviations.

| Rotate [deg] | Clean | 0 | | | 1 | | | | 3 | | | | 5 | | | |
|---|---|---|---|---|---|---|---|---|---|---|---|---|---|---|---|---|
| Translate [m] | | 0.0125 | 0.025 | 0.05 | 0 | 0.0125 | 0.025 | 0.05 | 0 | 0.0125 | 0.025 | 0.05 | 0 | 0.0125 | 0.025 | 0.05 |
| Monocular | 0.014 ±0.028 | 0.017 ±0.027 | 0.136 ±0.349 | 0.190 ±0.339 | 0.063 ±0.047 | 0.146 ±0.196 | 0.348 ±0.378 | 0.200 ±0.355 | 0.170 ±0.338 | 0.055 ±0.057 | 0.053 ±0.061 | 0.061 ±0.061 | 0.073 ±0.135 | 0.023 ±0.039 | 0.041 ±0.042 | 0.080 ±0.108 |
| RGB-D | 0.082 ±0.179 | 0.191 ±0.369 | 0.085 ±0.174 | 0.171 ±0.337 | 0.059 ±0.067 | 0.163 ±0.280 | 0.084 ±0.052 | 0.228 ±0.361 | 0.148 ±0.210 | 0.090 ±0.072 | 0.158 ±0.106 | 0.085 ±0.037 | 0.164 ±0.340 | 0.072 ±0.100 | 0.050 ±0.054 | 0.062 ±0.076 |

Table F20: RPE metric [m] of ORB-SLAM3 (Campos et al., 2021) under motion deviations.

| Rotate [deg] | Clean | 0 | | | 1 | | | | 3 | | | | 5 | | | |
|---|---|---|---|---|---|---|---|---|---|---|---|---|---|---|---|---|
| Translate [m] | | 0.0125 | 0.025 | 0.05 | 0 | 0.0125 | 0.025 | 0.05 | 0 | 0.0125 | 0.025 | 0.05 | 0 | 0.0125 | 0.025 | 0.05 |
| Monocular | 0.197 ±0.030 | 0.191 ±0.037 | 0.291 ±0.289 | 0.297 ±0.297 | 0.182 ±0.071 | 0.194 ±0.036 | 0.291 ±0.294 | 0.197 ±0.075 | 0.349 ±0.270 | 0.231 ±0.086 | 0.265 ±0.105 | 0.235 ±0.087 | 0.355 ±0.052 | 0.324 ±0.083 | 0.368 ±0.067 | 0.371 ±0.040 |
| RGB-D | 0.114 ±0.023 | 0.229 ±0.313 | 0.119 ±0.028 | 0.208 ±0.322 | 0.114 ±0.039 | 0.116 ±0.038 | 0.153 ±0.046 | 0.231 ±0.313 | 0.197 ±0.046 | 0.220 ±0.027 | 0.203 ±0.035 | 0.220 ±0.026 | 0.417 ±0.239 | 0.313 ±0.047 | 0.298 ±0.052 | 0.334 ±0.042 |

Table F21: Success rate (SR) of ORB-SLAM3 (Campos et al., 2021) under motion deviations.

| Rotate [deg] | Clean | 0 | | | 1 | | | | 3 | | | | 5 | | | |
|---|---|---|---|---|---|---|---|---|---|---|---|---|---|---|---|---|
| Translate [m] | | 0.0125 | 0.025 | 0.05 | 0 | 0.0125 | 0.025 | 0.05 | 0 | 0.0125 | 0.025 | 0.05 | 0 | 0.0125 | 0.025 | 0.05 |
| Monocular | 0.854 ±0.149 | 0.489 ±0.107 | 0.247 ±0.129 | 0.128 ±0.091 | 0.662 ±0.329 | 0.340 ±0.189 | 0.221 ±0.156 | 0.120 ±0.069 | 0.373 ±0.329 | 0.096 ±0.154 | 0.087 ±0.081 | 0.059 ±0.056 | 0.413 ±0.418 | 0.155 ±0.160 | 0.214 ±0.112 | 0.144 ±0.039 |
| RGB-D | 0.960 ±0.046 | 0.462 ±0.213 | 0.321 ±0.085 | 0.152 ±0.100 | 0.596 ±0.491 | 0.345 ±0.218 | 0.228 ±0.151 | 0.114 ±0.115 | 0.325 ±0.270 | 0.259 ±0.226 | 0.146 ±0.121 | 0.118 ±0.087 | 0.422 ±0.428 | 0.158 ±0.156 | 0.180 ±0.148 | 0.129 ±0.092 |

Table F22: ATE metric [m] of ORB-SLAM3 (Campos et al., 2021) with RGB-D input under desynchronization.

| Perturb Mode | Clean | Misaligned Frame Interval ($\Delta$) | | |
|---|---|---|---|---|
| | $\Delta = 0$ | $\Delta = 5$ | $\Delta = 10$ | $\Delta = 20$ |
| Static | $0.082 \pm 0.179$ | $0.069 \pm 0.168$ | $0.066 \pm 0.154$ | $0.065 \pm 0.163$ |
| Dynamic | | $0.070 \pm 0.157$ | $0.077 \pm 0.161$ | $0.083 \pm 0.178$ |

Table F23: RPE metric [m] of ORB-SLAM3 (Campos et al., 2021) with RGB-D input under desynchronization.

| Perturb Mode | Clean | Misaligned Frame Interval ($\Delta$) | | |
|---|---|---|---|---|
| | $\Delta = 0$ | $\Delta = 5$ | $\Delta = 10$ | $\Delta = 20$ |
| Static | $0.114 \pm 0.023$ | $0.123 \pm 0.024$ | $0.114 \pm 0.024$ | $0.115 \pm 0.023$ |
| Dynamic | | $0.116 \pm 0.025$ | $0.112 \pm 0.019$ | $0.117 \pm 0.027$ |

Table F24: Success rate (SR) of ORB-SLAM3 (Campos et al., 2021) with RGB-D input under desynchronization.

| Perturb Mode | Clean | Misaligned Frame Interval ($\Delta$) | | |
|---|---|---|---|---|
| | $\Delta = 0$ | $\Delta = 5$ | $\Delta = 10$ | $\Delta = 20$ |
| Static | $0.960 \pm 0.046$ | $0.960 \pm 0.036$ | $0.958 \pm 0.029$ | $0.954 \pm 0.030$ |
| Dynamic | | $0.955 \pm 0.039$ | $0.942 \pm 0.050$ | $0.948 \pm 0.041$ |

## G  MORE DETAILS ABOUT CORRESPONDENCE-GUIDED GAUSSIAN SPLATTING (CORRGS)

### G.1  SCHEMATIC PIPELINE OVERVIEW AND PSEUDOCODE

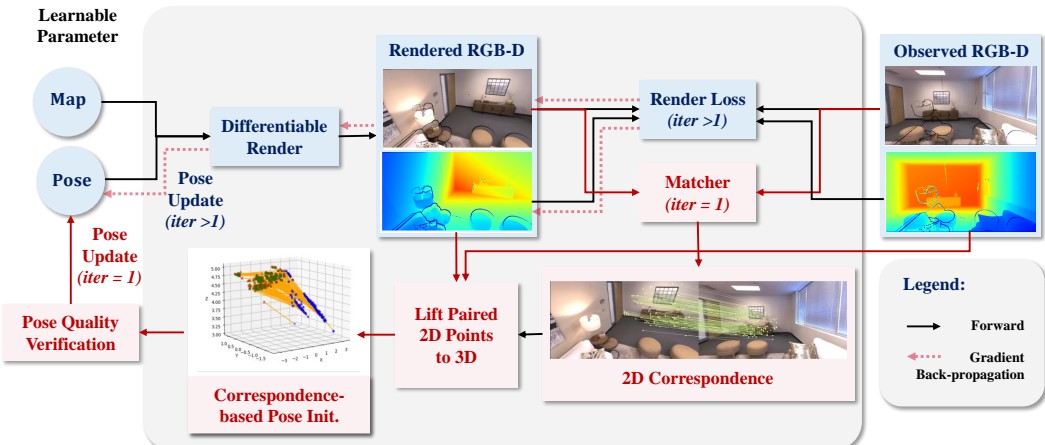

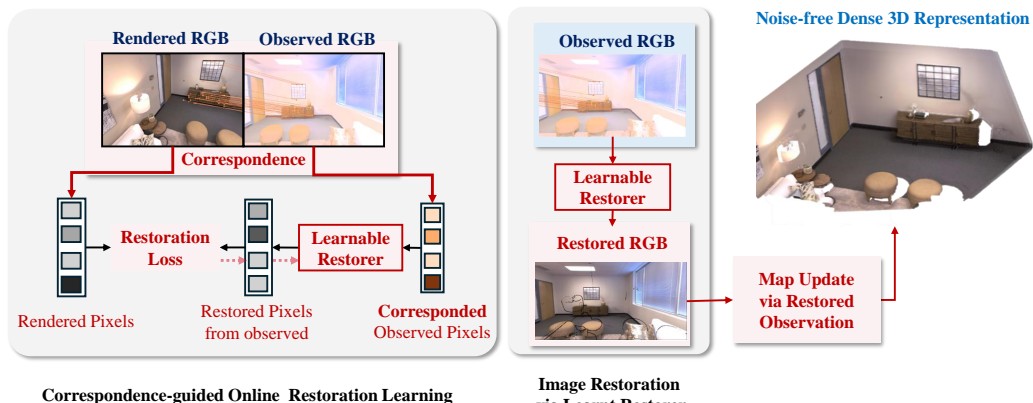

Figure G31: **Main components of Correspondence-guided Gaussian Splatting (CorrGS).** (**a**) Correspondence-guided Pose Learning (CPL) for robust tracking. (**b**) Correspondence-guided Appearance Restoration Learning (CARL) for noise-free and consistent 3D reconstruction.

---

**Algorithm 1** CorrGS: Correspondence-Guided Gaussian Splatting SLAM System(pseudocode)

---

**Require:** Noisy RGB image $\mathbf{I}_t$, depth map $\mathbf{D}_t$, camera intrinsics $\mathbf{K}$, historical poses $\mathcal{P}_{<t}$, historical map $\mathcal{M}_{<t}$ (Gaussian Splat representation)

**Ensure:** Refined camera pose $\mathcal{P}_t$, updated map $\mathcal{M}_t$

1: **1: Naive Pose Initialization**
2: Initialize camera pose $\mathcal{P}_t$ by propagating from historical poses $\mathcal{P}_{<t}$.
3: **2: Correspondence-based Pose Initialization**
4: Render synthesized RGB-D images $\hat{\mathbf{I}}_t$, $\hat{\mathbf{D}}_t$ using initial pose $\mathcal{P}_t$ and historical map $\mathcal{M}_{<t}$.
5: Compute 2D correspondences between $\hat{\mathbf{I}}_t$ and observed image $\mathbf{I}_t$.
6: Lift 2D correspondences to 3D correspondences using $\mathbf{D}_t$, $\hat{\mathbf{D}}_t$, and $\mathbf{K}$.
7: Update pose $\mathcal{P}_t^c$ via non-linear optimization over 3D correspondences using prior pose $\mathcal{P}_t$ as initialization.
8: **3: Correspondence-based Pose Quality Verification**
9: Compute rendering losses for poses $\mathcal{P}_t^c$ and $\mathcal{P}_t$.
10: Select the pose with the lower loss for further processing.
11: **4: Correspondence-guided Appearance Restoration Learning**
12: Learn restoration model $f(\cdot, \theta)$ using the color data from the correspondences.
13: Generate restored RGB frame $\mathbf{I}_t^R \leftarrow f(\mathbf{I}_t, \theta)$.
14: Update observed image $\mathbf{I}_t \leftarrow \mathbf{I}_t^R$.
15: **Repeat Steps 2–4 for 1 iteration for further refinement.**
16: **5: Differentiable Rendering-based Pose Refinement**
17: Refine pose $\mathcal{P}_t$ using differentiable rendering-based optimization.
18: **6: Map Update**
19: Integrate (restored) observed RGB image $\mathbf{I}_t$, depth map $\mathbf{D}_t$, and refined pose $\mathcal{P}_t$ into the historical map $\mathcal{M}_{<t}$ to produce the updated 3D map $\mathcal{M}_t$.
20: **return** Refined pose $\mathcal{P}_t$, updated noise-free map $\mathcal{M}_t$.

---

## G.2 PRELIMINARIES

**Gaussian map representation.** We represent the 3D environment using a set of oriented 3D Gaussians, capturing both geometry and appearance. Each Gaussian $\mathbf{G}_i$ is parameterized by its position $\mathbf{X}_i \in \mathbb{R}^3$, a covariance matrix $\Sigma_i \in \mathbb{R}^{3 \times 3}$ (which we simplify to isotropic Gaussians), opacity $\Lambda_i \in [0, 1]$, and RGB color $\mathbf{c}_i \in \mathbb{R}^3$. The simplified map $\mathcal{M}$ is defined as:

$$\mathcal{M} = \{\mathbf{G}_i : (\mathbf{X}_i, \Sigma_i, \Lambda_i, \mathbf{c}_i) \,|\, i = 1, \ldots, N\}. \tag{G29}$$

Each Gaussian $\mathbf{G}_i$ contributes to a point in 3D space $\mathbf{x} \in \mathbb{R}^3$ according to the Gaussian function weighted by its opacity:

$$f_i^{rend}(\mathbf{x}) = \Lambda_i \exp\left(-\frac{|\mathbf{x} - \mathbf{X}_i|^2}{2\mathbf{r}_i^2}\right), \tag{G30}$$

where $\mathbf{r}_i$ is the isotropic Gaussian radius. This representation supports efficient and differentiable rendering of the scene.

**Differentiable rendering-based pose optimization.** Our approach enables differentiable rendering of color, depth, and silhouette images from the Gaussian Splat Map into any camera frame. By leveraging this differentiable process, we compute gradients with respect to both the scene representation $\mathcal{M}$ (the Gaussians) and the camera pose $\mathcal{P} = (\mathbf{R}, \mathbf{t})$, where $\mathbf{R} \in SO(3)$ is the rotation matrix and $\mathbf{t} \in \mathbb{R}^3$ is the translation vector. The rendered images can be compared to input RGB-D frames to minimize the error and refine both the scene and camera parameters.

Rendering an RGB image follows the process of Gaussian Splatting (Kerbl et al., 2023): given a collection of Gaussians, we first sort them from front to back and then render the image by alpha-compositing the splatted 2D projections of each Gaussian. The color of a pixel $\mathbf{p} = (\mathbf{u}, \mathbf{v})$ is computed as:

$$\mathbf{I}(\mathbf{p}) = \sum_{i=1}^{N} \mathbf{c}_i f_i^{rend}(\mathbf{p}) \prod_{j=1}^{i-1} \left(1 - f_j^{rend}(\mathbf{p})\right), \tag{G31}$$

where $\mathbf{f}_i(\mathbf{p})$ is computed by projecting the 3D Gaussians into pixel space:

$$\mathbf{X}_i^{2D} = \mathbf{K}\frac{\mathcal{P}_t \mathbf{X}_i}{\mathbf{d}_i}, \quad \mathbf{r}_i^{2D} = \frac{\mathbf{f}\mathbf{r}_i}{\mathbf{d}_i}, \quad \mathbf{d}_i = (\mathcal{P}_t \mathbf{X}_i)_z. \tag{G32}$$

Here, $\mathbf{K}$ is the known camera intrinsic matrix, $\mathcal{P}_t$ is the estimated pose, *i.e.*, the camera extrinsic matrix for frame $t$, $\mathbf{f}$ is the focal length, and $\mathbf{d}_i$ is the depth of the $i^{th}$ Gaussian primitive.

Similarly, we render the depth image:

$$\mathbf{D}(\mathbf{p}) = \sum_{i=1}^{N} \mathbf{d}_i f_i^{rend}(\mathbf{p}) \prod_{j=1}^{i-1} \left(1 - f_j^{rend}(\mathbf{p})\right), \tag{G33}$$

These rendered images enable the optimization of the camera pose ($\mathcal{P}$) by minimizing a loss function $\mathcal{L}(\mathcal{P})$:

$$\mathcal{L}(\mathcal{P}) = \lambda_C \mathcal{L}_C(\hat{\mathbf{I}}, \mathbf{I}) + \lambda_D \mathcal{L}_D(\hat{\mathbf{D}}, \mathbf{D}), \tag{G34}$$

where $\lambda_C, \lambda_D > 0$ are weighting parameters for the photometric loss terms $\mathcal{L}_C$ and $\mathcal{L}_D$ for color and depth, respectively.

The pose is iteratively updated using gradient descent:

$$\mathcal{P}_{k+1} = \mathcal{P}_k - \eta \nabla_{\mathcal{P}} \mathcal{L}(\mathcal{P}_k), \tag{G35}$$

where $\eta > 0$ is the learning rate.

Rendered RGB    Observed RGB

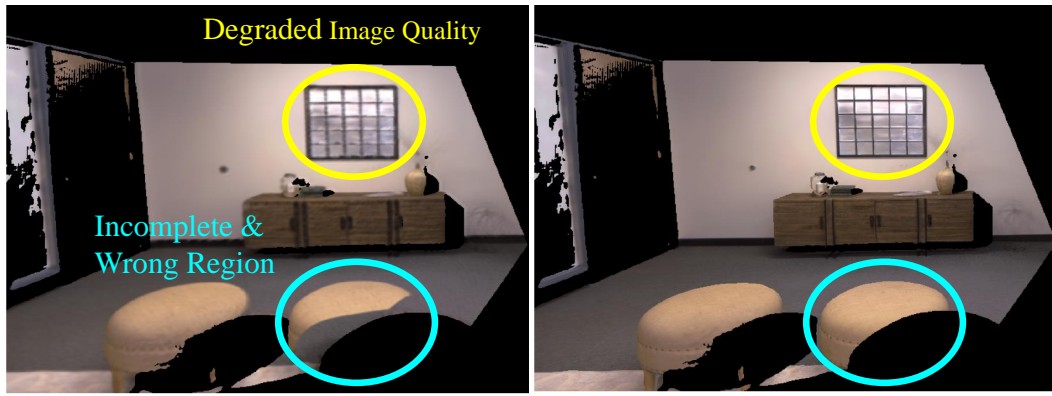

Figure G32: Comparison of rendered and observed image quality. The rendered image exhibits degradation due to pose noise, impacting correspondence calculation.

### G.3 CORRESPONDENCE-GUIDED POSE LEARNING (CPL)

#### G.3.1 NAIVE POSE INITIALIZATION

Pose learning starts by propagating the camera pose from historical poses $\mathcal{P}_{<t}$ as the naive initialization $\mathcal{P}_t$. Synthesized RGB-D images $\hat{\mathbf{I}}_t$ and $\hat{\mathbf{D}}_t$ are then rendered using $\mathcal{P}_t$ and the historical map $\mathcal{M}_{<t}$. These rendered images are used to establish correspondences with the observed RGB-D data ($\mathbf{I}_t$ and $\mathbf{D}_t$), refining the pose for more accurate tracking.

#### G.3.2 2D CORRESPONDENCE CALCULATION

We utilize an off-the-shelf feature matcher (e.g., LoFTR (Sun et al., 2021)) to compute 2D correspondences between the rendered $\hat{\mathbf{I}}_t$ and the observed RGB images. These correspondences consist of pairs of 2D pixel coordinates, $(\mathbf{u}_r, \mathbf{v}_r)$ for the rendered image and $(\mathbf{u}_o, \mathbf{v}_o)$ for the observed image. However, due to feature matching inaccuracies, such as degraded quality in the rendered image (see Fig. G32), errors $\epsilon_{2D}$ arise:

$$(\mathbf{u}_r, \mathbf{v}_r) = (\mathbf{u}_r^*, \mathbf{v}_r^*) + \epsilon_{2D,r}, \quad (\mathbf{u}_o, \mathbf{v}_o) = (\mathbf{u}_o^*, \mathbf{v}_o^*) + \epsilon_{2D,o}, \tag{G36}$$

where $(\mathbf{u}_r^*, \mathbf{v}_r^*)$ and $(\mathbf{u}_o^*, \mathbf{v}_o^*)$ represent the true correspondences, and $\epsilon_{2D,r}$, $\epsilon_{2D,o}$ are the 2D matching errors for the rendered and observed images, respectively.

#### G.3.3 LIFTING 2D CORRESPONDENCES TO 3D

Given the observed depth map $\mathbf{D}_o$ and rendered depth map $\mathbf{D}_r$, we lift the corresponding 2D pixel coordinates into 3D points via:

$$\mathbf{p}_r = \mathbf{D}_r(\mathbf{u}_r, \mathbf{v}_r)\mathbf{K}^{-1} \begin{pmatrix} \mathbf{u}_r \\ \mathbf{v}_r \\ 1 \end{pmatrix}, \tag{G37}$$

$$\mathbf{p}_o = \mathbf{D}_o(\mathbf{u}_o, \mathbf{v}_o)\mathbf{K}^{-1} \begin{pmatrix} \mathbf{u}_o \\ \mathbf{v}_o \\ 1 \end{pmatrix}, \tag{G38}$$

where $\mathbf{K}$ is the camera intrinsics matrix, and $\mathbf{p}_r$ and $\mathbf{p}_o$ are the corresponding 3D points in the rendered and observed point clouds, respectively.

Due to errors in depth measurements $\epsilon_{D,r}$ and $\epsilon_{D,o}$, the 3D points become perturbed:

$$\mathbf{p}_r = (\mathbf{D}_r^* + \epsilon_{D,r})\mathbf{K}^{-1} \begin{pmatrix} \mathbf{u}_r^* + \epsilon_{2D,r} \\ \mathbf{v}_r^* + \epsilon_{2D,r} \\ 1 \end{pmatrix}, \tag{G39}$$

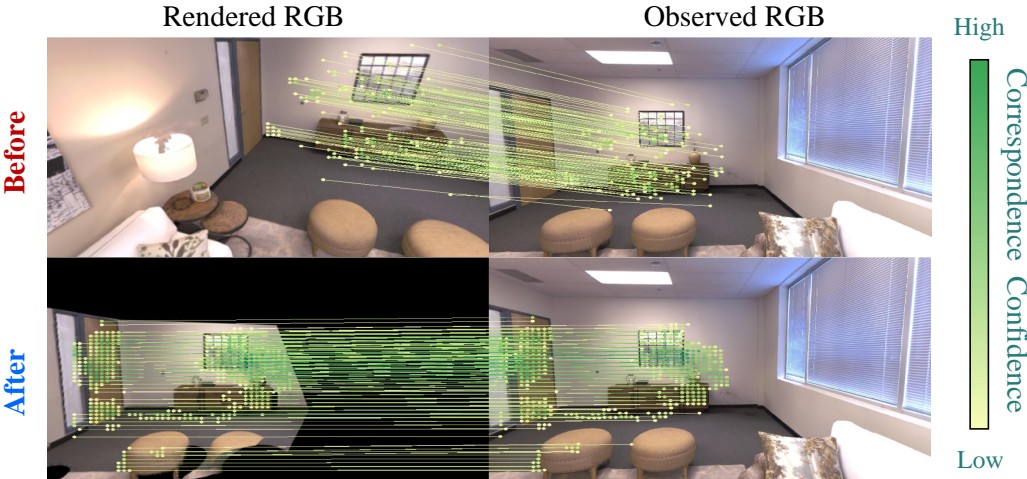

Figure G33: Effect of one-step pose initialization via correspondence.

$$\mathbf{p}_o = (\mathbf{D}_o^* + \epsilon_{D,o})\mathbf{K}^{-1} \begin{pmatrix} u_o^* + \epsilon_{2D,o} \\ v_o^* + \epsilon_{2D,o} \\ 1 \end{pmatrix}, \tag{G40}$$

where $\mathbf{D}_r^*$ and $\mathbf{D}_o^*$ are the true depths, and $\epsilon_{D,r}$ and $\epsilon_{D,o}$ are the depth errors.

### G.3.4 CORRESPONDENCE-BASED POSE INITIALIZATION

To estimate the relative pose transformation between the two corresponded 3D point clouds, *i.e.*, the rendered points $\mathbf{p}_r$ and the observed points $\mathbf{p}_o$, we leverage the correspondences to align them. By minimizing the reprojection error between the two sets of points, we iteratively estimate the relative pose transformation. This transformation consists of a rotation matrix $\mathbf{R}_{rel}$ and a translation vector $\mathbf{t}_{rel}$, which describe how the rendered point cloud should be aligned with the observed one. Once the relative transformation is computed, it is used to update the camera's pose estimation in the current frame, refining the pose incrementally based on the previous pose estimation.

**Relative transformation calculation.** We begin by solving the following non-linear least squares problem to iteratively refine the relative pose:

$$\min_{\mathbf{R}_{rel},\mathbf{t}_{rel}} \sum_{i=1}^{N} \rho \left( \|\mathbf{R}_{rel}\mathbf{p}_{r,i} + \mathbf{t}_{rel} - \mathbf{p}_{o,i}\|^2 \right), \tag{G41}$$

where $\mathbf{R}_{rel} \in \mathbf{SO}(3)$ is the relative rotation matrix, $\mathbf{t}_{rel} \in \mathbb{R}^3$ is the relative translation vector, $\|\cdot\|$ denotes the Euclidean norm, and $\rho(\cdot)$ is the `soft_l1` loss function. The optimization iteratively refines these parameters to estimate the relative pose between the two point clouds.

Fig. G33 demonstrates that, even in the case of a substantial view change, the image rendered after one step of pose initialization via correspondence closely matches the observed image.

The `soft_l1` loss function is defined as:

$$\rho(s) = 2 \left( \sqrt{1 + \frac{s}{2}} - 1 \right), \tag{G42}$$

where $s = \|\mathbf{r}_i\|^2$ is the squared residual. The smooth L1 loss behaves quadratically for small residuals and transitions to linear behavior for larger residuals, reducing the influence of outliers.

At the start of the optimization, the relative pose is initialized as the identity transformation, meaning no initial rotation or translation is applied. We then iteratively refine the pose by solving for $\mathbf{R}_{rel}$ and $\mathbf{t}_{rel}$ that minimize the reprojection error. The residuals at each iteration are computed as:

$$\mathbf{r}_i = \mathbf{R}_{rel}\mathbf{p}_{r,i} + \mathbf{t}_{rel} - \mathbf{p}_{o,i}, \tag{G43}$$

where $\mathbf{r}_i$ is the residual for the $i$-th correspondence.

The Jacobian of the residuals, $\mathbf{J}(\mathbf{R}_{rel}, \mathbf{t}_{rel})$, is approximated using the '3-point' finite-difference method. This involves numerically perturbing $\mathbf{R}_{rel}$ and $\mathbf{t}_{rel}$ and measuring the corresponding changes in the residuals:

$$\mathbf{J}(\mathbf{R}_{rel}, \mathbf{t}_{rel}) = \frac{\partial}{\partial(\mathbf{R}_{rel}, \mathbf{t}_{rel})} \mathbf{r}_i. \tag{G44}$$

The '3-point' method improves the accuracy of Jacobian estimation, leading to better convergence without the need for explicit analytical derivatives.

Starting with the identity transformation ($\mathbf{R}_{rel} = \mathbf{I}, \mathbf{t}_{rel} = \mathbf{0}$), the optimizer iteratively updates the pose parameters to minimize the total reprojection error. The residuals and Jacobian are recalculated at each step until the pose converges to the optimal estimate. This ensures a robust and accurate relative pose estimate, even in the presence of noisy or imperfect correspondences.

**Pose update after relative transformation calculation.** After estimating the relative pose $(\mathbf{R}_{rel}, \mathbf{t}_{rel})$, we update the current estimated pose $\mathcal{P}_t^c$ by applying this relative transformation to the previous pose estimation $\mathcal{P}_t$. The updated pose is computed as:

$$\mathcal{P}_t^c = \mathcal{P}_t \cdot \begin{pmatrix} \mathbf{R}_{rel} & \mathbf{t}_{rel} \\ \mathbf{0}^\top & 1 \end{pmatrix}. \tag{G45}$$

### G.3.5 COMPUTATIONAL COMPLEXITY OF CORRESPONDENCE-GUIDED INITIALIZATION

Let $m$ denote the number of correspondences, which is bounded by the image resolution, i.e., $m = h \times w$, where $h$ and $w$ represent the image height and width, respectively. The number of pose parameters, $n$, is fixed at 16, corresponding to the elements of a $4 \times 4$ transformation matrix.

The computational complexity per iteration is primarily determined by two factors: evaluating the residual function and approximating the Jacobian. The total complexity can be expressed as:

$$\mathcal{O}(m \times n) + \mathcal{O}(n^3). \tag{G46}$$

Substituting $m = h \times w$ and $n = 16$, this becomes:

$$\mathcal{O}(16 \times h \times w + 16^3). \tag{G47}$$

Since $h \times w \gg 16^3$, the dominant term simplifies to:

$$\mathcal{O}(h \times w). \tag{G48}$$

Thus, the computational complexity of the correspondence-guided pose initialization is primarily linear with respect to the image resolution.

This method is implemented using the `least_squares` function from `scipy` library (Virtanen et al., 2020). For a $1200 \times 680 \times 3$ image, the average processing time is approximately 10ms on our server, demonstrating the method's efficiency even at higher resolutions. The computational complexity scales with image resolution, ensuring efficient convergence in pose estimation.

### G.3.6 THEORETICAL ANALYSIS IN SMALL AND LARGE MOTION CASES OF CORRESPONDENCE-BASED POSE INITIALIZATION

**Theoretical analysis on the small motion case**. When the relative motion between the rendered and observed images is small, i.e., $\mathbf{R} \approx I$ and $\mathbf{t} \approx 0$, the optimization problem becomes highly sensitive to small perturbations in $\mathbf{R}$ and $\mathbf{t}$. The transformed 3D points can be approximated as:

$$\mathbf{R}\mathbf{p}_{r,i} + \mathbf{t} \approx \mathbf{p}_{r,i} + \Delta\mathbf{R}\mathbf{p}_{r,i} + \Delta\mathbf{t}, \tag{G49}$$

where $\Delta\mathbf{R}$ and $\Delta\mathbf{t}$ are small deviations from the true pose. The residual error then becomes:

$$\|\Delta\mathbf{R}\mathbf{p}_{r,i} + \Delta\mathbf{t}\|^2. \tag{G50}$$

This residual error is further influenced by the errors in 2D correspondences $\epsilon_{2D}$ and depth measurements $\epsilon_D$ (c.f., Eq. (G39) and Eq. (G40)) The resulting residual error, incorporating these perturbations, is:

$$\|\Delta\mathbf{R}(\mathbf{p}_{r,i} + \epsilon_{p,r}) + \Delta\mathbf{t} - (\mathbf{p}_{o,i} + \epsilon_{p,o})\|^2, \tag{G51}$$

where $\epsilon_{p,r}$ and $\epsilon_{p,o}$ encapsulate the combined effects of both 2D correspondence and depth errors.

Thus, in the small motion case, the condition number of this optimization problem increases, making it highly sensitive to these small errors.

**Theoretical analysis on the large motion case**. When there is significant relative motion between the rendered and observed images, the optimization process benefits from larger gradients, making it more stable and resistant to noise. The transformed points under large motion can be expressed as:

$$\mathbf{R}\mathbf{p}_{r,i} + \mathbf{t} = \mathbf{p}_{r,i}^{\text{Trans}} + \epsilon_{p,r}^{\text{Trans}}, \tag{G52}$$

where $\mathbf{p}_{r,i}^{\text{Trans}}$ represents the true transformed points, and $\epsilon_{p,r}^{\text{Trans}}$ includes the 2D correspondence and depth errors after transformation.

The residual error is given by:

$$\|\mathbf{p}_{r,i}^{\text{Trans}} + \epsilon_{p,r}^{\text{Trans}} - (\mathbf{p}_{o,i} + \epsilon_{p,o})\|^2. \tag{G53}$$

In the case of large motion, the magnitude of the gradients (derived from the residual error with respect to pose parameters) increases, providing more pronounced and clearer directionality during optimization. This stabilizes the optimization process by making it less susceptible to small perturbations in the input data. Thus, larger motion enhances the robustness of the optimization, leading to more reliable convergence.

### G.3.7 RENDERING-BASED POSE QUALITY VERIFICATION

To address the potential failure of correspondence-based pose initialization in small motion scenarios (see Fig. G34), we reject the pose $\mathcal{P}_t^c$ estimated from CPL if it produces a higher loss than the naively propagated pose $\mathcal{P}_t$, which is estimated using a linear motion model based on historical poses. We render RGB-D images $\hat{\mathbf{I}}_{\text{CPL},t}, \hat{\mathbf{D}}_{\text{CPL},t}$ from the correspondence-guided initialization and $\hat{\mathbf{I}}_{\text{naive},t}, \hat{\mathbf{D}}_{\text{naive},t}$ from the naively propagated pose. The losses are then computed as:

$$\mathcal{L}_{\text{CPL},t}(\mathcal{P}_t^c) = \lambda_C \mathcal{L}_C(\hat{\mathbf{I}}_{\text{CPL},t}, \mathbf{I}_t) + \lambda_D \mathcal{L}_D(\hat{\mathbf{D}}_{\text{CPL},t}, \mathbf{D}_t), \tag{G54}$$

$$\mathcal{L}_{\text{naive},t}(\mathcal{P}_t) = \lambda_C \mathcal{L}_C(\hat{\mathbf{I}}_{\text{naive},t}, \mathbf{I}_t) + \lambda_D \mathcal{L}_D(\hat{\mathbf{D}}_{\text{naive},t}, \mathbf{D}_t), \tag{G55}$$

After computing the losses for the CPL pose $\mathcal{P}_t^c$ and the naively propagated pose $\mathcal{P}_t$, we select the pose with the lower loss for further refinement. The selection criterion is defined as:

$$\mathcal{P}_t^* = \arg\min_{\mathcal{P}_t^c, \mathcal{P}_t} \left( \mathcal{L}_{\text{CPL},t}(\mathcal{P}_t^c), \mathcal{L}_{\text{naive},t}(\mathcal{P}_t) \right) \tag{G56}$$

### G.4 DIFFERENTIABLE RENDERING-BASED POSE OPTIMIZATION WITH THE CORRESPONDENCE-GUIDED POSE INITIALIZATION

After verifying the quality of the initial pose $\mathcal{P}_t^*$, it is refined through differentiable rendering-based pose optimization (see Sec. G.2). The synergy between the correspondence-guided initialization and differentiable optimization enables the system to leverage both precise correspondences and photometric consistency during optimization. This two-step process improves the robustness of the final pose estimate by first aligning point correspondences and then using the differentiable rendering to capture finer details and reduce ambiguities. As a result, this combination ensures more robust pose learning, thus enhancing the accuracy of pose tracking in the dense Neural SLAM system.

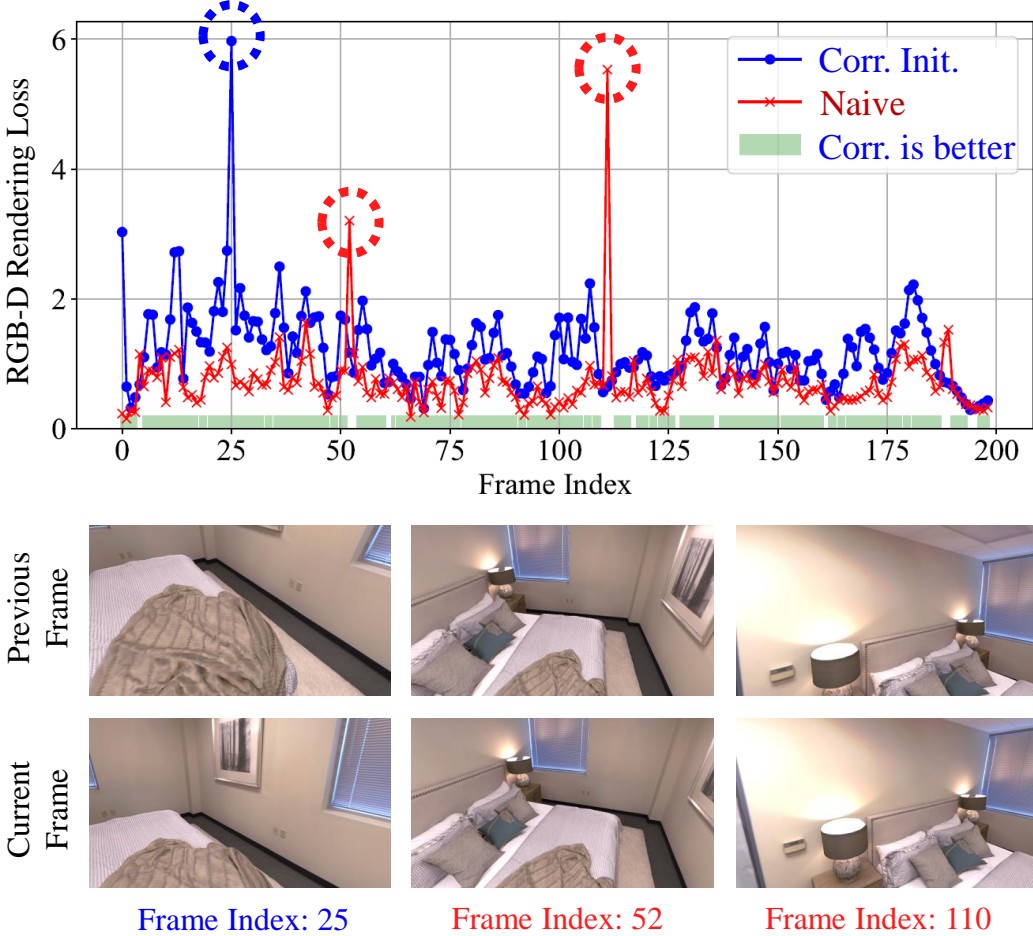

Figure G34: A pose initialized using correspondence in CPL demonstrates superior performance in scenarios involving rapid motion, effectively maintaining alignment and stability. However, it may fail to achieve similar accuracy under conditions of small motion, where the lack of distinct changes makes correspondence-based initialization less reliable.

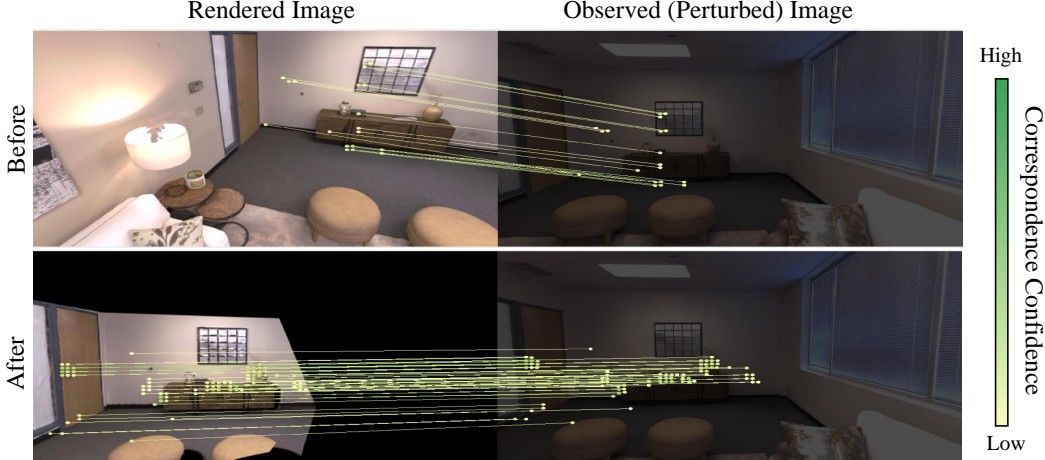

Figure G35: **Existing matchers (Sun et al., 2021) can produce weak but usable correspondences in cross-domain settings** (e.g., varying illumination), which can be leveraged for correspondence-guided pose learning to improve pose estimation accuracy. As shown in the bottom figure, the rendered image from the estimated pose is well-aligned with the observation, compared to the top figure where the initial pose results in a rendered image with significant view misalignment.

## G.5 Correspondence-guided Appearance Restoration Learning (CARL)

### G.5.1 Motivation and Insights

We find that existing matchers can generate weak but usable correspondences in noisy cross-domain settings, such as varying illumination. These correspondences can be leveraged for correspondence-guided pose learning to improve pose estimation accuracy. For instance, as illustrated in Fig. G35, the rendered image from the estimated pose aligns well with the observed image (bottom figure), compared to the large misalignment seen with the initial pose (top figure).

Building on this, we further utilize these weak correspondences for appearance restoration. By training a restoration model, $f(\cdot; \theta)$, CARL maps noisy colors from the observed images to clean counterparts derived from a progressively-updated noise-free historical map. This restored image is then used to refine the camera pose, iteratively improving both image quality and pose estimation, ensuring robustness in noisy, cross-domain environments.

### G.5.2 Restoration Model Learning

The restoration model $f(\cdot; \theta)$ is learned online, using 2D correspondences obtained from the pose initialization step. The goal is to minimize the discrepancy between the noisy colors $\mathbf{C}_{o,i}$ from the noisy observed points and the clean colors $\mathbf{C}_{r,i}$ rendered from the historical map:

$$\theta^* = \arg\min_{\theta} \sum_{i=1}^{N} \|f(\mathbf{C}_{o,i}; \theta) - \mathbf{C}_{r,i}\|^2 . \tag{G57}$$

This optimization process ensures that the restored colors closely match the clean appearance which is maintained in the progressively-updated m3D map, providing an improved and noise-reduced version of the current frame.

### G.5.3 Restoring Perturbed Observations and Improving Tracking via CARL

**CARL restores (perturbed) observations for noise-free mapping.** Once the restoration model $f(\cdot, \theta)$ is trained, it is applied to the entire image to reduce noise and restore the appearance of the observed scene. Let $\mathbf{I}_t$ be the observed RGB image at time $t$, which may contain noise or degradation, and $\mathbf{I}_t^R$ be the restored image. The restoration model is applied pixel-wise across the entire image:

$$\mathbf{I}_t^R = f(\mathbf{I}_t; \theta) \tag{G58}$$

The restored image $\mathbf{I}_t^R$, depth map $\mathbf{D}_t$, and refined pose $\mathcal{P}_t$ are integrated into the historical map $\mathcal{M}_{<t}$ to update the 3D map $\mathcal{M}_t$, ensuring consistency.

**CARL enables more robust pose tracking optimization.** CARL also enhances tracking robustness by improving pose estimation through a second-round of correspondence calculation and pose optimization. Once the image has been restored, the corresponding can be recomputed. And then, the updated observed 3D points $\mathbf{p}_{o,i}^R$ are then used in a second-iteration of pose optimization. The objective function for the second-round relative pose refinement (with the restored points) is:

$$\min_{\mathbf{R}_{rel}, \mathbf{t}_{rel}} \sum_{i=1}^{N} \rho \left( \left\| \mathbf{R}_{rel} \mathbf{p}_{r,i} + \mathbf{t}_{rel} - \mathbf{p}_{o,i}^R \right\|^2 \right), \tag{G59}$$

By iteratively refining the pose using the restored 3D points, CARL ensures more robust tracking, especially in challenging environments where noise or dynamic elements are present.

### G.6 REACTIVE STRATEGY FOR HANDLING CORRESPONDENCE CALCULATION FAILURES

In real-world cases where an incoming frame is severely perturbed and the correspondence calculation fails, the system bypasses CPL and switches to the conventional differentiable rendering-based pose optimization for pose estimation. Additionally, CARL is disabled for these frames to avoid degradation in the restoration process. These ***problematic frames are also excluded from the map update stage to prevent error propagation*** into the 3D representation, ensuring the overall accuracy and integrity of both pose estimation and the maintained map in the dense Neural SLAM system.

# H MORE QUALITATIVE RESULTS ON CORRGS

## H.1 MORE QUALITATIVE RESULTS ON SYNTHETIC (CLEAN) SPARSE-VIEW VIDEO

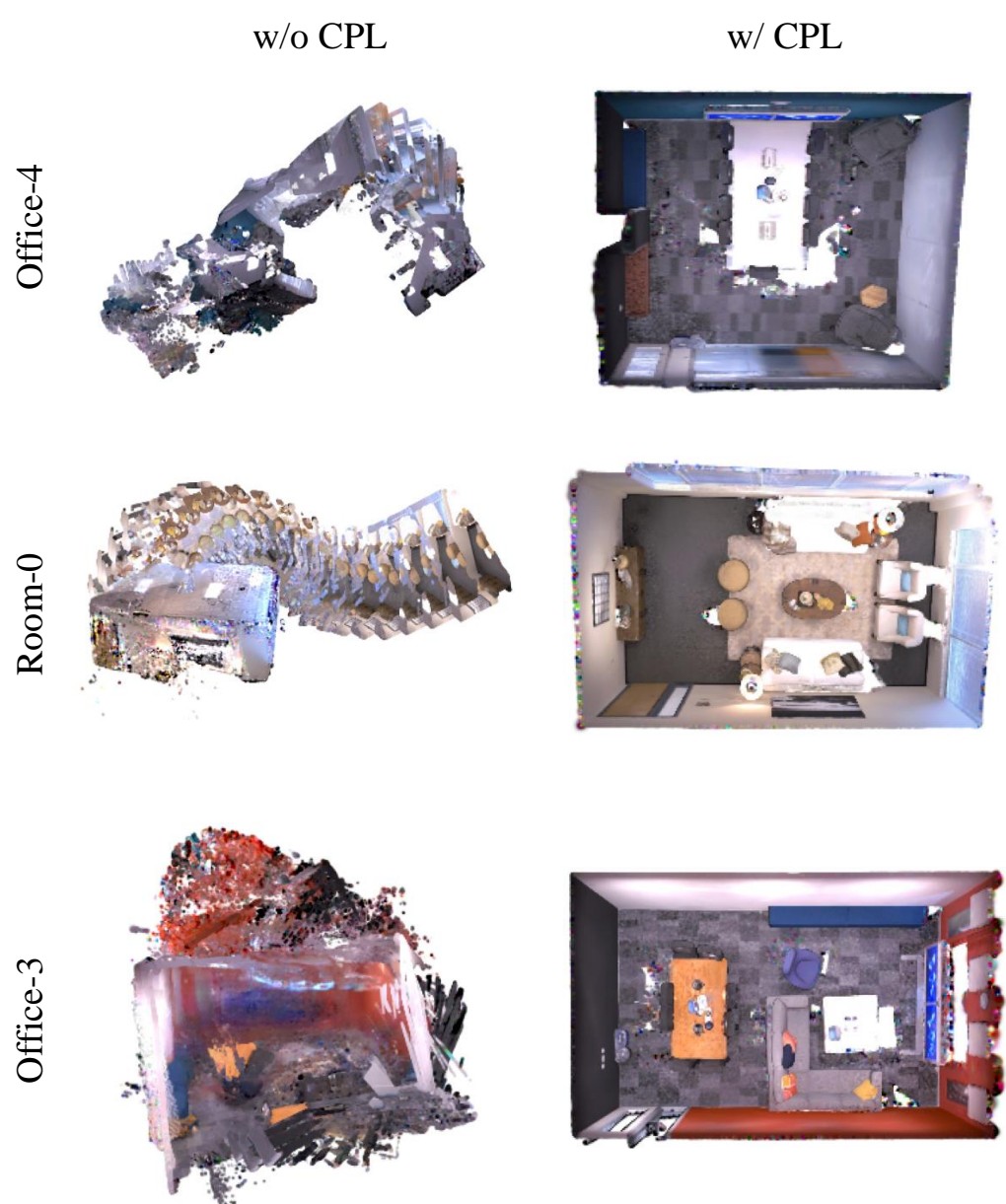

Figure H36: Qualitative ablation study of Correspondence-guided Pose Learning (CPL) on 3D reconstruction with sparse-view (clean) videos. CPL ensures robust pose tracking, enabling high-quality 3D reconstruction even under extreme fast motion.

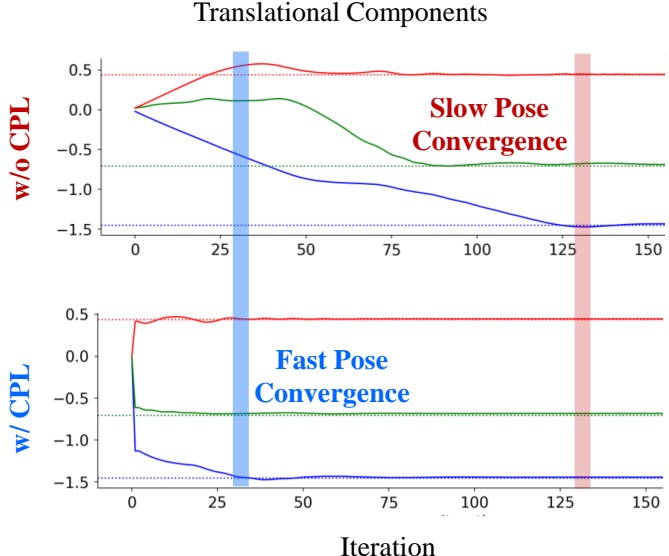

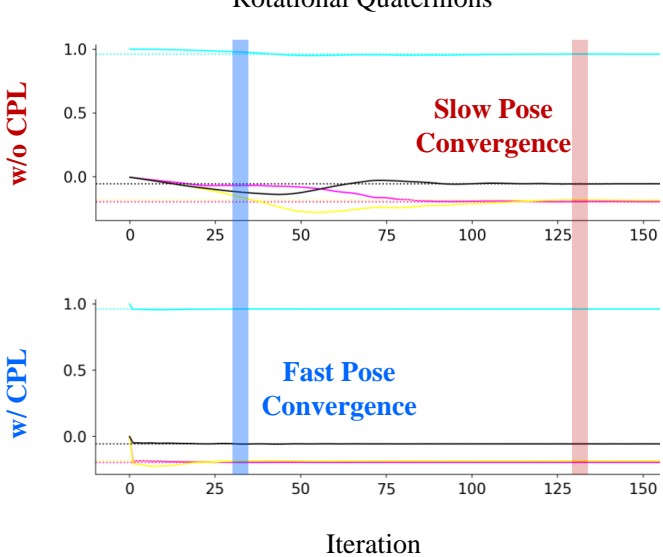

Figure H37: CPL facilitates faster convergence of pose optimization, significantly reducing the number of iterations required to align the estimated pose with the ground-truth. Dotted lines represent the ground-truth poses, while solid lines denote the estimated poses throughout the optimization.

## H.2 MORE QUALITATIVE RESULTS ON SYNTHETIC NOISY SPARSE-VIEW VIDEO

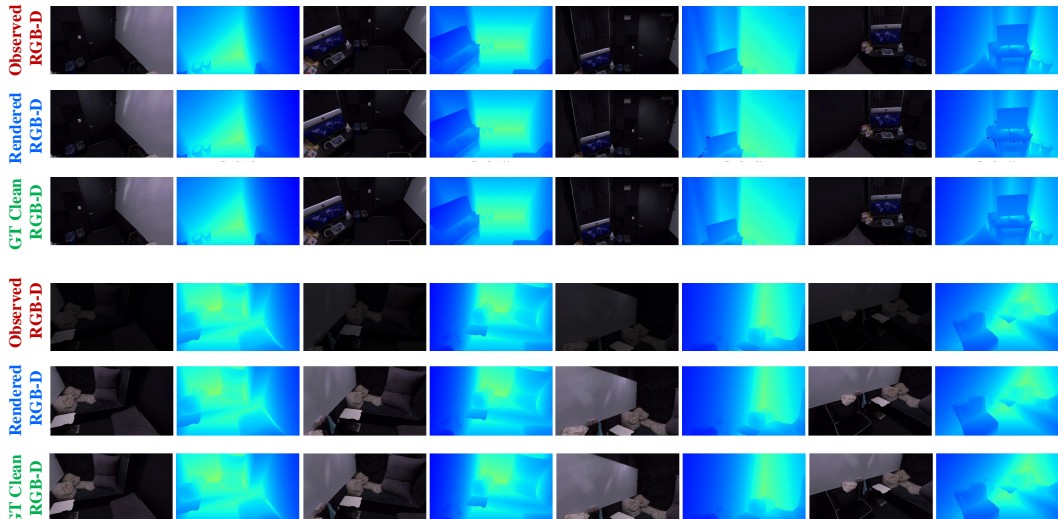

Figure H38: Comparison of noise-free RGB and depth frames rendered using CorrGS with observed noisy RGB and depth frames (scene *Office-1*). Ground-truth clean RGB-D frames are included for reference, demonstrating qualitative results on synthetic data under dynamic illumination changes.

**Noise-free Dense 3D Reconstruction (Baseline)**

**Noise-free Dense 3D Reconstruction (CorrGS, Ours)**

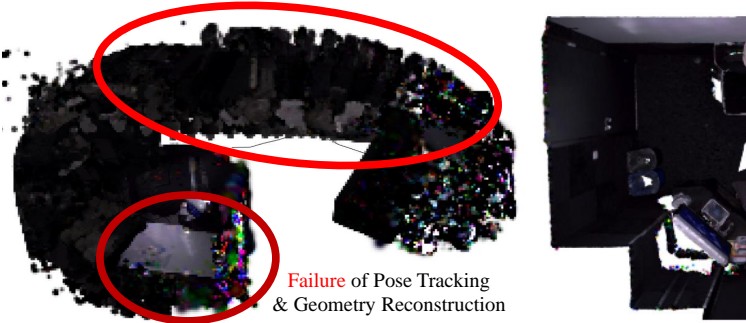

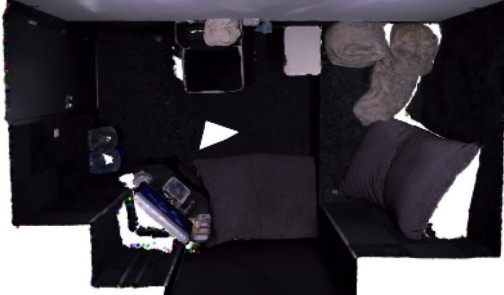

Figure H39: Qualitative comparison of photorealistic 3D maps constructed using CorrGS and the baseline on synthetic, noisy, sparse-view data (scene *Office-1*). The baseline struggles to accurately reconstruct both geometry and appearance, whereas our method produces a clean 3D representation.

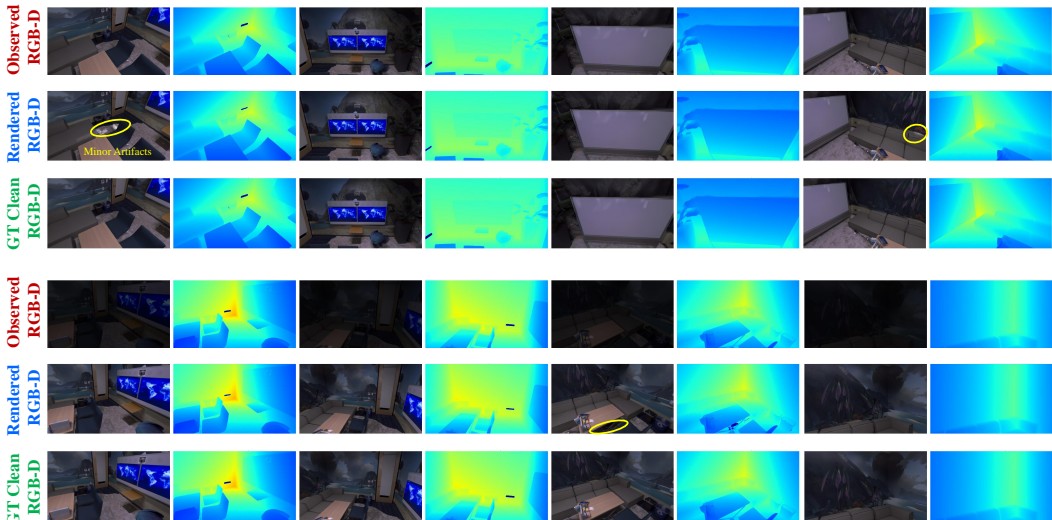

Figure H40: Comparison of noise-free RGB and depth frames rendered using CorrGS with observed noisy RGB and depth frames (scene *Office-0*). Ground-truth clean RGB-D frames are included for reference, demonstrating qualitative results on synthetic data under dynamic illumination changes. While there are some small local artifacts, the overall reconstruction and restoration quality is promising.

**Noise-free Dense 3D Reconstruction (Baseline)**  **Noise-free Dense 3D Reconstruction (CorrGS, Ours)**

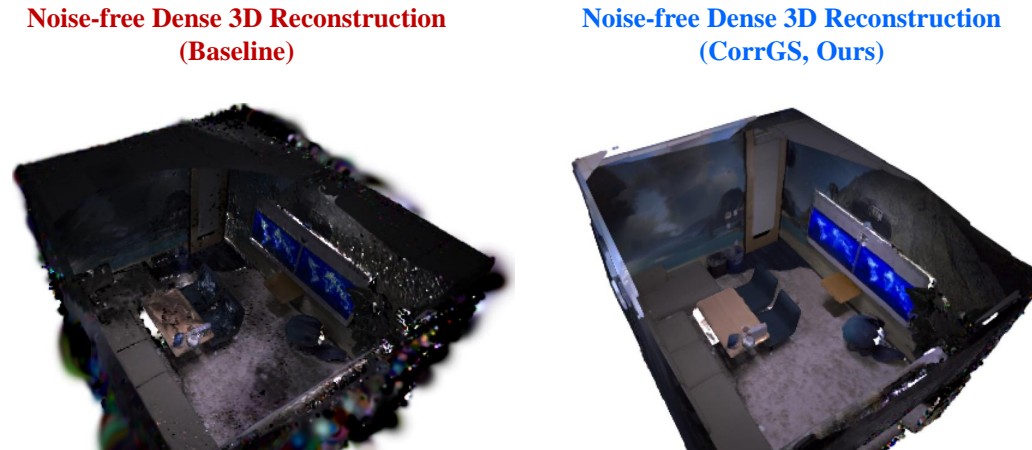

Figure H41: Qualitative comparison of photorealistic 3D maps constructed using CorrGS and the baseline on synthetic, noisy, sparse-view data (scene *Office-0*). The baseline struggles to accurately reconstruct both geometry and appearance, whereas our method produces a clean 3D representation.

## H.3 MORE QUALITATIVE RESULTS ON REAL-WORLD NOISY SPARSE-VIEW VIDEO

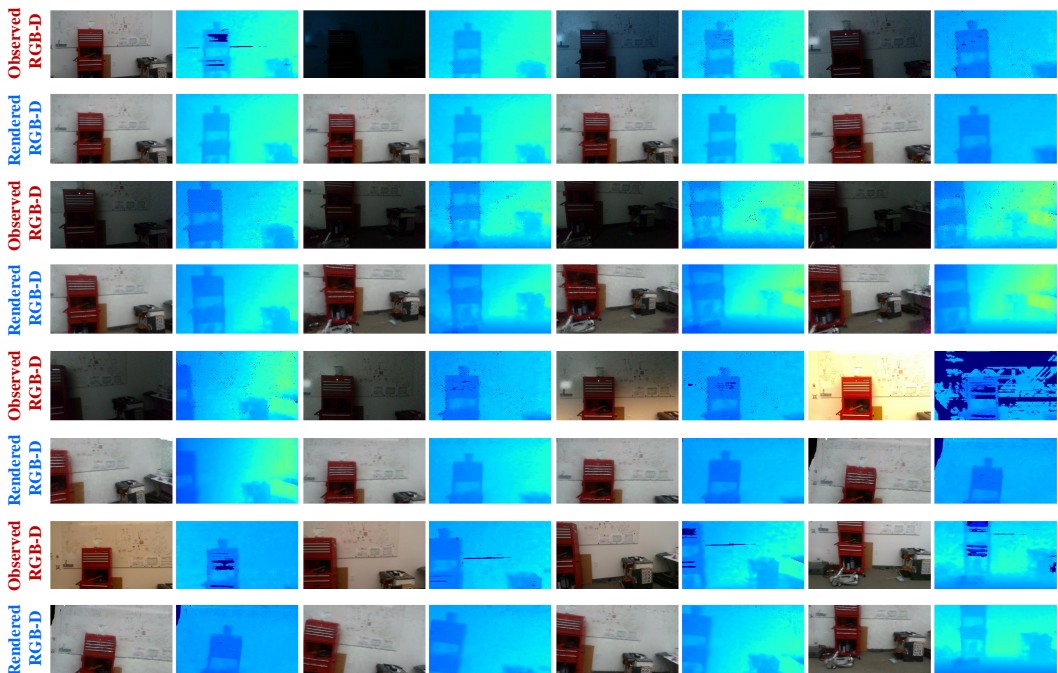

Figure H42: Comparison of the rendered noise-free RGB and depth frames (via CorrGS) with the observed noisy RGB and depth, showcasing qualitative results on real-world data with dynamic illumination changes. The sequences involve fast-motion, sparse-view scenarios, transitioning between normal light, darkness, and overexposure, demonstrating our method's robustness in reconstructing noise-free, dense 3D maps under challenging conditions.

## Noise-free Photorealistic 3D Reconstruction

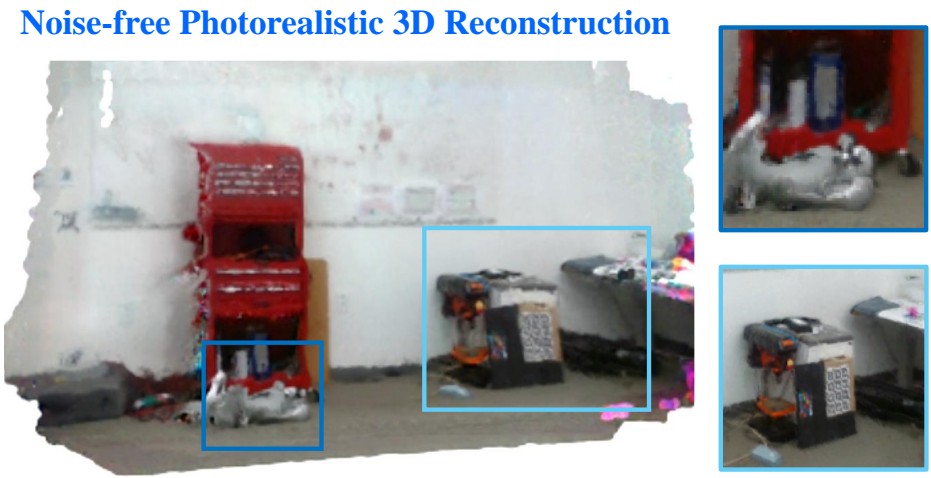

Figure H43: Qualitative results of constructed noise-free photorealistic 3D map via CorrGS on real-world noisy sparse-view data.

**(a) Observed Depth Distribution of Each Frame (Min, Mean, Max Depth)**

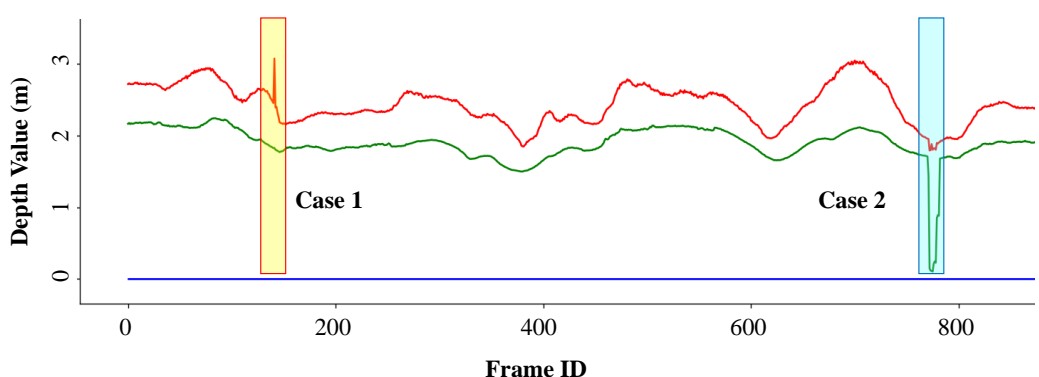

**(b) Degraded Rendering due to Depth Degradation**

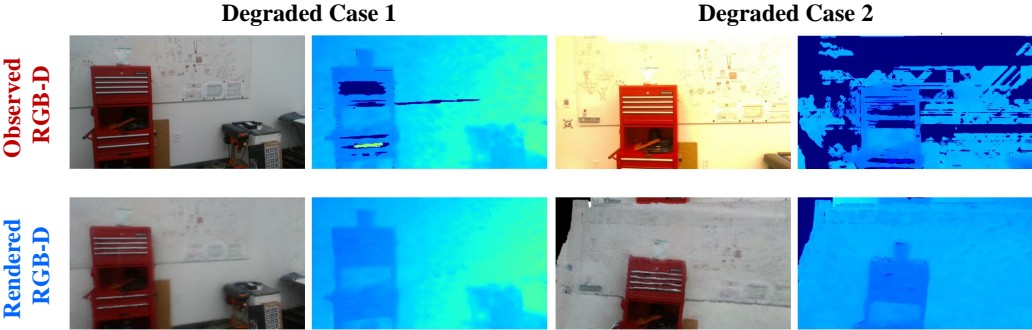

Figure H44: **Failure cases of degraded rendering** by CorrGS on real-world noisy, sparse-view data. The highlighted regions correspond to two failure cases (Case 1 and Case 2) where noisy depth data leads to degraded RGB rendering in specific frames. Addressing depth restoration remains a valuable future direction.

## H.4    ROBUSTNESS OF CORRGS TO POSE INITIALIZATION ERRORS

To evaluate the robustness of CorrGS against pose initialization errors, we tested it on 100 overlapping RGB-D image pairs from each of three room scenes in our dataset, which features diverse poses. The proposed method demonstrates robust performance against significant pose initialization errors, maintaining accuracy under angular discrepancies of up to 40° and translational offsets of up to 1.0 m. Evaluations across three distinct scenes—room0, room1, and room2—highlight its resilience to both rotational and translational noise, as illustrated in Figures H45 and H46.

Fig. H45 shows that PQV maintains low rotation error across all scenes, even as absolute rotation increases. Room0 exhibits the lowest error with minimal degradation, reflecting the framework's adaptability in simpler environments. While room1 and room2 show slightly higher errors due to more complex spatial arrangements, performance remains stable and consistent, emphasizing PQV's robustness to angular noise across varying environmental complexities.

Fig. H46 illustrates PQV's response to translational noise, where translation error remains small and gradually increases with larger offsets. Room0 again achieves the lowest error, demonstrating its effectiveness in less complex conditions. In contrast, room2 exhibits higher variance, likely due to increased structural or geometric challenges. Despite this, PQV consistently delivers stable performance without significant deviations, confirming its ability to handle varying translational conditions across diverse environments.

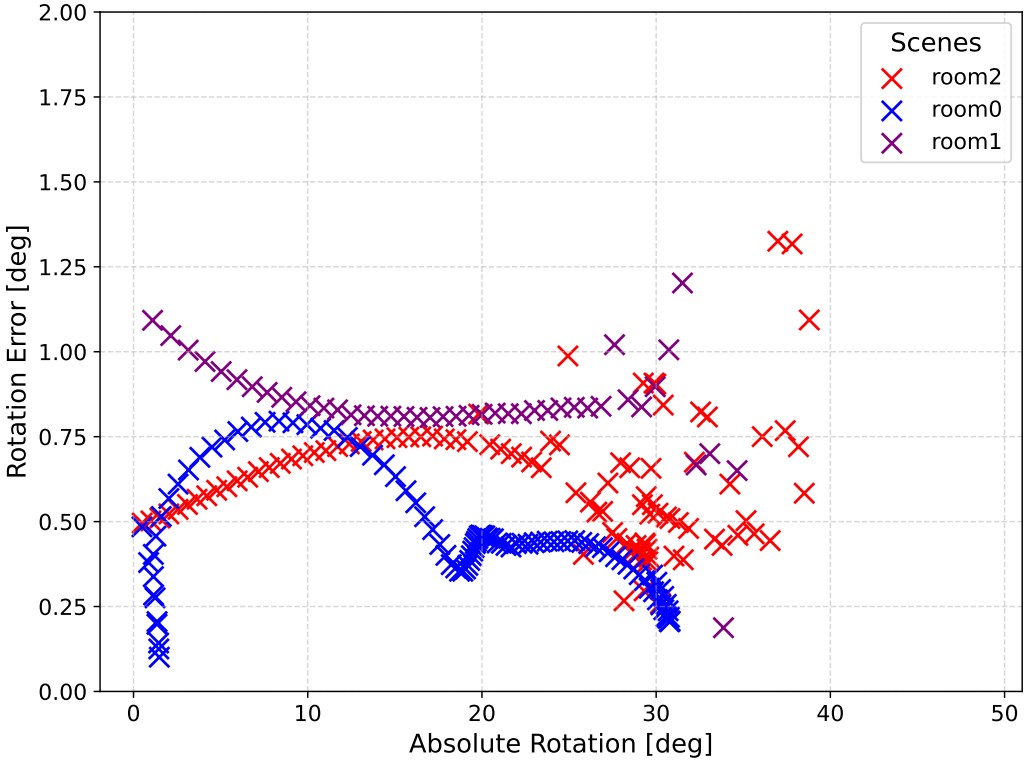

Figure H45: Rotation error under varying levels of absolute rotation, demonstrating consistent pose estimation accuracy across diverse environments despite large angular deviations.

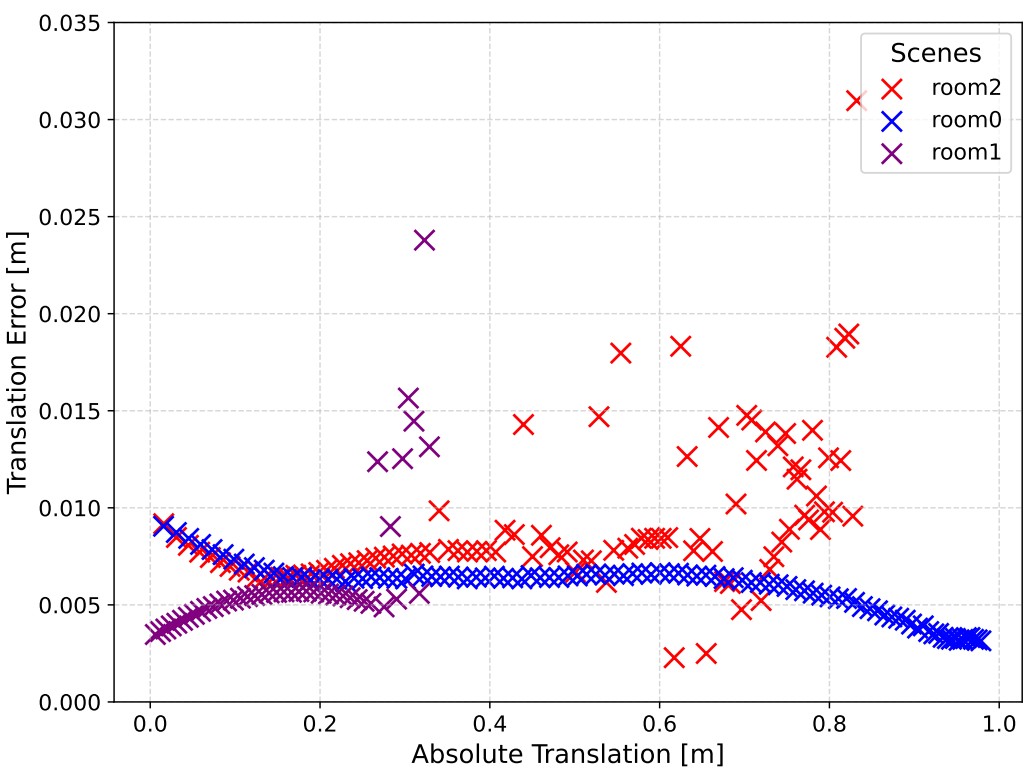

Figure H46: Translation error under increasing absolute translation, highlighting the robustness of the method across scenes even with significant translational discrepancies.

## H.5 Additional CorrGS Results in Outdoor Settings

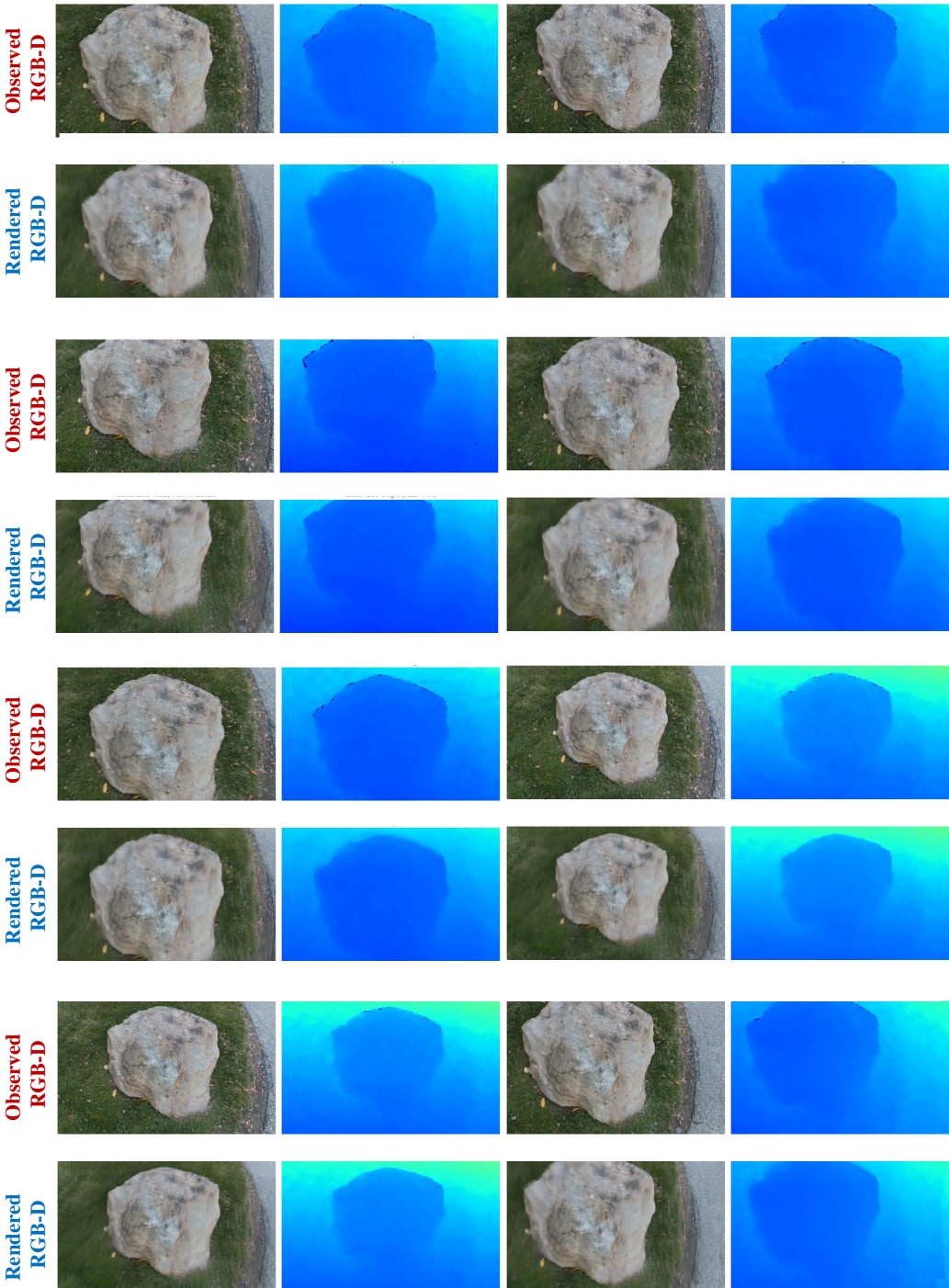

Figure H47: Rendered RGB-D results from the reconstructed map in the outdoor scene during the morning. The average PSNR is 28 [dB] and average depth L1 loss is 1.80 [cm].

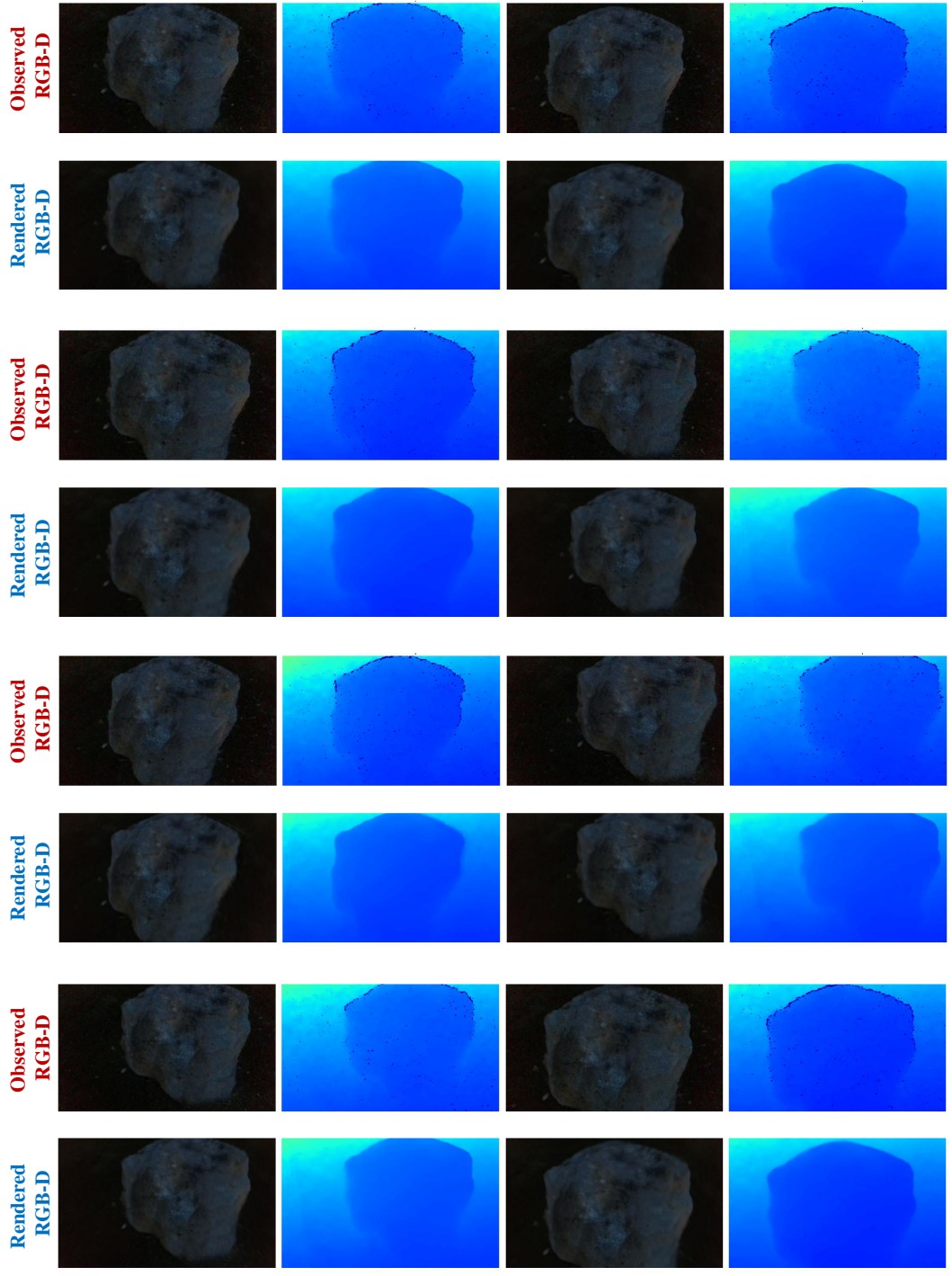

Figure H48: Rendered RGB-D results from the reconstructed map in the outdoor scene during the night. The average PSNR is 34 [dB] and average depth L1 loss is 2.17 [cm].

# I    THEORETICAL INSIGHTS ON CORRGS

We begin by analyzing the causes of inefficiency and failure in differentiable pose optimization, specifically for the Gaussian Splat map representation, under fast pose motion. We then develop a theorem to demonstrate that incorporating Correspondence-Guided Pose Learning (CPL) and Correspondence-Guided Appearance Restoration Learning (CARL), which constructs our CorrGS framework, enhances robustness under these challenging conditions.

**Theorem F.** *In differentiable rendering-based SLAM systems utilizing Gaussian Splat representation, incorporating Correspondence-Guided Pose Learning (CPL) and Correspondence-Guided Appearance Restoration Learning (CARL) mitigates the inefficiency and potential failure in pose optimization under fast motion and noisy video inputs. Specifically, CPL reduces the magnitude of the gradients and improves the conditioning of the Hessian matrix by providing better pose initialization, while CARL reduces the variability in the gradients caused by noise, leading to more stable and efficient convergence of the optimization process.*

*Proof.* We analyze the effects of CPL and CARL on the pose optimization problem under the Gaussian Splat representation, focusing on how they address the issues caused by large gradients and poor conditioning of the Hessian matrix under fast motion and noisy inputs.

## I.1    POSE OPTIMIZATION IN GAUSSIAN SPLAT REPRESENTATION

In differentiable rendering-based SLAM with Gaussian Splat representation, the camera pose $\mathcal{P} = \{\mathbf{R}, \mathbf{t}\}$ is parameterized using a unit quaternion $\mathbf{q} = [q_r, q_i, q_j, q_k]^T \in \mathbb{R}^4$ for rotation and a translation vector $\mathbf{t} \in \mathbb{R}^3$. The rotation matrix $\mathbf{R}$ is derived from the quaternion $\mathbf{q}$:

$$\mathbf{R} = 2 \begin{pmatrix} \frac{1}{2} - (q_j^2 + q_k^2) & q_i q_j - q_r q_k & q_i q_k + q_r q_j \\ q_i q_j + q_r q_k & \frac{1}{2} - (q_i^2 + q_k^2) & q_j q_k - q_r q_i \\ q_i q_k - q_r q_j & q_j q_k + q_r q_i & \frac{1}{2} - (q_i^2 + q_j^2) \end{pmatrix}. \tag{I60}$$

The loss function $\mathcal{L}(\mathcal{P})$ measures the discrepancy between the rendered images $(\hat{\mathbf{I}}, \hat{\mathbf{D}})$ and the observed images $(\mathbf{I}, \mathbf{D})$:

$$\mathcal{L}(\mathcal{P}) = \lambda_C \left| \hat{\mathbf{I}}(\mathcal{P}) - \mathbf{I} \right|^2 + \lambda_D \left| \hat{\mathbf{D}}(\mathcal{P}) - \mathbf{D} \right|^2, \tag{I61}$$

where $\lambda_C, \lambda_D > 0$ are weighting parameters.

The pose is updated iteratively using gradient descent:

$$\mathcal{P}_{k+1} = \mathcal{P}_k - \eta \nabla_{\mathcal{P}} \mathcal{L}(\mathcal{P}_k), \tag{I62}$$

where $\eta > 0$ is the learning rate.

## I.2    IMPACT OF FAST MOTION ON GRADIENTS AND HESSIAN

Under fast motion, the change in pose $\Delta\mathcal{P} = \Delta\mathbf{R}, \Delta\mathbf{t}$ between successive frames is large. This leads to large residuals in the loss function and consequently large gradients and Hessian entries.

**Gradients *w.r.t.* translation t.** The projected 2D coordinates $\mathbf{m}_i$ of a 3D Gaussian $G_i$ depend on the camera pose $\mathcal{P}$. The gradient of the projected coordinates with respect to translation $\mathbf{t}$ is:

$$\frac{\partial \mathbf{m}_i}{\partial \mathbf{t}} = \frac{\partial \mathbf{m}_i}{\partial \mathbf{X}_i^c}, \quad \text{where} \quad \mathbf{X}_i^c = \mathbf{R}\mathbf{X}_i + \mathbf{t}, \tag{I63}$$

and $\mathbf{X}_i$ is the 3D position of the Gaussian. The gradients are given by:

$$\frac{\partial \mathbf{m}_i}{\partial \mathbf{t}} = \begin{pmatrix} \frac{\mathbf{f}_x}{z_c} & 0 & -\frac{\mathbf{f}_x x_c}{z_c^2} \\ 0 & \frac{\mathbf{f}_y}{z_c} & -\frac{\mathbf{f}_y y_c}{z_c^2} \end{pmatrix}, \tag{I64}$$

where $\mathbf{f}_x$ and $\mathbf{f}_y$ are the focal lengths, and $(x_c, y_c, z_c)^T = \mathbf{X}_i^c$.

Under large translations $\Delta \mathbf{t}$, the terms involving $-\frac{\mathbf{f}_x x_c}{z_c^2}$ and $-\frac{\mathbf{f}_y y_c}{z_c^2}$ become large, resulting in large gradients $\frac{\partial \mathbf{m}_i}{\partial \mathbf{t}}$. This is because significant shifts in the 3D points relative to the camera amplify the effect of translation on the projected coordinates.

**Gradients *w.r.t.* rotation $\mathbf{R}$ (Quaternion $\mathbf{q}$).** The gradients of the transformed 3D points with respect to the quaternion components are:

$$\frac{\partial \mathbf{X}_i^c}{\partial q_r} = 2 \begin{pmatrix} q_r x_c + q_j z_c - q_k y_c \\ q_r y_c + q_k x_c - q_i z_c \\ q_r z_c + q_i y_c - q_j x_c \end{pmatrix}, \tag{I65}$$

$$\frac{\partial \mathbf{X}_i^c}{\partial q_i} = 2 \begin{pmatrix} q_i x_c - q_j y_c - q_k z_c \\ q_i y_c + q_j x_c + q_r z_c \\ q_i z_c + q_r y_c - q_k x_c \end{pmatrix}, \tag{I66}$$

$$\frac{\partial \mathbf{X}_i^c}{\partial q_j} = 2 \begin{pmatrix} q_j x_c + q_i y_c - q_r z_c \\ q_j y_c - q_i x_c - q_k z_c \\ q_j z_c + q_k y_c + q_r x_c \end{pmatrix}, \tag{I67}$$

$$\frac{\partial \mathbf{X}_i^c}{\partial q_k} = 2 \begin{pmatrix} q_k x_c - q_r y_c + q_i z_c \\ q_k y_c + q_r x_c - q_j z_c \\ q_k z_c - q_i x_c - q_j y_c \end{pmatrix}. \tag{I68}$$

Under large rotations $\Delta \mathbf{R}$, the rotation gradients become large due to significant changes in the rotation matrix $\mathbf{R}$. This leads to substantial changes in the positions of the 3D Gaussians when projected into the 2D image plane.

**Impact on Hessian matrix.** The Hessian matrix $\mathbf{H}$ of the loss function with respect to the pose parameters includes second derivatives such as:

$$\mathbf{H} = \nabla_{\mathcal{P}}^2 \mathcal{L} = \begin{pmatrix} \frac{\partial^2 \mathcal{L}}{\partial q_r^2} & \frac{\partial^2 \mathcal{L}}{\partial q_r \partial q_i} & \cdots \\ \frac{\partial^2 \mathcal{L}}{\partial q_i \partial q_r} & \frac{\partial^2 \mathcal{L}}{\partial q_i^2} & \cdots \\ \vdots & \vdots & \ddots \end{pmatrix}. \tag{I69}$$

Large gradients result in large entries in the Hessian matrix, including off-diagonal elements representing interactions between different pose parameters. This indicates strong coupling between the parameters, leading to poor conditioning of the Hessian and making the optimization landscape more challenging.

### I.3    Inefficiency and Potential Failure in Pure Differentiable Pose Optimization

In gradient-based optimization, the pose update is:

$$\mathcal{P}_{k+1} = \mathcal{P}_k - \eta \nabla_{\mathcal{P}} \mathcal{L}(\mathcal{P}_k). \tag{I70}$$

With large gradients due to fast motion, the update step $\eta \nabla_{\mathcal{P}} \mathcal{L}(\mathcal{P}_k)$ can be excessively large, causing the optimization to overshoot and potentially diverge.

Using a Taylor series expansion of the loss function around $\mathcal{P}_k$:

$$\mathcal{L}(\mathcal{P}_{k+1}) \approx \mathcal{L}(\mathcal{P}_k) - \eta \left| \nabla_{\mathcal{P}} \mathcal{L}(\mathcal{P}_k) \right|^2 + \frac{1}{2} \eta^2 \nabla_{\mathcal{P}} \mathcal{L}(\mathcal{P}_k)^\top \mathbf{H} \nabla_{\mathcal{P}} \mathcal{L}(\mathcal{P}_k). \tag{I71}$$

For the loss to decrease, we require:

$$\eta < \frac{2 \left| \nabla_{\mathcal{P}} \mathcal{L}(\mathcal{P}_k) \right|^2}{\nabla_{\mathcal{P}} \mathcal{L}(\mathcal{P}_k)^\top \mathbf{H} \nabla_{\mathcal{P}} \mathcal{L}(\mathcal{P}_k)}. \tag{I72}$$

With large gradients and poor conditioning of the Hessian, the permissible $\eta$ becomes very small. If $\eta$ is not adjusted accordingly, the optimization can diverge, or convergence can be extremely slow.

### I.4 EFFECT OF CORRESPONDENCE-GUIDED POSE LEARNING

CPL mitigates these issues by providing a better initial pose estimate $\mathcal{P}_t^*$ through minimizing the geometric error between correspondences:

$$\mathcal{P}_t^* = \arg\min_{\mathcal{P}_t} \sum_{i=1}^{N} |\mathbf{R}_t \mathbf{p}_{r,i} + \mathbf{t}_t - \mathbf{p}_{o,i}|^2, \tag{I73}$$

where $\mathbf{p}_{r,i}$ and $\mathbf{p}_{o,i}$ are corresponding 3D points from the rendered and observed frames, respectively.

By solving this optimization problem, CPL provides a pose estimate closer to the true pose $\mathcal{P}_t^{\text{true}}$. This reduces the initial residuals in the loss function:

$$\left| \hat{\mathbf{I}}(\mathcal{P}_t^*) - \mathbf{I} \right|^2 \ll \left| \hat{\mathbf{I}}(\mathcal{P}_{t-1}) - \mathbf{I} \right|^2, \tag{I74}$$

and similarly for $\hat{\mathbf{D}}$. Consequently, the magnitude of the gradients $\nabla_{\mathcal{P}} \mathcal{L}(\mathcal{P}_t^*)$ is reduced.

With a better initial pose, the Hessian matrix $\mathbf{H}$ is better conditioned near the true pose. The second derivatives become smaller, and the interactions between parameters are less pronounced. The condition number of the Hessian decreases:

$$\kappa(\mathbf{H}(\mathcal{P}_t^*)) < \kappa(\mathbf{H}(\mathcal{P}_{t-1})), \tag{I75}$$

where $\kappa(\mathbf{H})$ denotes the condition number of the Hessian matrix at pose $\mathcal{P}$. A lower condition number indicates better conditioning, which enhances the convergence properties of the optimization algorithm.

### I.5 EFFECT OF CORRESPONDENCE-GUIDED APPEARANCE RESTORATION LEARNING

Noisy video inputs introduce variability into the observed images $\mathbf{I} = \mathbf{I}^{\text{true}} + \mathbf{N}$, where $\mathbf{N}$ represents additive noise (e.g., Gaussian noise).

This noise affects the residuals in the loss function:

$$\hat{\mathbf{I}}(\mathcal{P}) - \mathbf{I} = \left( \hat{\mathbf{I}}(\mathcal{P}) - \mathbf{I}^{\text{true}} \right) - \mathbf{N}. \tag{I76}$$

The noise term $\mathbf{N}$ introduces variability into the gradients:

$$\nabla_{\mathcal{P}} \mathcal{L}_C = 2\lambda_C \nabla_{\mathcal{P}} \hat{\mathbf{I}}^\top \left( \hat{\mathbf{I}}(\mathcal{P}) - \mathbf{I}^{\text{true}} - \mathbf{N} \right). \tag{I77}$$

The term involving $\mathbf{N}$ adds stochasticity to the gradient, leading to increased variance and potential misalignment in the gradient direction.

CARL mitigates this issue by learning a restoration model $f(\cdot; \theta)$ to obtain denoised images $\tilde{\mathbf{I}} = f(\mathbf{I}; \theta) \approx \mathbf{I}^{\text{true}}$.

Using the denoised images in the loss function:

$$\mathcal{L}_C(\mathcal{P}) = \lambda_C \left| \hat{\mathbf{I}}(\mathcal{P}) - \tilde{\mathbf{I}} \right|^2, \tag{I78}$$

the residuals become:

$$\hat{\mathbf{I}}(\mathcal{P}) - \tilde{\mathbf{I}} \approx \hat{\mathbf{I}}(\mathcal{P}) - \mathbf{I}^{\text{true}}, \tag{I79}$$

eliminating the noise term $\mathbf{N}$. Consequently, the gradients:

$$\nabla_{\mathcal{P}} \mathcal{L}_C = 2\lambda_C \nabla_{\mathcal{P}} \hat{\mathbf{I}}^\top \left( \hat{\mathbf{I}}(\mathcal{P}) - \mathbf{I}^{\text{true}} \right), \tag{I80}$$

are free from the variability introduced by noise, leading to more stable and accurate gradient computations.

## I.6 COMBINED IMPACT OF CPL AND CARL

By integrating CPL and CARL, we address both the issues of large gradients due to fast motion and the variability in gradients due to noisy inputs.

**CPL reduces initial residuals and gradients.** Providing a better initial pose estimate $\mathcal{P}_t^*$ reduces the magnitude of the gradients and improves the conditioning of the Hessian matrix, allowing for a larger permissible learning rate $\eta$ without risking divergence.

**CARL stabilizes gradient computations.** By denoising the observed images, CARL reduces the variability in the residuals and gradients caused by noise, leading to more stable and reliable updates during optimization.

**Synergistic effect.** The improved pose estimate from CPL enhances the alignment between the rendered and observed images, which in turn improves the effectiveness of CARL in restoring the appearance. Accurate pose estimation ensures that correspondences used in CARL are valid, leading to better denoising performance.

## I.7 OVERALL IMPROVEMENT IN OPTIMIZATION EFFICIENCY OF CORRGS

The combined effect of CPL and CARL leads to:

- **Reduced gradient magnitudes**, mitigating the risk of overshooting and divergence.

- **Improved conditioning of the Hessian matrix**, enhancing convergence rates.

- **Stabilized gradient directions**, ensuring consistent progress towards the optimal pose.

- **Enhanced robustness** under fast motion and noisy video inputs.

Therefore, incorporating CPL and CARL into differentiable rendering-based SLAM systems utilizing Gaussian Splat representation effectively mitigates the inefficiency and potential failure in pose optimization under challenging conditions, leading to more robust and efficient convergence of the optimization process.

$\square$

Table J25: **Effects of perturbing the stereo camera baseline length on trajectory estimation performance of the ORBSLAM3 model with stereo (Top) and stereo-inertia (Bottom) SLAM setting**. We leverage three sequences, including *MH01*, *MH02*, and *MH03*, on the EuRoC (Burri et al., 2016) dataset. We introduce random noise $\eta$ along the axis of the stereo camera baseline. The noise is sampled from a Gaussian distribution with zero mean and variance $\sigma^2$, represented as $\eta \sim \mathcal{N}(0, \sigma^2)$ [m]. The stereo camera has a predefined baseline length of 0.1 meters.

| | *MH01* Sequence | | | *MH02* Sequence | | | *MH03* Sequence | | |
| --- | --- | --- | --- | --- | --- | --- | --- | --- | --- |
| **Metrics** | $\sigma = 0.00$ | $\sigma = 0.001$ | $\sigma = 0.01$ | $\sigma = 0.00$ | $\sigma = 0.001$ | $\sigma = 0.01$ | $\sigma = 0.00$ | $\sigma = 0.001$ | $\sigma = 0.01$ |
| Stereo Setting | | | | | | | | | |
| *ATE-w/o Scale*↓ [m] | **0.036** | 1.646 | 8.284 | **0.016** | 2.943 | 19.42 | **0.028** | 1.444 | 6.211 |
| *Scale* | 1.006 | 1.129 | 0.261 | 0.998 | 2.251 | 0.172 | 0.998 | 0.941 | 0.395 |
| *ATE-w/ Scale*↓ [m] | **0.024** | 1.580 | 2.221 | **0.013** | 1.674 | 2.415 | **0.027** | 1.429 | 1.985 |
| Stereo-inertia Setting | | | | | | | | | |
| *ATE-w/o Scale*↓ [m] | **0.068** | 0.353 | 5.470 | **0.050** | 2.370 | 3.954 | **0.053** | 0.572 | 673.8 |
| *Scale* | 1.012 | 0.987 | 0.410 | 1.004 | 2.172 | 0.491 | 1.004 | 1.018 | 0.001 |
| *ATE-w/ Scale*↓ [m] | **0.046** | 0.349 | 1.865 | **0.046** | 0.312 | 1.024 | **0.052** | 0.568 | 3.576 |

## J    OTHER POTENTIAL DIRECTIONS TO EXPLORE

Towards more robust SLAM for spatial perception, we present additional potential research avenues.

### J.1    ROBUSTNESS EVALUATION FOR MORE DIVERSE SLAM SYSTEMS

**We believe it is valuable to extend the robustness study to a broader range of SLAM systems, beyond the dense RGB-D SLAM framework considered in this research.** Different SLAM architectures face distinct challenges when exposed to real-world noise and perturbations.

To illustrate this, we present two case studies that demonstrate the fragility of stereo and multi-agent SLAM systems when subjected to perturbations. These findings emphasize the importance of future robustness evaluations across diverse SLAM configurations to ensure their resilience.

**Robustness analysis of stereo SLAM system under perturbation**. We conduct a robustness analysis of a stereo SLAM system, which utilizes two cameras to capture observations of the environment. In this study, we focus on the inter-camera transformation matrix, a crucial sensor parameter that describes the transformation between the left and right cameras. Our objective is to investigate the effects of introducing Gaussian noise as the perturbation to the inter-camera transformation matrix of the stereo SLAM system, which mimics the inaccuracy of the transformation matrix. This inaccuracy can be caused by factors such as noisy extrinsic matrix calibration, vibration due to a less rigid camera holder, or errors in camera placement or assembly. Specifically, we introduce random noise $\eta$ along the stereo camera baseline direction, sampled from a Gaussian distribution with zero mean and variance $\sigma^2$, denoted as $\eta \sim N(0, \sigma^2)$. In our preliminary robustness study, we utilize the ORB-SLAM3 model with the stereo camera input setting and the stereo-inertia setting. We apply random perturbations to the camera baseline length for every frame in three sequences (MH01, MH02, and NM03) from the EuRoC dataset (Burri et al., 2016). The results of our study are summarized in Table J25, which includes three severity levels of noises: $\sigma = 0.00$, $\sigma = 0.001$, and $\sigma = 0.01$. Under the clean setting ($\sigma = 0.00$). When we introduced perturbations to the baseline axis of the inter-camera transformation matrix, increasing the noise levels (*e.g.*, $\sigma = 0.001$ and $\sigma = 0.01$) resulted in a noticeable rise in trajectory estimation error, which is quantified by *ATE-w/o Scale* and *ATE-w/ Scale* metrics. This indicates a higher susceptibility to errors and deviations from the ground truth trajectory during trajectory estimation. These findings highlight the significance of taking a holistic approach when considering the robustness of the whole SLAM system. To ensure a comprehensive evaluation of the SLAM model's robustness, it is crucial to consider not only the software (SLAM model) but also the hardware, specifically the inaccuracies in predefined or estimated sensor configurations.

**Robustness analysis of multi-agent SLAM system under perturbation.**. We investigate the robustness of the multi-agent SLAM framework COVINS-G (Patel et al., 2023b) under Gaussian noise image-level perturbation. We evaluate the performance of the model on the EuRoC (Burri et al., 2016) dataset using the ORB-SLAM3 (Campos et al., 2021) model with monocular-inertia

Table J26: **Effects of Gaussian noise perturbation on trajectory estimation performance** for the multi-agent SLAM framework COVINS-G (Patel et al., 2023b) with the ORBSLAM3 (Campos et al., 2021) model as the front-end are presented. The noise is sampled from a Gaussian distribution $\eta \sim \mathcal{N}(0, \sigma^2)$. The experiments were conducted using a mono-inertia input setting for each agent.

| Metrics | $\sigma^2 = 0.00$ | $\sigma^2 = 0.02$ | $\sigma^2 = 0.04$ |
|---|---|---|---|
| ATE↓ [cm] | **0.071** | 0.071 | 0.088 |
| SSE↓ [cm] | **11.545** | 12.517 | 20.354 |

input as the front-end module. The experimental setup involves introducing varying severity levels of perturbation, measured by the parameter $\sigma$, on all the agents of a multi-agent system. The results are reported based on running five agents in sequential order on the *MH* scene of EuRoC (Burri et al., 2016). Two metrics, Average Trajectory Error (ATE) and Sum of Squared Errors (SSE), are used to assess the multi-agent SLAM system's performance. These metrics provide a comprehensive evaluation of trajectory estimation accuracy and the accumulation of errors, respectively. As is shown in Table J26, the multi-agent SLAM system demonstrates consistent performance in the clean setting ($\sigma = 0.00$) and low perturbation severity ($\sigma = 0.02$), indicating the system's ability to tolerate low random noise. However, as the perturbation severity increases to $\sigma = 0.04$, the system's performance degrades noticeably. The ATE experiences a significant increase, suggesting the vulnerability of the model to higher levels of perturbation. Similarly, the SSE metric rises to 20.354 cm, reflecting the accumulation of errors caused by the higher noise level.

## J.2 ADDITIONAL FUTURE DIRECTIONS

**Robustness evaluation in unbounded 3D scene.** Our work does not encompass the robustness of SLAM systems in unbounded scenes, such as outdoor environments (Dosovitskiy et al., 2017). Investigating the robustness of SLAM systems in such scenarios holds significant potential for future research. It can contribute to a better understanding of the practical applicability and generalizability of SLAM systems in complex scenes.

**Computationally-efficient robustness evaluation.** Our findings have identified discernible indicators within certain SLAM models that can reflect degraded observations, *i.e.*, the reconstruction quality of RGB images and depth maps for the SplaTAM-S (Keetha et al., 2024) model. Future research could explore leveraging and designing robustness indicators to evaluate the robustness of SLAM systems more efficiently. By incorporating such indicators, we have the potential to enable unsupervised performance evaluation of SLAM, especially in scenarios where obtaining ground-truth annotations is challenging or costly.

**Real-world robustness evaluation.** While our work primarily focuses on synthesis-based robustness analysis, we recognize the value of real-world verification and validation for SLAM systems. Conducting extensive field tests in more challenging environments (Ebadi et al., 2023), where SLAM systems are subjected to agile locomotion types (Kaufmann et al., 2023), would provide empirical validation of simulation results and uncover additional challenges. This real-world evaluation would bridge the gap between simulated environments and actual deployment conditions, ensuring the practical reliability and robustness of SLAM systems.

**Future directions for CorrGS.** While our method presents a proof-of-concept approach, there is room for further enhancement. In future work, we aim to explore the scalability of CorrGS in outdoor environments with unpredictable motion patterns and large-scale scenes. These environments present unique challenges such as dynamic lighting and large-scale motion, and further research will focus on improving generalization to handle these complexities. Additionally, we plan to explore the trade-offs between computational efficiency and robustness, optimizing CorrGS for real-time applications with limited resources through the use of lightweight architectures and GPU-based parallel processing. Furthermore, improving CorrGS's robustness to erratic motion and extreme lighting conditions (e.g., sudden changes from sunlight to shadows) is another important avenue. Finally, we intend to enhance computational efficiency by incorporating adaptive learning mechanisms that activate only under frame quality degradation, reducing unnecessary processing.

## K  BROADER IMPACT

The impact of this work goes far beyond robotics and Dense Neural SLAM. By enhancing robustness under noisy sensing conditions, we address a fundamental challenge faced by autonomous systems—reliable perception and localization under noise, motion, and misaligned sensors. This improvement is crucial for safety and efficiency in systems such as self-driving cars and drones, enabling them to navigate and operate with greater confidence in unpredictable environments.

Central to this work is the first comprehensive taxonomy of RGB-D perturbations for mobile agents, which serves as a flexible tool for understanding and improving robustness across diverse applications. This taxonomy, which categorizes key disruptions like motion blur and sensor desynchronization, is broadly applicable in fields like augmented reality, 3D reconstruction, embodied AI, and surgical robotics. By providing a structured framework, it enables these systems to better handle real-world imperfections.

The Robust-Ego3D benchmark, built on this taxonomy, provides a groundbreaking platform for evaluating SLAM algorithms under real-world conditions, offering insights into model vulnerabilities and driving the development of more resilient solutions. Researchers across computer vision and AI can leverage this benchmark to refine their models in complex environments, broadening the utility of SLAM technology.

Furthermore, the CorrGS method demonstrates how visual correspondences can empower SLAM to maintain high-quality 3D reconstructions even from sparse, noisy inputs. This opens up new possibilities for deploying autonomous systems in resource-constrained or high-stakes environments, such as search and rescue, space exploration, and industrial inspection.

By pushing the frontier of robustness alongside accuracy, this work provides the tools and frameworks essential for building AI systems that can thrive in real-world, unpredictable conditions. These advances are critical to ensuring that AI can be trusted in safety-critical and mission-driven domains.

## L  SOCIAL IMPACT

The proposed pipeline for noisy data synthesis and the *Robust-Ego3D* benchmark have the potential to advance the development of robust SLAM systems. By enabling the evaluation of SLAM models under diverse perturbations, this work can facilitate the creation of more resilient robotic systems capable of operating reliably in unstructured and challenging environments. Robust SLAM systems can enhance safety, efficiency, and effectiveness in various domains, such as autonomous navigation, exploration, and mapping in hazardous or inaccessible areas.

However, it is essential to acknowledge potential negative implications. The synthesized perturbations and noisy environments, while designed to evaluate robustness, might not fully capture the complexities of real-world scenarios. Overreliance on these simulated environments could lead to overlooking unforeseen challenges, highlighting the importance of complementing simulations with real-world testing and validation. While we curated diverse perturbations, inherent biases or blind spots in the taxonomy or synthesis process may remain. We emphasize the need for continuous refinement and expansion of the perturbation taxonomy to capture a broader range of real-world disturbances and mitigate potential biases. Additionally, the potential for unethical misuse of robust SLAM systems should be considered. Manipulated or biased data could be used to construct environments that present a distorted view of reality, leading to erroneous decision-making or harmful consequences. To mitigate this risk, robust validation, fact-checking mechanisms, and adherence to ethical guidelines must be integrated into the development and deployment processes.

## M    AVAILABILITY AND MAINTENANCE

The code and datasets from this study will be publicly accessible. The project repository contains the following resources:

- **Robustness benchmark**, The repository includes code for SLAM robustness evaluation under customized perturbations.
- **Baseline models for benchmarking**. Detailed instructions are provided for running all baseline models, facilitating the reproduction of all results presented in this paper.
- **Experiment reproduction**. Comprehensive guidelines for reproducing all experiments can be found in the *Instructions.md* file within the repository.
- **CorrGS: the proposed method**. The repository contains the implementation of the proposed CorrGS method, along with detailed instructions for running experiments that demonstrate its effectiveness in addressing robustness challenges in SLAM systems.

These resources enable researchers to utilize and extend this work.

We encourage the community to propose more robust SLAM models to further advance the frontier of robust embodied agents, ensuring their safe and reliable deployment in real-world environments.

## N    LICENSE

The benchmark and code are licensed under Apache License 2.0.

## O    PUBLIC RESOURCES USED

We acknowledge the following public resources used in this work:

- Classification-Robustness[3] ....................................... Apache License 2.0
- Replica[4] ................................................ Research-only License
- Nice-SLAM[5] .............................................. Apache License 2.0
- Co-SLAM[6] ............................................... Apache License 2.0
- SplaTAM[7] ............................. BSD 3-Clause "New" or "Revised" License
- GO-SLAM[8] ............................................... Apache License 2.0
- ORB-SLAM3[9] ................................... GNU General Public License v3.0
- LoFTR[10] ................................................ Apache License 2.0

---

[3] https://github.com/hendrycks/robustness
[4] https://github.com/facebookresearch/Replica-Dataset.
[5] https://github.com/cvg/nice-slam.
[6] https://github.com/HengyiWang/Co-SLAM.
[7] https://github.com/spla-tam/SplaTAM.
[8] https://github.com/youmi-zym/GO-SLAM.
[9] https://github.com/UZ-SLAMLab/ORB_SLAM3.
[10] https://github.com/zju3dv/LoFTR.

