# OpenReview forum: "Scalable Benchmarking and Robust Learning for Noise-Free Ego-Motion and 3D Reconstruction from Noisy Video"
_ICLR.cc/2025/Conference — ICLR 2025 Poster_

### Official Review · Reviewer_nb61 · 2024-10-28

**Soundness:** 3
**Presentation:** 3
**Contribution:** 3
**Rating:** 8
**Confidence:** 4

**Summary:**

The paper presents a benchmark to evaluate dense neural SLAM systems that allow photo-realistic renderings. The benchmark starts from synthetic data (Replica) and considers several data degradation methods including trajectory perturbation, sensor brightness adjustment, motion blurs, etc. The authors compare several different neural SLAM methods including iMap, NICE-SLAM, and SplaTAM, as well as a traditional ORB-SLAM3 method. Extensive analysis and comparisons are conducted to understand the effect of pose accuracy under different noises added to Replica. The paper also presents a new method named CorrGS that leverages correspondence-initialized poses to improve the initialization of the system, which is demonstrated to work well on the benchmark and surpasses existing methods.

**Strengths:**

1. The overall contributions made by this paper are pretty extensive, with a benchmark, a thorough analysis of existing methods, and a new proposed CorrGS method that improves over the existing Neural-SLAM methods. The benchmark considers many aspects that a real capture could differ from synthetic captures that are typically overlooked by existing methods, and carefully decouples these factors into several fronts and provides their influence on the trajectory estimation accuracy.

2. While many benchmarks have been provided for real-world SLAM systems, this paper focuses specifically on Neural SLAM systems where the end product is not only camera trajectories but also a neural representation that allows photo-realistic renderings. This demonstrates the novelty of the paper.

3. The quality of the figures and the tables are of high-quality, with many details provided and essential details presented. One can easily tell the impact of different degradation methods to the performance of the SLAM systems.

**Weaknesses:**

1. I had a hard time reading through the entire paper when I was trying to understand the technical details, and I think that the papers need some refactors. There is extensive cross-referencing to the appendix in the middle of the main text, making the flow of understanding constantly interrupted by having to refer to the details later. While the benchmark, the analysis, and the new CorrGS method are all valid contributions, it would be nice to re-prioritize their importance and put more texts and the details of the most important part in the main text, while putting other parts in the appendix. For example, the entire section 3 could be largely shortened or even removed since the mathematical formulations are rather simple and do not help convey the simple idea behind the scenes.

2. The noise added to the synthetic Replica dataset (e.g. perturbation on depth sensor imaging) does not authentically simulate the real-world depth images. Real-world ToF cameras usually expose systematic noise with planes being pushed to curves, but the paper's enhancement method simply adds independent Gaussian noise to the depth images. This does not align with real-world observations.

3. The introduced benchmark is heavily based on Replica that only contains a limited number of indoor scenes. To further improve the utility of the benchmark it is advisable that the authors investigates into more diverse capturing settings, such as outdoor scenes (where the depth camera still works) and highly-dynamic scenes.

4. While the main goal of the paper is to provide a benchmark that helps these Neural-SLAM methods perform better on real-world settings (by synthesizing real-world noises), it does not test its flagship method CorrGS on many real-world scenes. The only reconstruction results provided is Sec 6.3, where a relatively small chunk of real-world videos are reconstructed. It would be intriguing to see CorrGS being applied to more real-world datasets such as ScanNet++.

**Questions:**

I think the paper is written in a very clear way with many details already put in the appendix. I have no specific questions at the moment.

---

> ### Author Response · Authors · 2024-11-23
> **Official Comments by Authors**
>
> **Dear Reviewer `nb61`**,
>
> Thank you for your thoughtful feedback and recognition of our work. Below, we provide detailed responses to your comments and concerns. The revised manuscript has been uploaded, with all major changes highlighted in blue for clarity.
>
> ---
>
> **Q1: The problem formulation and cross-referencing can be simplified to improve readability.**
>
> **A1:** Thank you for the suggestion! We have streamlined the problem formulation and reduced the usage of  cross-referencing in the revised paper.
>
> ---
>
>
> **Q2: Depth perturbation types can be extended.**
>
> **A2:** Thank you for the suggestion! We agree that depth noise types can be further extended. While our ultimate goal is to develop a generator that accurately models real-world depth noise, this work establishes an important foundation by highlighting how simplified depth perturbations, including missing data, noise, and range limitations, effectively expose critical weaknesses in existing methods, motivating future advancements in this domain.
>
> ---
>
> **Q3: The benchmark can be extended to more diverse scenes.**
>
> **A3:** Thank you for the suggestion! As our noisy data synthesis pipeline is scene-agnostic, extending the benchmark to include perturbations in more diverse 3D scenes is straightforward. We have included the discussion in our revised paper.
>
> ---
>
> **Q4: The evaluation of CorrGS can be extended to more diverse real-world scenes, e.g., ScanNet++.**
>
> **A4:** Thank you for the suggestion! We would like to clarify that existing real-world RGB-D dense SLAM datasets, such as ScanNet++, are nearly noise-free, which is why we did not evaluate the real-world robustness of CorrGS on this dataset. Instead, we self-collected scenes under dynamic illumination and rapid motion to demonstrate its performance in the main paper. We agree that extending the evaluation to more real-world scenes with natural perturbations is valuable, and we have included this discussion in the future work paragraph of our paper.

---

> > ### Comment · Reviewer_nb61 · 2024-11-25
> >
> > Thank you for the clarification. However, I still think that proper quantitative evaluations on real-world datasets are necessary to justify the robustness of the proposed method. After all, this algorithm is intended to be used in real-world scenarios, no?

---

> ### Author Response · Authors · 2024-11-26
> **Reply to Reviewer nb61's Further Comments**
>
> Dear Reviewer `nb61`,
>
> We sincerely thank you for the insightful further feedback!
>
> We fully agree that quantitative evaluations of CorrGS on additional real-world data are essential to showcasing the robustness of our method. To address this, we have included two qualitative RGB-D rendering results from our CorrGS model, captured in a natural outdoor setting featuring a large rock on grass near a paved surface. These results were obtained both in the morning and at night, under a 10x faster motion setting, to further challenge and evaluate our method's performance.
>
> Our model achieved an average PSNR of 28 dB  for RGB and an average depth L1 loss of 1.80 cm for the morning sequence, and a PSNR of 34 dB for RGB  with an average depth L1 loss of 2.17 cm for the night sequence. These results demonstrate the capability of our approach to accurately reconstruct scene geometry while preserving high-quality appearance details. Furthermore, we observed that our method is robust to illumination changes, even under very low-light conditions at night. This highlights the effectiveness of our model in handling complex lighting variations that are typical in real-world scenarios.
>
> We acknowledge that obtaining quantitative evaluations with ground truth in real-world settings remains a challenge due to the lack of accurate reference data for RGB-D comparisons. However, we believe the results presented here serve as an effective indication of our method's robustness and practical applicability.
>
> Moreover, the range limitations of our RealSense D435i camera (0.3–3.0 m) restricted us from capturing larger scenes, which we plan to explore in future work. Nonetheless, the new results, now included in Section H.5 of the appendix, further demonstrate the effectiveness of our method in real-world scenarios, even under challenging conditions.
>
> Warm regards,
>
> The Authors of Submission 696

---

> > ### Comment · Reviewer_nb61 · 2024-11-26
> >
> > Thank you for your prompt response! Yes, I think with the two new challenging demonstrative cases, the effectiveness of the method is further justified. I will increase my rating.

---

> > > ### Author Response · Authors · 2024-11-26
> > > **Thank you for your support!**
> > >
> > > Dear Reviewer `nb61`,
> > >
> > > Thank you for your kind acknowledgment! Your thoughtful evaluation and constructive suggestions have been invaluable in helping us improve our work during the review and discussion phases.
> > >
> > > Thank you once again for your support and insights!
> > >
> > > Warm regards,
> > >
> > > The Authors of Submission 696

---

### Official Review · Reviewer_5SrC · 2024-11-03

**Soundness:** 3
**Presentation:** 3
**Contribution:** 3
**Rating:** 6
**Confidence:** 3

**Summary:**

The paper claims to have three-fold contributions. 1.A scalable noisy data synthesis pipeline that generates realistic RGB-D perturbations, addressing the gap between controlled and real-world environments for ego-motion estimation and 3D reconstruction tasks. 2. Robust-Ego3D, a benchmark based on the previous proposed pipeline that evaluates model robustness under diverse noisy conditions, setting a new standard for realistic model testing. 3. It also proposed CorrGS, a method that improves pose estimation and 3D reconstruction quality under noise, outperforming prior approaches in challenging scenarios.

**Strengths:**

This paper clearly defines the challenges posed by noise perturbations in ego-motion estimation and 3D reconstruction, offering a systematic exploration of how to evaluate and address these issues. This structured problem formulation provides a solid foundation for advancing robustness research in noisy environments and helps guide both model development and evaluation methodologies.

Author provides a very extensive set of experiments showcasing both the benchmark quality and methodology effectiveness across numerous noisy environments, demonstrating the generalizability of the solution.

The proposed scalable pipeline for generating realistic RGB-D perturbations is a major strength, as it introduces a systematic way to create customizable noisy datasets that mirror real-world conditions

The CorrGS method is an effective solution for improving pose estimation and 3D reconstruction under noise. It outperforms state-of-the-art models, particularly in challenging scenarios like fast motion.

**Weaknesses:**

While the paper identifies issues related to depth sensor noise, the evaluation and solutions for depth restoration are not as exhaustive as the RGB counterparts leaving some gaps in how to handle severe depth data inconsistencies.

While the scalable noisy data synthesis pipeline is impressive, it relies heavily on synthetic noise generation. This might not fully capture the unpredictable nature of some real-world scenarios.

This paper provides a new benchmark for evaluating ego-motion and 3D reconstruction quality in noisy environments, but it does not introduce any new evaluation metrics, relying instead on existing ones like ATE, PSNR, and Depth L1 Loss. While this is not a significant weakness, it might even be considered somewhat nit-picky to note. However, introducing new metrics specifically tailored to the unique challenges posed by noisy environments could have further strengthened the paper’s contributions.

Although the internal clean model is progressively refined, the method starts with the baseline assumption of a rough model that is iteratively improved. In some extreme real-world scenarios, where no good initialization is available, the system might initially struggle, which could be a practical limitation. Again, I acknowledge this may be somewhat nit-picky as well.

**Questions:**

Q1. How does the system perform when the initial internal model is significantly misaligned or inaccurate? Does CorrGS require a reasonable starting point to work effectively, or can it handle scenarios where initial pose estimates are far off?

Q2. While CPL refines the pose estimates, how sensitive is CorrGS to extremely bad initial pose estimates? Are there scenarios where the initial pose is so far off that CPL struggles to recover accurate alignment, even with PQV in place?

---

> ### Author Response · Authors · 2024-11-23
> **Official Comments by Authors**
>
> **Dear Reviewer `5SrC`**,
>
>
> Thank you for your thoughtful feedback and recognition of our work. Below, we provide detailed responses to your comments and concerns. The revised manuscript has been uploaded, with all major changes highlighted in blue for clarity.
>
> ---
>
> **Q1: Depth perturbation types and restoration can be extended.**
>
> **A1:** Thank you for the suggestion! 1) We agree that depth noise modeling can be expanded for more extensive study. While our long-term goal is to develop a generator that replicates real-world depth noise, this work lays the foundation by showing how simplified depth noise modeling reveals key vulnerabilities in existing methods, aiming to inspire further advancements in this area. 2) While this work primarily focuses on RGB appearance denoising, we recognize that depth noise presents a significant challenge in certain scenarios. For example, as shown in Fig.  H42 and Fig.  H44 of the appendix, severe depth noise can emerge under strong overexposure, occasionally leading to minor pose estimation inaccuracies. Recovering from depth perturbations is currently beyond the scope of the CorrGS model. However, we view this as an important avenue for future work and have included a discussion on this topic in the Sec. 6.3 section of the main paper. We appreciate your suggestion and believe it highlights a valuable direction for extending the capabilities of CorrGS.
>
> ---
>
> **Q2: Synthetic noise generation might not capture the unpredictable nature of some real-world scenarios.**
>
> **A2:** Thank you for the valuable feedback! We completely agree with the reviewer on this important point. Our long-term vision is to develop a noise generator capable of accurately replicating the complex and unpredictable nature of real-world noise. While this remains an ambitious goal, our current work takes an essential first step by highlighting that even simplified, individually modeled noise can expose significant vulnerabilities in existing dense SLAM methods. We hope this serves as a call to action for the community to prioritize research in this critical area. Additional insights on the value of using synthetic perturbations for benchmarking model robustness are provided in Section B.5 of the Appendix.
>
> ---
>
>
> **Q3: Suggestion on proposing new metrics for robustness evaluation.**
>
> **A3:** Thank you for the valuable suggestion! To ensure consistency and facilitate direct comparisons between the standard noise-free settings and the proposed noisy settings, we primarily adopt standard SLAM evaluation metrics used in clean settings. However, we recognize the importance of developing specialized metrics to address the unique challenges of noisy scenarios and will have included this as a promising direction for future research in our paper.
>
> ---
>
>
> **Q4: How does CorrGS handle extreme cases when the initial model is significantly misaligned?**
>
> **A4:** Thank you for the insightful questions! CorrGS relies on a well-defined noise-free state, derived from clean RGB-D observations in the initial video frames, as a reference for reconstructing 3D maps and poses from noisy video. Without such initial observations, the noise-free state becomes ambiguous, undermining both the reconstruction process and alignment of noisy frames.
> In cases where the initial internal model is significantly misaligned or inaccurate, CorrGS struggles to restore a noise-free 3D representation. Instead, it may produce noisy outputs or fail to reconstruct the 3D structure entirely. This limitation, which arises from the assumption that historical frames are available to establish a reference state, has been explicitly noted in the Future Work section. Establishing a robust initialization remains a critical prerequisite for our approach.
>
> ---
>
> **Q5: How sensitive is CorrGS to extremely bad initial pose estimates?**
>
> **A5:**  We thank the reviewer for highlighting the importance of robustness analysis under varied pose initialization errors. The pose quality verification (PQV) step enables strong performance under varying noise conditions, **effectively handling pose initialization errors with up to 40° rotation errors and 1.0m translation errors between views**. To illustrate this, we have included new  experimental results (please see Figures H45 and H46 in Sec. H.4 of the Appendix). These figures detail the behavior of pose estimation across diverse noise scenarios, emphasizing the framework's resilience to both rotational and translational discrepancies. We hope this additional analysis clarifies the robustness bound of our method.

---

> ### Author Response · Authors · 2024-11-26
> **Looking Forward to Discussion with You**
>
> Dear Reviewer  `5SrC`,
>
> We sincerely appreciate your time and effort in reviewing our manuscript and providing valuable feedback.
>
> ---
>
> Thank you for your recognition of our work. In response to your comments, we have made the following revisions to our manuscript:
>
> - We have clarified the scope of our depth noise modeling and highlighted its foundational role in exposing key vulnerabilities of existing methods. A discussion on extending CorrGS to handle depth perturbations is now included in Section 6.3.
> - We acknowledge the limitations of synthetic noise and have elaborated on its role as a first step toward benchmarking robustness. Insights are provided in Section B.5 of the Appendix.
> - While we maintain standard SLAM metrics for consistency, we have identified the development of robustness-specific metrics as a key future direction.
> - We have explicitly noted CorrGS’s reliance on a noise-free reference state for alignment and reconstruction and addressed its limitations under extreme misalignment in the Future Work section.
> - We have expanded our analysis of pose initialization errors, including new results in Appendix Section H.4, which demonstrate CorrGS’s resilience to rotational and translational noise.
>
> ---
>
> We hope these revisions address your concerns and welcome any further feedback.
>
> We look forward to actively participating in the Author-Reviewer Discussion session and welcome any additional feedback you might have on the manuscript or the changes we have made.
>
> Thank you for your constructive comments!
>
> ---
>
> Warm regards,
>
> The Authors of ICLR Submission 696

---

### Official Review · Reviewer_iqVc · 2024-11-03

**Soundness:** 4
**Presentation:** 2
**Contribution:** 3
**Rating:** 6
**Confidence:** 5

**Summary:**

The paper addresses the challenge of ego-motion estimation and photorealistic 3D reconstruction from noisy video data. The authors present a customizable data synthesis pipeline that generates large-scale datasets with varied noise and perturbations, simulating real-world complexities such as rapid motion and sensor degradation. This pipeline powers the proposed Robust-Ego3D benchmark, which rigorously evaluates model performance under noise, revealing that current models significantly degrade under realistic conditions. Then, benchmark state-of-the-art models using the noisy data in extensive detail. Finally, the authors propose Correspondence-guided Gaussian Splatting (CorrGS), which maintains a clean internal 3D representation, aligning it with noisy observations to improve pose estimation and 3D reconstruction fidelity.

**Strengths:**

The most novel contribution of this paper is a novel data synthesis pipeline capable of generating large-scale, customizable datasets with various noise and perturbations. This contribution fills a crucial gap in Neural SLAM research by addressing real-world challenges in ego-motion estimation and 3D reconstruction.

The experiments are extensive, meticulously designed, and methodologically sound. Each experiment is carefully aligned with the core claims of the paper, providing solid evidence to support the proposed pipeline and benchmarking framework. The evaluation spans synthetic and real-world settings, presenting a thorough analysis of model performance under diverse noisy conditions. This approach strengthens the validity and robustness of the findings.

Tables and figures effectively illustrate complex concepts, making it easier to grasp the data synthesis pipeline, experimental setup, and results.

**Weaknesses:**

The paper reads more like a technical report, with an emphasis on experimental results over theoretical analysis or in-depth discussion. While the experiments are thorough, they could benefit from additional interpretive insights that go beyond reporting outcomes. For example, rather than solely presenting quantitative improvements, the authors could analyze why CorrGS outperforms other methods under certain noise conditions, offering more theoretical or practical insights into its advantages.

Lack of Contextual Foundation: The abstract and introduction lack a thorough discussion of why noise-free data in current reconstruction models presents a significant limitation. While the paper briefly mentions that models degrade in noisy conditions, it could strengthen its foundation by discussing why these models struggle under real-world noise and how noise impacts reconstruction quality, ego-motion estimation, and mapping accuracy. To improve, the authors should provide an analysis of these limitations with references to studies or examples that highlight the common challenges noise introduces in such models in the introduction section.

The transition from Problem to Method: The paper transitions too quickly from briefly mentioning the limitations of noise-free models to discussing the proposed pipeline and benchmark, missing an opportunity to underscore the problem’s significance. Expanding on how noise introduces challenges for different aspects of 3D reconstruction and SLAM (such as depth estimation, alignment, or tracking stability) would ground the paper’s contributions more effectively. A focused narrative in the introduction that explains why noise resilience is critical would better support the rationale for the proposed solution.

Limited Analysis of Model Vulnerabilities: Although the paper introduces a robust benchmark for noisy conditions, it lacks a detailed analysis of specific vulnerabilities of existing models across different types of noise. This analysis could enrich the understanding of how and why noise affects certain models more than others and help position CorrGS as a targeted solution to these specific gaps. Adding such an analysis would make the benchmark more actionable and informative for future model improvement.

It can be understood the authors conducted extensive experiments and have a lot of content wish to express, yet the logic of proposing a novel method to fill a gap is overwhelmed by the extensive details. I think this work would be more fit for a journal instead of ICLR.

**Questions:**

What's the necessity of the problem formulation section? generate a dataset and then evaluate the model by calculating the evaluation matrix via the output and ground truth. This is a standard pipeline to conduct experiments. Why it was separately mentioned in an independent section? Please help me to understand what I missed if I misunderstood it.

---

> ### Author Response · Authors · 2024-11-23
> **Official Comments by Authors**
>
> **Dear Reviewer `iqVc`**,
>
> Thank you for your thoughtful feedback and recognition of our work. Below, we provide detailed responses to your comments and concerns. The revised manuscript has been uploaded, with all major changes highlighted in blue for clarity.
>
> ---
>
> **Q1: More insight into why CorrGS achieves better performance under noisy conditions.**
>
> **A1:**  Thank you for the comment! CorrGS outperforms prior methods through two key innovations: 1) Correspondence-guided pose learning, which enhances **geometric alignment under dynamic motion**, and 2) Correspondence-guided appearance restoration learning, which enables **effective appearance restoration for mapping**. By leveraging a progressively updated, clean 3D Gaussian Splatting representation rendered into RGB-D frames, CorrGS aligns noisy observations to achieve robust pose estimation and photorealistic reconstruction, even under severe noise and motion. These advances are validated by our experiments, demonstrating superior noise resilience and dynamic robustness.
>
> ---
>
> **Q2: For the introduction, it is suggested to further highlight the common challenges noise introduces to models.**
>
> **A2:** Thank you for the valuable feedback! We **have revised the introduction to emphasize the key challenges** noise introduces to models.
>
> ---
>
> **Q3: It is suggested to further emphasize the importance of noise resilience for the model in the abstract and introduction.**
>
> **A3:** Thank you for the valuable suggestion! We **have revised the abstract and the introduction to further highlight the importance of noise resilience**, as robust performance in real-world scenarios demands handling imperfect data caused by environmental factors, sensor limitations, and dynamic changes.
>
> ---
>
> **Q4: More analyses of specific vulnerabilities of existing models across different types of noise.**
>
> **A4:** Thank you for the valuable suggestion! **We have conducted comprehensive benchmarking analyses of model vulnerabilities under various types of noise, using perturbation settings two orders of magnitude larger than previous works.** These results, encompassing both experimental and theoretical insights, are detailed in Section 4 of the updated main paper. Additionally, we have provided extensive quantitative and qualitative analyses in the Appendix (Sections D, E, and F, pages 39–64) to explore model vulnerabilities in greater depth. We encourage the reviewer to refer to these sections for more insights.
>
> ---
>
> **Q5: What is the rationale behind the problem formulation, and is the section redundant?**
>
> **A5:** Sorry for the confusion. To clarify, the problem formulation section was intended to provide a structured overview of the key challenges addressed in our work: generating noisy video data from clean 3D models with specific poses and perturbations, developing a model capable of estimating clean 3D reconstructions and poses from noisy video inputs, and establishing an evaluation framework for perturbation-conditioned feedback to enhance model robustness. **While these elements do align with standard ML workflows, we included the section to clarify how our contributions interconnect.** However, we agree that merging this section with others would streamline the presentation, and **we have revised the content and merged it with following sections** accordingly.

---

> ### Author Response · Authors · 2024-11-26
> **Looking Forward to Discussion with You**
>
> Dear Reviewer  `iqVc`,
>
> We sincerely appreciate your time and effort in reviewing our manuscript and providing valuable feedback.
>
> ---
>
> In response to your comments, we have made the following revisions to our updated manuscript:
>
> - We have clarified how CorrGS achieves better performance under noisy conditions by leveraging correspondence-guided pose learning and appearance restoration to enhance geometric alignment and mapping.
> - We have revised the introduction and abstract to further emphasize the challenges posed by noise and the importance of noise resilience in real-world scenarios.
> - We have conducted comprehensive analyses of model vulnerabilities under various noise types, benchmarking with perturbations two orders of magnitude larger than prior works. These results are detailed in Section 4 and the Appendix.
> - We have streamlined the problem formulation section by merging it with subsequent sections for improved readability.
>
> ---
>
> We hope these revisions adequately address your concerns.
>
> We look forward to actively participating in the Author-Reviewer Discussion session and welcome any additional feedback you might have on the manuscript or the changes we have made.
>
> Thank you once again for your valuable feedback!
>
> ---
>
> Warm regards,
>
> The Authors of Submission 696

---

### Official Review · Reviewer_ccbx · 2024-11-03

**Soundness:** 3
**Presentation:** 3
**Contribution:** 3
**Rating:** 6
**Confidence:** 3

**Summary:**

The paper proposes a structured approach to advancing robust ego-motion estimation and 3D reconstruction in the presence of noisy video data. The authors tackle three core challenges: developing a scalable noisy data generation pipeline, creating the Robust-Ego3D benchmark to evaluate model robustness under varied perturbations, and introducing a novel model called Correspondence guidded Gaussian Splatting. The data generation pipeline is designed to synthesize large scale, customizable   datasets simulating realistic RGBD sensor noise and motion perturbations, which expose vulnerabilities in existing models. With these data, the Robust-Ego3D benchmark allows a systematic robustness evaluation of models across diverse noisy conditions.  CorrGS leverages correspondences between internally generated and observed frames to achieve photorealistic 3D reconstructions and robust pose tracking, significantly outperforming SOTA methods in both synthetic and real world experiments under rapid motion and complex noise.

**Strengths:**

The paper addresses the largely unexplored domain of robustness benchmarking for ego-motion and 3D reconstruction under realistic, noisy conditions. Unlike traditional methods that assume noise-free data, the paper  introduces a customizable pipeline for generating data with varied noise levels, motion deviations, and synchronization perturbations, filling a critical gap in existing SLAM evaluation techniques. Furthermore, CorrGS’s correspondence-based method is innovative, as it maintains a noise-free internal 3D representation while dynamically aligning it with noisy external inputs, a novel approach to handling real-world sensor noise. I believe the authors employ a rigorous methodological framework, incorporating a comprehensive set of experiments across both synthetic and real-world noisy datasets. They evaluate CorrGS and other baseline models on metrics including absolute trajectory rrror, PSNR, and depth loss, providing a strong quantitative foundation to support their claims. The ablation studies and theoretical analyses of perturbation effects on model performance offer a clear view of CorrGS's robustness and justify each of the model's components. The paper is clearly structured, with a logical flow from problem definition to solution details. For example, the pipeline for noisy data generation and perturbation taxonomy is visualized in Figure 1, which helps clarify complex processes. Additionally, the explanation of each perturbation type provides readers with a clear understanding of the challenges the benchmark addresses. The ablation studies further aid in understanding the impact of individual components within CorrGS on final model performance.

**Weaknesses:**

While the paper addresses RGB noise effectively, it could expand on the depth perturbations. Depth noise effects are discussed, the authors could delve further into the challenges that depth noise presents in reconstruction. Additional experiments on models’ performance under severe depth noise and missing depth data would strengthen the robustness evaluation and underscore CorrGS's adaptability. The PQV step is essential for their method’s robustness under complex conditions, but the rationale behind its design choices could be explained in greater detail. I believe it would be valuable to know if different loss functions  were tested and if so, why those choices were made. Demonstrating its impact across different noise levels would give a clearer picture of its effectiveness and necessity under varied conditions. The paper primarily compares CorrGS to other dense Neural SLAM models, but additional comparisons with classical non-neural SLAM methods would highlight CorrGS’s improvements in robustness.

**Questions:**

1. How does the pose quality verification step perform under different levels of noise? Did the authors experiment with varying PQV thresholds to understand its sensitivity to noise, and if so, what insights did they gain?
2. Does CorrGS account for scenarios with high-frequency dynamic changes?
3.Different sensor types produce different noise profiles. Did the authors experiment with adjusting CorrGS for specific types of sensor noise (such asmotion blur for RGB or range limitations for depth sensors)?
4. The Robust-Ego3D benchmark and CorrGS are tested primarily on indoor or synthetic datasets. Do the authors envision any adjustments needed to apply these methods to outdoor environments, where conditions such as changing weather, shadows, or variable terrain could add complexity?

---

> ### Author Response · Authors · 2024-11-23
> **Official Comments by Authors (Part 1)**
>
> **Dear Reviewer `ccbx`**,
>
> Thank you for your thoughtful feedback and recognition of our work. Below, we provide detailed responses to your comments and concerns. The revised manuscript has been uploaded, with all major changes highlighted in blue for clarity.
>
> ---
>
> **Q1: Can CorrGS be extended to tackle the challenge of depth perturbations?**
>
> **A1:** Thank you for the insightful suggestion. While this work primarily focuses on RGB appearance denoising, we recognize that depth noise presents a significant challenge in certain scenarios. For example, as shown in Fig.  H42 and Fig.  H44 of the appendix, severe depth noise can emerge under strong overexposure, occasionally leading to minor pose estimation inaccuracies. Recovering from depth perturbations is currently beyond the scope of the CorrGS model. However, we view this as an important avenue for future work and have included a discussion on this topic in the Sec. 6.3 of the main paper. We appreciate your suggestion and believe it highlights a valuable direction for extending the capabilities of CorrGS.
>
> ---
>
> **Q2: What is the chosen loss function for the PQV step, and what are the insights behind its design?**
>
> **A2:** Thank you for the question. In the PQV step, we use the standard RGB-D rendering loss (Eq. G34 in the Appendix) to evaluate the quality of the pose. The purpose of the PQV step is to identify and reject degenerate cases where correspondence-based pose initialization becomes unreliable. These degenerate cases typically occur when the two viewpoints are either very close together or identical. In such scenarios (see Fig. G34. of the Appendix), it is challenging to accurately estimate the relative pose by optimizing the transformation between the point clouds of the two viewpoints, as the lack of sufficient variation undermines correspondence-based methods. Please see Sec. G.3.7 of the Appendix for more details.
>
> ---
>
>
> **Q3: Comparison of CorrGS to classical non-neural SLAM methods?**
>
> **A3:** Thank you for the valuable suggestion. While our work primarily focuses on dense Neural SLAM methods for photorealistic 3D reconstruction, we have extended our comparison to include two classical non-neural SLAM methods as suggested:
> - **ORB-SLAM3:** CorrGS significantly outperforms ORB-SLAM3 under fast-motion scenarios in the 10× speed-up motion setup of our Robust-Ego3D benchmark (ATE RMSE: 13 cm of ORB-SLAM3 vs. 0.45 cm of our CorrGS), demonstrating superior robustness and accuracy. Additionally, ORB-SLAM3 loses tracking under combined fast-motion and dynamic illumination conditions, whereas CorrGS successfully handles these challenging perturbations (see Table 6 in the main paper).
>
> - **RTAB-Map:** This dense non-neural SLAM method struggles to maintain tracking in the same 10× speed-up motion  setup.
>
> These results underline the robustness and adaptability of CorrGS compared to classical approaches in challenging scenarios.
>
> ---
>
> **Q4: How does the pose quality verification step perform under varying noise levels, and does it use thresholds?**
>
> **A4:** Thank you for the valuable question!
> - **Performance under pose noise:** We thank the reviewer for highlighting the importance of robustness analysis under varied pose noise levels. The pose quality verification (PQV) step enables strong performance under varying noise conditions, effectively handling pose initialization errors with up to 40° rotation errors and 1.0m translation errors between views. To illustrate this, we have included new Figures H45 and H46 in Sec. H.4 of the Appendix. These figures detail the behavior of pose estimation error across diverse initial pose noise values, emphasizing the framework's resilience to both rotational and translational discrepancies. We hope this additional analysis clarifies the robustness of our method.
> - **Thresholding:** There is no fixed hand-crafted threshold for PQV. Unlike threshold-dependent methods, PQV employs a dynamic comparison of RGB-D rendering losses. It contrasts the losses from correspondence-guided initialization against propagated poses (e.g., extrapolated or copied from prior frames). This mechanism ensures adaptive performance by selecting the pose with the lower rendering loss, eliminating the need for predefined thresholds or hyperparameters. The adaptive nature of PQV allows it to maintain resilience against noisy inputs.
>
> ---

---

> > ### Comment · Reviewer_ccbx · 2024-11-25
> >
> > Thank you for these thoughtful responses, they have clarified the questions I had regarding this paper and  I will take into account in my re-evaluation of my score.

---

> ### Author Response · Authors · 2024-11-23
> **Official Comments by Authors (Part 2)**
>
> **Q5: Can CorrGS handle high-frequency dynamics, and does it include adaptations tailored to specific noise profiles?**
>
> **A5**: Thank you for the valuable question!
> - **Handling high-frequency dynamics.** CorrGS is designed to handle highly dynamic changes by employing online learning to adapt between the current perturbed observation and the historical map, which is maintained to be consistent and clean. In our experiments, we demonstrated CorrGS’s capability to tackle real-world scenarios, including high-dynamic video with varied illumination (e.g., transitions between darkness, normal lighting, and overexposure), depth noise, and motion blur in RGB-D data. These results are detailed in the Sec. 5.3 of the main paper and Sec. H.3 of the appendix.
> - **Adaptation to specific noise.** We did not explicitly adjust CorrGS for specific noise types in our experiments. Instead, CorrGS uniformly approaches denoising as a problem of learning the transformation between the perturbed observation and the rendered (clean) frame of the historical map, without tailoring it to individual noise profiles.
>
> ---
>
> **Q6: Any envisioned adjustments for extending Robust-Ego3D and CorrGS from indoor to outdoor?**
>
> A6: Thank you for the thoughtful question. Extending beyond static indoor scenes, outdoor environments present unique challenges that warrant careful attention:
> - **1) Dynamic elements**. The presence of moving objects, such as vehicles and pedestrians, complicates consistent ego-motion estimation and 3D reconstruction.
> - **2) Unbounded spatial scales**. Outdoor scenes often span large, unbounded spaces, posing difficulties for existing online 3D reconstruction methods, even without external disruptions.
> - **3) Evolving visual disruptions**. Factors like shifting shadows, changing weather, and varying lighting conditions add complexity to visual processing and reconstruction accuracy.
> - **4) RGB-D sensing limitations**. Commercial RGB-D cameras, such as the Intel RealSense D435i (that we use) with a depth range of up to 3 meters, impose significant constraints on dense 3D reconstruction in outdoor environments.
> To address these challenges, future work must explore strategies for both **benchmarking** and **model design** that effectively mimic these effects in controlled environments and tackle them systematically. This would involve designing datasets and models that account for dynamic, large-scale, and visually complex outdoor scenarios, thereby enhancing the robustness and applicability of our methods.

---

> ### Author Response · Authors · 2024-11-25
>
> Dear Reviewer  `ccbx`,
>
> We are pleased to hear that our responses have clarified your questions. Thank you again for your recognition of our work and for your thoughtful feedback and effort during the review process!
>
> Warm regards,
>
> The Authors of Submission 696

---

### Comment · Area_Chair_uUps · 2024-11-20

Dear reviewers and authors, this is a reminder that November 13 to November 26 at 11:59pm AoE: Reviewers and authors can exchange responses with each other as often as they wish. Thanks!

---

> ### Author Response · Authors · 2024-11-20
>
> Dear Area Chair uUps,
>
> Thank you for the reminder about the response period. We would like to express our sincere gratitude to you and the reviewers for your time and valuable feedback. We are actively preparing our responses and will engage promptly within the specified timeframe.
>
> Best regards,
>
> The Authors

---

> ### Comment · Area_Chair_uUps · 2024-11-24
>
> Dear reviewers, this is a reminder that November 26 is the Last day for reviewers to ask questions to authors.

---

### Author Response · Authors · 2024-12-04
**Summary of Author-Reviewer Discussion Session (Part 1)**

Dear Reviewers, Area Chairs, and Program Chairs,

As the Author-Reviewer Discussion concludes, we extend our sincere gratitude for your time, effort, and constructive feedback during this process. Your thoughtful suggestions have been invaluable in refining our submission.

---

We are encouraged that **all the reviewers recognize the clarity of our motivation, the comprehensiveness of our perturbation taxonomy, the systematic robustness evaluation, the novelty of our CorrGS method, and the overall readability of our paper**.

Below, we summarize the key acknowledgments from reviewers:

- **Reviewer `ccbx`**:
  - "addresses the **largely unexplored domain** of robustness benchmarking for ego-motion and 3D reconstruction under realistic, noisy conditions"
  - "filling **a critical gap** in existing SLAM evaluation techniques"
  - "CorrGS’s correspondence-based method is **innovative**"
  - "the authors employ a **rigorous methodological** framework"
  - "providing a **strong quantitative foundation** to support their claims"
  - "The **ablation studies and theoretical analyses** of perturbation effects on model performance **offer a clear view**"
  - "The paper is **clearly structured**, with a logical flow from problem definition to solution details"
- **Reviewer `iqVc`**:
  - "This contribution **fills a crucial gap** in Neural SLAM research by addressing real-world challenges in ego-motion estimation and 3D reconstruction."
  - "The **experiments are extensive, meticulously designed, and methodologically sound**."
  - "The evaluation spans synthetic and real-world settings, presenting a **thorough analysis**"
  - "**Tables and figures effectively illustrate** complex concepts"
- **Reviewer `5SrC`**:
  - "This paper **clearly defines the challenges** posed by noise perturbations in ego-motion estimation and 3D reconstruction"
  - "offering a **systematic exploration** of how to evaluate and address these issues"
  - “This structured problem formulation provides **a solid foundation for advancing robustness research** in noisy environments and **helps guide both model development and evaluation methodologies**”
  - “provides a **very extensive set of experiments** showcasing both the benchmark quality and methodology effectiveness across numerous noisy environments, demonstrating the generalizability of the solution”
  - “scalable pipeline for generating realistic RGB-D perturbations is a major strength, as it introduces **a systematic way to create customizable noisy datasets**”
  - “CorrGS method is an **effective solution** for improving pose estimation and 3D reconstruction under noise”
- **Reviewer `nb61`**:
  - "The **overall contributions** made by this paper are **pretty extensive**"
  - "The benchmark **considers many aspects that a real capture could differ from synthetic captures that are typically overlooked by existing methods**, and carefully decouples these factors into several fronts and provides their influence on the trajectory estimation accuracy."
  - "this paper focuses specifically on Neural SLAM systems where the end product is not only camera trajectories but also a neural representation that allows photo-realistic renderings. This **demonstrates the novelty of the paper**"
  - "The quality of the figures and the tables **are of high-quality**, with many details provided and essential details presented."

---

---

> ### Author Response · Authors · 2024-12-04
> **Summary of Author-Reviewer Discussion Session (Part 2)**
>
> ---
>
> We are pleased to report that our responses and revisions **have addressed all concerns raised by participating reviewers (Reviewer `ccbx` and Reviewer `nb61`)**. Additionally, **we are delighted that Reviewer `nb61` increased the evaluation score from 6 to 8** during the rebuttal stage.
>
> ---
>
> We have summarized the key improvements and clarifications below. The **revised manuscript has been uploaded**, with all major changes highlighted in blue.
>
>
> - **CorrGS Model Robustness and Real-World Applicability**
>   - Conducted extensive robustness analyses of CorrGS under varied pose initialization errors and included additional experiments **demonstrating adaptability to diverse pose noise conditions**, addressing concerns from Reviewers `ccbx` and `5SrC`.
>   - Enhanced qualitative results with **new outdoor sequences to highlight CorrGS’s potential in real-world applications**, addressing Reviewer `nb61`’s feedback.*
>   - Expanded comparisons to include classical non-neural SLAM methods (e.g., ORB-SLAM3 and RTAB-Map) to **highlight CorrGS’s superiority**, as suggested by Reviewer `ccbx`.
>
> - **Clarity and Readability Enhancements**
>   - Streamlined problem formulation and reduced cross-references to improve flow, addressing suggestions from Reviewers `iqVc` and `nb61`.
>   - Clarified the rationale behind the chosen loss function for the PQV step in CorrGS, based on Reviewer `ccbx`’s suggestion.
>   - Provided additional insights into why CorrGS achieves superior performance under noisy conditions, addressing Reviewer `iqVc`’s feedback.
>   - Highlighted the critical importance of resilience to noise for 3D reconstruction models in the introduction and abstract, incorporating Reviewer `iqVc`’s suggestion.
>
> ---
>
> We recognize the reviewers’ valuable suggestions for extending benchmarks to more diverse environments, improving depth noise modeling, and developing robustness-specific metrics. These ideas will guide our future research efforts.
>
> ---
>
> **We appreciate the thoughtful discussions and are encouraged by the positive recognition of our work.** The reviewers’ acknowledgments of our contributions and their constructive input have been instrumental in refining our paper.
>
> Thank you once again for your valuable feedback and support. We look forward to advancing this line of research in collaboration with the community.
>
> ---
>
> Warm regards,
>
> The Authors of Submission 696

---

### Meta-Review · Area_Chair_uUps · 2024-12-22

**Metareview:**

The paper tackles ego-motion estimation and 3D reconstruction under noisy conditions with three contributions: a scalable data synthesis pipeline for realistic noise, the Robust-Ego3D benchmark for evaluating robustness, and the novel CorrGS method for refining 3D reconstruction and pose estimation. CorrGS outperforms state-of-the-art methods in challenging scenarios, setting a new standard for robust 3D vision.

Strengths:
Innovative Framework: Introduces a novel data synthesis pipeline and Robust-Ego3D benchmark to tackle noisy conditions in ego-motion and 3D reconstruction.
Effective Solution: Proposes CorrGS, a method that outperforms state-of-the-art models under challenging scenarios like fast motion.
Strong Validation: Extensive experiments across synthetic and real-world datasets demonstrate robustness and generalizability.
Clear Presentation

Weaknesses:
Depth Noise Gaps: Limited evaluation of severe depth inconsistencies and realistic sensor noise.
Synthetic Bias: Relies on synthetic noise, missing real-world complexities.
Benchmark Scope: Limited to Replica's indoor scenes, lacking outdoor and dynamic scenarios.
Real-World Tests: Insufficient validation of CorrGS on diverse real-world datasets.

Although there are some weaknesses, all reviewers give positive scores based on the contributions of the work.
We are glad to accept this paper.

**Additional Comments On Reviewer Discussion:**

The authors effectively addressed reviewers’ concerns during the rebuttal:

CorrGS Robustness and Real-World Applicability: Added robustness experiments for pose noise, outdoor sequence results, and comparisons with classical SLAM methods like ORB-SLAM3, satisfying Reviewers ccbx, nb61, and 5SrC.
Clarity Improvements: Streamlined the problem formulation, clarified the PQV loss function, and added insights into noise resilience, addressing suggestions from Reviewers iqVc and nb61.
Future Work: Acknowledged suggestions to expand benchmarks, improve depth noise modeling, and explore robustness-specific metrics.
These updates, along with reviewer satisfaction, support the decision to accept.

---

### Decision · Program_Chairs · 2025-01-22

Accept (Poster)